# VCR: A Task for Pixel-Level Complex Reasoning in Vision Language Models via Restoring Occluded Text

**Tianyu Zhang**[1,2,3*], **Suyuchen Wang**[1,3*], **Lu Li**[4], **Ge Zhang**[5],
**Perouz Taslakian**[2], **Sai Rajeswar**[2], **Jie Fu**[6], **Bang Liu**[1,3,7], **Yoshua Bengio**[1,3,7]

[1] Mila, Quebec AI Institute  [2] ServiceNow Research  [3] Université de Montréal
[4] University of Pennsylvania  [5] University of Waterloo  [6] HKUST  [7] CIFAR AI Chair

{tianyu.zhang, yoshua.bengio}@mila.quebec
{suyuchen.wang, bang.liu}@umontreal.ca
luli1@upenn.edu ge.zhang@uwaterloo.ca
{perouz.taslakian, sai.rajeswar}@servicenow.com jiefu@ust.hk

## Abstract

We introduce Visual Caption Restoration (VCR), a novel vision-language task that challenges models to accurately restore partially obscured texts using pixel-level hints within images through complex reasoning. This task stems from the observation that text embedded in images intrinsically differs from common visual elements and text due to the need to align the modalities of vision, text, and text embedded in images. While many works incorporate text into image for visual question answering, they mostly rely on OCR or masked language modeling, reducing the task to text-based processing. However, text-based processing becomes ineffective in VCR as accurate text restoration depends on the combined information from provided images, context, and subtle cues from the tiny, exposed areas of masked texts. We develop a pipeline to generate synthetic images for the VCR task using image-caption pairs, with adjustable caption visibility to control the task difficulty. With this pipeline, we construct **VCR-WIKI** for VCR using Wikipedia images with captions, including $2.11M$ English and $346K$ Chinese training entities, plus $5K$ validation and $5K$ test entities in both languages, each in *easy* and *hard* configurations. We also make a hidden test set **VCR-HIDDEN** to avoid potential over-fitting on **VCR-WIKI**. Our results reveal that current vision language models significantly lag behind human performance in the VCR task, and merely fine-tuning the models on our dataset does not lead to notable improvements. We release **VCR-WIKI** and the data construction code to facilitate future research.

## 1 Introduction

Recent advances in large language models, such as ChatGPT (OpenAI et al., 2023) and Llama (Touvron et al., 2023), have spurred significant interest and progress in the field of vision-language models (VLMs). With models like GPT-4V (OpenAI et al., 2023) and LLaVA (Liu et al., 2023a; 2024a; 2023b) blending textual and visual information, the intersection of computer vision and natural language processing has become a vibrant research frontier. These integrated models aim to leverage the potential of vision and language modalities to understand and interpret multimedia content more effectively.

Amidst this evolving landscape, we introduce VCR, a novel vision-language task designed to challenge existing models uniquely. VCR challenges these models to restore obscured texts within images, which demands an intricate synthesis of text,

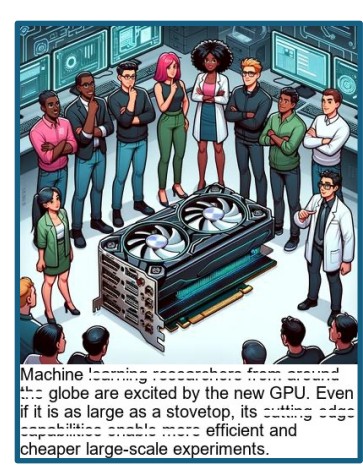

Figure 1: An example of the VCR task.

---

*Equal contribution.

vision, and text embedded in the image. The VCR task is grounded in two key insights: (1) text embedded within images, with its characteristics different from common visual elements, represents a distinct modality that requires careful alignment of vision, textual data, and the structure of written texts, and (2) neuroscience findings that suggest that humans are proficient in recognizing partially occluded objects through sophisticated visual and cognitive processes (Thinés et al., 2013; Pessoa et al., 1998; van Lier & Gerbino, 2015; Fyall et al., 2017; Li et al., 2023a). By leveraging these insights, VCR seeks to explore how well vision-language models can handle texts embedded within images, aligning visual elements and natural language to mimic human-like multimodal understanding and recognition.

The Visual Question Answering (VQA) task (Antol et al., 2015; Wang et al., 2018; Mishra et al., 2019b; Singh et al., 2019) has been a popular benchmark in assessing how well models align and interpret visual and linguistic information. Traditional VQA mainly addresses visible elements, overlooking the nuanced relationship between embedded text and image context. This highlights the limitations of current models in handling integrated visual-textual data, especially when text is obscured or altered.

To address these limitations, our VCR task builds on the premise that effective text restoration from images requires an integrated understanding beyond the capabilities of current VQA benchmarks. For example, in extreme cases, models rely on existing Optical Character Recognition (OCR) system to extract text from documents (Singh et al., 2019; Borisyuk et al., 2018). The extracted text is then used as context for generating answers without a true semantic alignment between the text and the visual elements of the document. This approach, while effective in simple scenarios, falls short in more complex settings where text is intricately woven into the visual narrative of the image.

To develop the VCR task, in this work, we introduce a pipeline for generating synthetic images that allows for manipulation of the visibility of the textual components of the image. This not only enhances the challenge posed by the task but also provides a scalable way to adjust task difficulty. The resulting dataset, **VCR-WIKI**, comprises 2.11M English data and 346K Chinese data sourced from Wikipedia, featuring captions in both languages across 'easy' and 'hard' difficulty levels. Our evaluations indicate that existing vision-language models significantly underperform compared to human benchmarks, underscoring the need for novel model architectures and training paradigms specifically geared towards this complex intermodal alignment. We also constructed a hidden test set for the VCR task (**VCR-HIDDEN**) to avoid potential over-fitting on **VCR-WIKI**.

We release **VCR-WIKI** and its construction code to stimulate further research of developing of models that can more adeptly navigate the nuanced landscape of the restoration of text embedded in images to bridge the gap between human and machine perception. The code and datasets are available at GitHub and Hugging Face.

**Contributions** The main contributions of this paper are:

**C1** Introduce the VCR task to challenge VLMs to restore occluded texts in images.

**C2** Develop a pipeline for generating synthetic images with embedded text that allows for adjusting the visibility of such text, thus providing a rich testing environment for VCR.

**C3** Create and release **VCR-WIKI**, a dataset with multilingual captions and construct the hidden test set **VCR-HIDDEN**, designed to benchmark VLMs on text restoration tasks.

**C4** Conduct empirical evaluations that show significant gaps between current models and human performance on the VCR task. This highlights the effectiveness of VCR for assessing advancements in VLMs and underscores the necessity for innovative model architectures and training techniques. New models will be actively added to our Github leaderboard.

## 2 VCR TASK DESCRIPTION

In this section, we compare the VCR task with other existing tasks and answer the following questions:

**Q1** What is the difference between VCR and other visual reconstruction tasks?

**Q2** Why should we care about VCR?

For better clarity, we define *text embedded in image ($TEI$)* as text incorporated within the image, *visual image (VI)* as the non-textual portion of the image, and *string text (ST)* as the separate textual element associated with the image (typically the question prompt). A VCR task element can thus be expressed as $(ST, (VI, TEI))$, where $ST$ is a string while $VI$ and $TEI$ are presented in image form. We adopt this notation to facilitate explanation, not to imply physical separation of $VI$ and $TEI$ in the image. Please refer to Figure 3 for an illustration of $VI$, $TEI$, and $ST$.

**A1** VCR relates to both VQA and OCR tasks. VQA takes images and questions as input to generate free-form responses with non-unique ground-truth, creating evaluation challenges regarding non-unique answers. In contrast to VQA, OCR is a task where the ground-truth responses are unique: OCR takes as input complete characters in image form and outputs a string representing the characters in the image, without considering the image context. VCR bridges these tasks by reconstructing unique text while considering visual context. Figure 2 shows a hard-mode VCR task where humans can fill blanks easily, but models with only OCR capabilities cannot recover covered text without context, as pixel-level character hints no longer yield unique solutions.

**A2** The proposed VCR task is significant in two aspects.

First, it connects to fundamental neuroscience findings on human cognitive abilities to recognize partially occluded objects (Fyall et al., 2017; Li et al., 2023a). While existing models struggle with occluded information, humans excel by combining low-level visual processing with high-level cognitive functions in the prefrontal cortex. VCR serves as a critical probe distinguishing between low-level recognition and high-level reasoning cognition—a distinction essential for advancing AI systems toward human-like perception capabilities.

Second, VCR presents a unique challenge substantially different from existing benchmarks by specifically targeting text-image alignment capabilities. Unlike traditional VQA or occluded object restoration tasks that test general visual reasoning, VCR creates a specialized evaluation framework that tests a model's ability to maintain semantic consistency across multiple modalities simultaneously. It requires deep integration of visual content ($VI$) with partially visible textual elements ($TEI$), necessitates inference capabilities that go well beyond pattern recognition toward genuine comprehension, and demands contextual reasoning that mirrors human cognitive processes when faced with incomplete information.

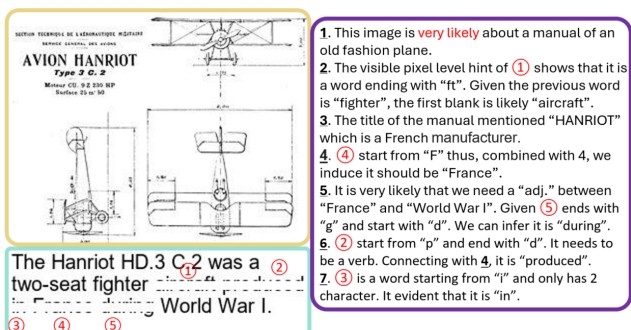

Figure 2: How humans would possibly solve a VCR task.

What makes VCR particularly valuable as a benchmark is its precision in targeting the specific frontier of vision-language integration. By occluding text rather than objects, VCR creates a controlled environment where success requires sophisticated cross-modal reasoning rather than mere recognition or memorization. The ability to adjust difficulty through varying occlusion levels provides researchers with a finely calibrated instrument to measure incremental progress in model capabilities.

For practitioners developing next-generation multimodal systems, VCR offers several distinct advantages: (1) it provides a reliable measure of text-visual alignment capabilities currently lacking in standard benchmarks; (2) it simulates real-world scenarios where text is partially visible, damaged, or obscured; and (3) it offers clear evaluation metrics with unique ground-truth answers, unlike subjective benchmarks that suffer from evaluation ambiguity. Figure 2 demonstrates how humans solve a hard-mode VCR task, highlighting the cognitive processes that advanced AI systems should aim to replicate.

## 3 DATASET CREATION

The VCR task requires aligning visual images ($VI$) with text embedded in images ($TEI$). Therefore, the dataset creation process relies on a set of highly correlated image-text pairs. We utilize the primary

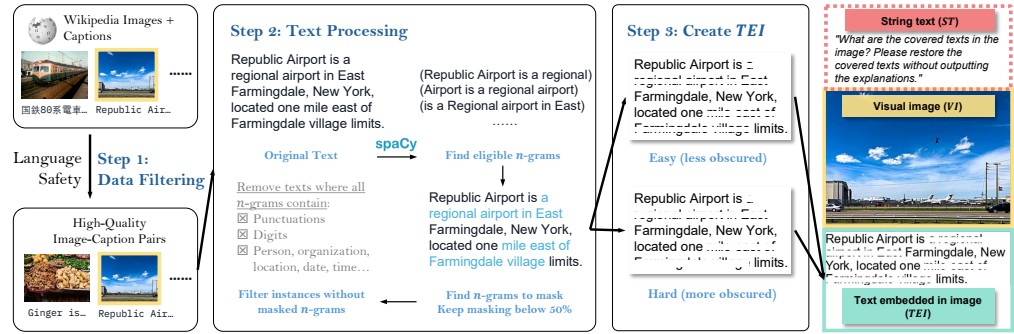

Figure 3: Illustration of the dataset creation pipeline for **VCR-WIKI**. visual image ($VI$), text embedded in image ($TEI$) and string text ($ST$) in an example of the English Hard configuration of **VCR-WIKI**. The solid line-enclosed contents ($VI$ and $TEI$) are part of the image, whereas the dotted line-enclosed content ($ST$) is given separately from the image.

images and their corresponding captions from Wikipedia as the data source[1] to create **VCR-WIKI**, a Wikipedia-based VCR dataset. The pipeline for creating **VCR-WIKI** is shown in Figure 3. Before constructing the dataset, we first filter out instances with sensitive content, including NSFW and crime-related terms, to mitigate AI risk and biases.

The **VCR-WIKI** dataset is formatted as a VQA task, where each instance includes an image, a question, and a ground-truth answer. The images are synthesized from text-image pairs by stacking the image ($VI$) with its corresponding text description ($TEI$) vertically, mimicking the format of a captioned image. This stacked image is referred to as a stacked $VI$ +$TEI$ image. Each $VI$ +$TEI$ image is resized to a width of 300 pixels. To avoid excessive image height, we truncate $TEI$ to a maximum of five lines. We filter the dataset to exclude instances with $VI$ +$TEI$ images exceeding 900 pixels in height to avoid drastic resolution changes during data pre-processing.

We use spaCy to randomly select 5-grams in the caption for masking. To ensure the restoration process is doable by a human without too much domain knowledge, the 5-grams do not contain numbers, person names, religious or political groups, facilities, organizations, locations, dates, and times labeled by spaCy. The total masked token does not exceed 50% of the tokens in the caption. We pick 5-grams for masking as it balances linguistic complexity and task feasibility, capturing meaningful grammatical structures while avoiding dataset reduction or overly simplified tasks observed with longer or shorter spans. We exclude instances that do not have any maskable 5-grams. The selected 5-grams are partially obscured by a white rectangle that reveals only the upper and lower parts of the text, with the proportion of coverage varying according to task difficulty. Furthermore, to assess the impact of $VI$ on model performance, we create an ablation for each image, maintaining the resolution of the $VI$ +$TEI$ image, but retaining only the $TEI$ part in the center of the image.

The VCR task involves a predefined question that prompts the model to produce the obscured text in the image. The ground-truth answer corresponds to the caption displayed in the uncovered portion of the stacked image. Due to the extensive availability of VLMs and a significant user base in both English and Chinese, we have chosen to develop the dataset in these two languages. For each language, we meticulously select the height of the masking rectangle to create two task variants: (1) an *easy* version, where the task is easy for native speakers but open-source OCR models almost always fail, and (2) a *hard* version, where the revealed part consists of only one to two pixels for the majority of letters or characters, yet the restoration task remains feasible for native speakers.

To avoid test data leakage, we create a **hidden test (VCR-HIDDEN)** for the VCR task which will not be publicized. While it follows our dataset's general construction principles regarding masked span length and the span selection criteria, it differs in five key aspects: (1) Image-text pairs no longer come from the Wikipedia; (2) $TEI$ is randomly positioned either above or below $VI$; (3) Masked regions are randomly selected from both "easy" and "hard" settings; (4) Multiple popular fonts are used randomly for the $TEI$ component; and (5) The $VI$ +$TEI$ image width varies uniformly between 600 - 1000 pixels. We include results on **VCR-HIDDEN** in Table 1 and Appendix C.

---

[1] Datasource: `https://huggingface.co/datasets/wikimedia/wit_base`.

## 3.1 DATASET FORMAT AND STATISTICS

The **VCR-WIKI** dataset comprises four configurations: English Easy, English Hard, Chinese Easy and Chinese Hard. Each configuration can be further divided into training, validation, and test splits. The validation and test splits contain 5,000 entities each. The training set for English configurations and Chinese configurations contains 2,095,733 and 336,448 instances, respectively, which can be used for model continuous pretraining. To avoid test data leakage, we also newly include hidden test sets (**VCR-HIDDEN**) for both languages as described in Section 3, which contains 100 entities for each language. We include detailed statistics of the dataset in Table 4 in Appendix A.

## 4 EXPERIMENTS

In this section, we report the experimental results of existing state-of-the-art vision-language models on both **VCR-WIKI** and the **VCR-HIDDEN** hidden test.

## 4.1 MODELS

**Closed-source and Open-source Models.** In this paper, we report results for several state-of-the-art closed-source and open-source models from the OpenVLM Leaderboard[2], as well as selected state-of-the-art models as of February 2025 for **VCR-WIKI**. For the **VCR-HIDDEN** hidden test, we report results of state-of-the-art closed-source and open-source models available as of February 2025. See Appendix B for complete model specifications. We commit to evaluating emerging state-of-the-art VLMs to reflect cutting-edge advancements. Results for later models will be actively updated in the leaderboard hosted on `https://github.com/tianyu-z/VCR`.

**Fine-tuned Models.** To test whether VLMs can learn to conduct VCR via fine-tuning, we select three models from the open-sourced models: CogVLM2-Llama3-19B-Chat, MiniCPM-Llama3-V2.5, and Qwen2-VL-7B-Instruct, and fine-tune them on a subset of VCR's training set.

More specifically, we fine-tune CogVLM2-Llama3-19B-Chat, MiniCPM-Llama3-V2.5, and Qwen2-VL-7B-Instruct in the English Hard configuration, and CogVLM2-Llama3-19B-Chinese-Chat, MiniCPM-Llama3-V2.5, and Qwen2-VL-7B-Instruct on the Chinese Hard configuration. The models are finetuned using LoRA (Hu et al., 2022) with $r = 8$ and $\alpha = 32$. We adopt the schedule-free AdamW optimizer (Defazio et al., 2024) with a learning rate $2e-4$. The effective batch size is 64. Each model is trained on the first 16,000 examples of the training set for 1 epoch. All fine-tuning experiments are performed on a single node with 4 NVIDIA L40S 48G GPUs.

## 4.2 METRICS

We measure the quality of the model's restoration of each masked $n$-gram (where $n = 5$ in our setting, as specified in Section 3). Due to the variability of different models' outputs, for each masked $n$-gram $m \in \mathbb{V}_e^n$, where $\mathbb{V}_e$ is the vocabulary of the evaluation tokenizer[3], we extract the most similar $n$-gram $\hat{m} \in \mathbb{V}_e^n$ with the least edit distance in the model's generation.

We report the two metrics below in our experiment section to measure the restoration quality: **Exact Match** ($EM$) $\equiv EM(m, \hat{m}) = \mathbb{I}(m = \hat{m})$, which measures whether the restored $n$-gram $\hat{m}$ totally matches the ground-truth $m$; and **Jaccard Index** ($J$) $\equiv \frac{|S(m) \cap S(\hat{m})|}{|S(m) \cup S(\hat{m})|}$, which measures the similarity of $\hat{m}$ and $m$ as bag-of-words.

## 4.3 RELATIONSHIP TO OTHER BENCHMARKS.

We evaluated 38 Vision-Language Models (VLMs) across 23 different benchmarks, using the VLM performance scores as features of each benchmark to compute a correlation matrix. Based on this

---

[2]We selected the highest-performing open-source models with fewer than 40 billion parameters from `https://huggingface.co/spaces/opencompass/open_vlm_leaderboard` as of May 2024 and their later versions for **VCR-WIKI**.

[3]We utilize spaCy's `en_core_web_sm`'s and `zh_core_web_sm`'s tokenizer for English and Chinese evaluation, respectively.

matrix, in Figure 4, we applied K-Means clustering and visualized the results in 2D by plotting the first two principal components derived from the correlation matrix rows for each benchmark. Additionally,

we provide a heatmap of the correlation matrix in Appendix E.

$VCR_{ZH, EASY}$ and $VCR_{ZH, HARD}$ were excluded from these processes due to the limited availability of VLMs that support Chinese. According to Figure 4, $VCR_{EN, EASY}$ shows a tentative similarity to ChartQA and TextVQA, as all three benchmarks evaluate the ability to extract and reason about text from natural images and documents. However, $VCR_{EN, EASY}$ does not exhibit significant similarity to the other benchmarks. Meanwhile, $VCR_{EN, HARD}$ stands apart from all other benchmarks. We attribute this to the fact that $VCR_{EN, HARD}$ emphasizes caption recovery with minimal pixel-level information, a skill not tested by any of the other benchmarks. Therefore, we assert that the VCR series benchmarks assess unique aspects of VLMs that are not covered by any of the other benchmarks in our evaluation.

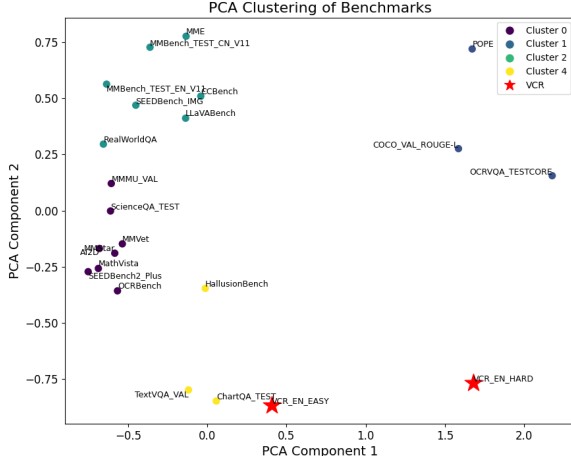

Figure 4: Projection of benchmarks onto the first two principal components derived from the correlation matrix of VLM performance scores. Each point represents a benchmark, and proximity indicates higher similarity based on model performance correlation.

## 4.4 EXPERIMENTAL RESULTS

For **VCR-HIDDEN**, Table 1 presents state-of-the-art VLMs' performance as of February 2025, with more comprehensive results for more models in Table 6 (English) and Table 7 (Chinese). In addition, Table 2 shows exact match scores and Jaccard indices on **VCR-WIKI**. Figure 5 compares our fine-tuned models against base models across all VCR settings. For faster benchmarking, Tables 8 and 9 in the Appendix provide results on smaller **VCR-WIKI** test sets containing 100 or 500 samples. In this section, we analyze models' performance as of our paper submission date (October 2024) on the VCR task, highlighting key insights through comparative evaluations.

**VCR Remains a Challenging Task for SOTA VLMs.** Despite high-performing models like Qwen2-VL excelling in $VCR_{EN, EASY}$ and $VCR_{EN, HARD}$ settings, most recent models struggle significantly, especially under harder settings where metrics approach zero. This highlights not only the inherent difficulty of the VCR task but also that subpar performance on **VCR-WIKI** stems from *a lack of reasoning capabilities or sufficient text-image alignment rather than unfamiliarity with the underlying text*, as many VLMs are pretrained on similar data. These results emphasize the need for advancements in VLM designs to achieve robust performance across all settings. Besides, to avoid over-fitting on the train set we released, we also include results from our hidden test set **VCR-HIDDEN**, which is not publically accessible. For most recent models, we observe a performance decline on **VCR-HIDDEN** compared with **VCR-WIKI**. We will keep updating the both the public **VCR-WIKI** and private **VCR-HIDDEN** leaderboards in our Github repository.

**Enhanced OCR Capabilities Do Not Necessarily Translate to Improved VCR Performance.** Our analysis reveals that models proficient in OCR, such as InternLM-XComposer2-VL, and those excelling in image document understanding, like DocOwl 1.5 and Monkey, demonstrate subpar performance across most VCR settings. This discrepancy suggests that while these models can accurately recognize text within images, they lack the advanced reasoning capabilities required to effectively interpret and utilize this information within the context of the VCR task. This finding also highlights a fundamental distinction between OCR tasks and the more complex VCR task.

**Language-Specific Performance: Need for Enhanced Multilingual Capabilities.** A significant performance degradation is observed when models are evaluated on Chinese configurations, despite assertions of basic English-Chinese bilingual capabilities. This decline is particularly surprising given the logographic nature of Chinese characters, which theoretically offer higher recognizability

Table 1: Performance of some SOTA-level VLMs (as of February 2025) on the **VCR-HIDDEN** in English and Chinese. We label the best result of each setting and metric with **bold fonts**. A superscript of * marks that the model was released after the initial public release of the **VCR-WIKI** dataset (June 10, 2024). Subscripts show bootstrapped standard deviation. Refer to Table 6 for the complete results in English and Table 7 for the complete results in Chinese.

| Language | Open/closed source | Model name | Model size | Exact match (%) ↑ | | | Jaccard index (%) ↑ | | |
|---|---|---|---|---|---|---|---|---|---|
| | | | | $VI + TEI$ | $TEI$ | $\Delta$ | $VI + TEI$ | $TEI$ | $\Delta$ |
| English | Closed | Claude 3.7 Sonnet* | - | $74.00_{2.99}$ | $58.50_{3.38}$ | 15.50 | $81.53_{2.34}$ | $61.47_{3.26}$ | 20.06 |
| | | Gemini 2.0 Flash* | - | $48.50_{3.55}$ | $42.00_{3.55}$ | 6.50 | $66.35_{2.76}$ | $61.16_{2.76}$ | 5.20 |
| | | GPT-4o | - | $41.50_{3.43}$ | $45.50_{3.55}$ | -4.00 | $48.09_{3.39}$ | $51.51_{3.30}$ | -3.43 |
| | | o1* | - | $26.50_{3.17}$ | $18.50_{2.64}$ | 8.00 | $31.17_{3.11}$ | $22.06_{2.79}$ | 9.11 |
| | | Grok 2 Vision* | - | $16.50_{2.43}$ | $26.00_{3.15}$ | -9.50 | $36.05_{2.47}$ | $43.83_{2.79}$ | -7.78 |
| | Open | CogVLM2 | 19B | $53.00_{3.72}$ | $51.00_{3.56}$ | 2.00 | $64.97_{2.76}$ | $61.01_{2.98}$ | 3.97 |
| | | DeepSeek-VL2* | 28B | $23.50_{2.93}$ | $32.00_{3.31}$ | -8.50 | $31.75_{2.94}$ | $45.50_{2.90}$ | -13.75 |
| | | InternVL2.5* | 78B | $83.50_{2.70}$ | $\mathbf{79.50_{2.83}}$ | 4.00 | $91.47_{1.39}$ | $\mathbf{87.60_{1.88}}$ | 3.87 |
| | | Llama-3.2-Vision* | 90B | $51.00_{3.52}$ | $33.50_{3.20}$ | 17.50 | $63.21_{2.88}$ | $50.55_{2.86}$ | 12.66 |
| | | QvQ-Preview* | 72B | $66.00_{3.40}$ | $62.50_{3.25}$ | 3.50 | $75.76_{2.66}$ | $72.57_{2.73}$ | 3.19 |
| | | Qwen2.5-VL* | 72B | $\mathbf{86.50_{2.43}}$ | $\mathbf{79.50_{2.88}}$ | 7.00 | $\mathbf{92.28_{1.52}}$ | $86.43_{2.01}$ | 5.86 |
| | | Ovis2* | 34B | $62.00_{3.52}$ | $60.50_{3.43}$ | 1.50 | $73.30_{2.70}$ | $71.34_{2.85}$ | 1.97 |
| Chinese | Closed | Claude 3.7 Sonnet* | - | $3.00_{1.16}$ | $0.50_{0.50}$ | 2.50 | $9.93_{1.44}$ | $0.92_{0.52}$ | 9.02 |
| | | Gemini 2.0 Flash* | - | $1.00_{0.70}$ | $1.00_{0.69}$ | 0.00 | $12.36_{1.11}$ | $11.77_{1.09}$ | 0.59 |
| | | GPT-4o | - | $1.50_{0.89}$ | $0.50_{0.50}$ | 1.00 | $3.78_{1.01}$ | $2.22_{0.66}$ | 1.56 |
| | Open | CogVLM2-Chinese | 19B | $3.00_{1.17}$ | $1.50_{0.82}$ | 1.50 | $16.05_{1.47}$ | $13.15_{1.21}$ | 2.90 |
| | | InternVL2.5* | 78B | $21.00_{2.91}$ | $11.50_{2.30}$ | 9.50 | $46.23_{2.52}$ | $34.80_{2.12}$ | 11.43 |
| | | QvQ-Preview* | 72B | $18.50_{2.75}$ | $23.50_{3.01}$ | -5.00 | $24.71_{2.73}$ | $34.85_{2.90}$ | -10.13 |
| | | Qwen2.5-VL* | 72B | $\mathbf{33.50_{3.35}}$ | $\mathbf{28.00_{3.21}}$ | 5.50 | $\mathbf{52.65_{2.78}}$ | $\mathbf{46.83_{2.72}}$ | 5.82 |
| | | Ovis2* | 34B | $2.00_{1.00}$ | $1.00_{0.73}$ | 1.00 | $14.12_{1.39}$ | $13.49_{1.28}$ | 0.62 |

compared to alphabetic scripts (Wu et al., 2024a; Zhao et al., 2022). These results indicate a critical need for targeted improvements in multilingual support to ensure consistent performance across different languages.

**Model Size Does Not Guarantee Superior Performance.** Comparative analysis between Llama-3.2-11B and Llama-3.2-90B models reveals that both exhibit similar performance levels on the $VCR_{EN, EASY}$ and $VCR_{EN, HARD}$ settings. This observation suggests that merely increasing model size does not inherently enhance VCR performance. Instead, advancements in the cognitive abilities of models, achieved through improved training strategies, reasoning frameworks, or architectural innovations, are essential for meaningful performance gains in VCR tasks.

**Model Resolution Is Not Directly Correlated with Performance Enhancement.** InternLM-XComposer2-VL-4K, despite its higher resolution, demonstrates significantly lower performance on the $VCR_{EN, EASY}$ setting compared to its lower-resolution counterparts. This decline may result from more aggressive image partitioning strategies that disrupt the spatial continuity of text or from more intensive pixel or token compression techniques that lead to the loss of crucial local details. Both factors are critical for the accurate interpretation required in VCR tasks.

**Inclusion of $VI$ Input Images Negatively Impacts Performance.** The addition of $VI$ generally results in negative performance changes ($\Delta < 0$), indicating that the image information is not being effectively leveraged by the models. This negative impact may stem from the importance of key information locations, which could be compromised by image partitioning strategies that fail to preserve spatial relationships essential for accurate reasoning.

**Model Design Influences Performance Gains from VCR-WIKI Finetuning.** As shown in Figure 5, finetuning on the **VCR-WIKI** dataset yields varying performance improvements across different model designs. Specifically: 1) **CogVLM2** demonstrates substantial performance enhancements across all four settings after finetuning, indicating that its overall design may be well-aligned with the image-text reasoning demands of VCR, though further empirical validation is needed to substantiate this hypothesis. 2) **MiniCPM-V2.5** shows only marginal performance increases from an already low baseline, particularly in the $VCR_{ZH, EASY}$ and $VCR_{ZH, HARD}$ settings. This limited improvement indicates potential design limitations that hinder its ability to effectively address the complexities of the VCR task. 3) **Qwen2-VL-7B** maintains relatively high performance both before and after finetuning, implying that the model is sufficiently pre-trained on relevant tasks to perform well on the VCR task without extensive additional training.

We hope that through controlled variables, the **VCR-WIKI** dataset delivers *fully comparable* results across languages, difficulty levels, image inclusion, and fine-tuning stages. Each comparison is intended to guide specific and targeted improvements in VLM development.

Table 2: Performance of vision language models on **VCR-WIKI** in English and Chinese, for easy and hard modes. We label the best result of each setting and metric with **bold fonts**, the best open-source model with underline, and the best open-source model released before **VCR-WIKI**'s initial public release (June 10, 2024) with *italic font*. A superscript of * marks that the model was released after the initial public release of **VCR-WIKI**. Subscripts show bootstrapped standard deviation. For more latest models' results, please visit our GitHub repository.

| Language | Mode | Open/closed source | Model name | Model size | Exact match (%) ↑ | | | Jaccard index (%) ↑ | | |
|---|---|---|---|---|---|---|---|---|---|---|
| | | | | | $VI+TEI$ | $TEI$ | $\Delta$ | $VI+TEI$ | $TEI$ | $\Delta$ |
| English | Easy | Closed | Claude 3 Opus | - | $62.0_{0.13}$ | $77.0_{0.5}$ | -15 | $77.67_{0.32}$ | $88.41_{0.39}$ | -10.74 |
| | | | Claude 3.5 Sonnet | - | $63.85_{1.71}$ | $72.8_{1.56}$ | -8.94 | $74.65_{1.33}$ | $83.48_{1.14}$ | -8.83 |
| | | | Gemini 1.5 Pro | - | $62.73_{1.66}$ | $82.98_{1.3}$ | -20.25 | $77.71_{1.21}$ | $91.56_{0.76}$ | -13.85 |
| | | | GPT-4 Turbo | - | $78.74_{0.13}$ | $81.94_{0.25}$ | -3.2 | $88.54_{0.24}$ | $92.18_{0.3}$ | -3.65 |
| | | | GPT-4o | - | $91.55_{0.29}$ | $94.56_{0.13}$ | -3.01 | $96.44_{0.11}$ | $\mathbf{97.76_{0.06}}$ | -1.32 |
| | | | GPT-4V | - | $52.04_{0.24}$ | $37.86_{0.22}$ | 14.17 | $65.36_{0.39}$ | $54.13_{0.41}$ | 11.23 |
| | | | Qwen-VL-Max | - | $76.8_{0.5}$ | $85.53_{0.19}$ | -8.74 | $85.71_{0.28}$ | $91.45_{0.29}$ | -5.74 |
| | | | Reka Core | - | $66.46_{1.64}$ | $78.51_{1.42}$ | -12.05 | $84.23_{0.86}$ | $90.45_{0.7}$ | -6.22 |
| | | Open | Cambrian-1* | 34B | $79.69_{0.43}$ | $81.28_{0.43}$ | -1.59 | $89.27_{0.28}$ | $92.54_{0.19}$ | -3.27 |
| | | | CogVLM2 | 19B | $83.25_{0.07}$ | $78.29_{0.04}$ | 4.96 | $89.75_{0.1}$ | $88.07_{0.08}$ | 1.68 |
| | | | DeepSeek-VL | 7B | $38.01_{0.12}$ | $45.94_{0.1}$ | -7.93 | $60.02_{0.15}$ | $64.72_{0.04}$ | -4.7 |
| | | | DeepSeek-VL2* | 28B | $4.07_{0.21}$ | $5.21_{0.24}$ | -1.14 | $5.05_{0.23}$ | $5.97_{0.25}$ | -0.93 |
| | | | DocOwl-1.5-Omni | 8B | $0.84_{0.01}$ | $1.55_{0.02}$ | -0.71 | $13.34_{0.03}$ | $14.62_{0.04}$ | -1.28 |
| | | | Idefics3* | 8B | $25.99_{0.48}$ | $31.43_{0.51}$ | -5.44 | $47.22_{0.42}$ | $54.00_{0.39}$ | -6.78 |
| | | | InternLM-XComposer2-VL | 7B | $46.64_{0.1}$ | $46.4_{0.11}$ | 0.24 | $70.99_{0.1}$ | $72.14_{0.07}$ | -1.14 |
| | | | InternLM-XComposer2-VL-4K | 7B | $5.32_{0.24}$ | $3.71_{0.21}$ | 1.60 | $22.14_{0.28}$ | $18.78_{0.25}$ | 3.37 |
| | | | InternLM-XComposer2.5-VL* | 7B | $41.35_{0.55}$ | $25.37_{0.51}$ | 15.97 | $63.04_{0.42}$ | $49.95_{0.41}$ | 13.09 |
| | Easy | | InternVL-V2* | 40B | $84.67_{0.40}$ | $87.71_{0.37}$ | -3.04 | $92.64_{0.22}$ | $95.10_{0.16}$ | -2.47 |
| | | | InternVL-V2* | 76B | $83.20_{0.43}$ | $90.25_{0.33}$ | -7.05 | $91.26_{0.24}$ | $96.10_{0.14}$ | -4.83 |
| | | | Llama-3.2* | 11B | $79.85_{0.45}$ | $67.53_{0.53}$ | 12.32 | $90.58_{0.22}$ | $81.11_{0.33}$ | 9.47 |
| | | | Llama-3.2* | 90B | $80.54_{0.43}$ | $71.05_{0.51}$ | 9.48 | $89.81_{0.26}$ | $84.22_{0.30}$ | 5.59 |
| | | | MiniCPM-V2.5 | 8B | $31.81_{0.08}$ | $40.05_{0.09}$ | -8.25 | $53.24_{0.1}$ | $63.2_{0.1}$ | -9.96 |
| | | | Monkey | 7B | $50.66_{0.1}$ | $56.2_{0.08}$ | -5.54 | $67.6_{0.09}$ | $72.82_{0.08}$ | -5.22 |
| | | | Pixtral* | 12B | $18.41_{0.42}$ | $11.60_{0.36}$ | 6.81 | $41.25_{0.37}$ | $31.60_{0.33}$ | 9.65 |
| | | | Ovis2* | 34B | $74.13_{0.48}$ | $73.64_{0.49}$ | 0.49 | $83.13_{0.35}$ | $86.79_{0.28}$ | -3.66 |
| | | | Qwen-VL | 7B | $49.71_{0.17}$ | $52.15_{0.15}$ | -2.44 | $69.94_{0.07}$ | $72.28_{0.08}$ | -2.34 |
| | | | Qwen2-VL* | 7B | $89.70_{0.34}$ | $93.44_{0.26}$ | -3.74 | $93.84_{0.24}$ | $97.47_{0.12}$ | -3.62 |
| | | | Qwen2-VL* | 72B | $91.30_{0.32}$ | $94.64_{0.26}$ | -3.34 | $94.04_{0.23}$ | $97.42_{0.14}$ | -3.38 |
| | | | Qwen2.5-VL* | 7B | $\mathbf{94.81_{0.25}}$ | $93.79_{0.27}$ | 1.01 | $\mathbf{98.09_{0.10}}$ | $97.24_{0.13}$ | 0.84 |
| | | | Qwen2.5-VL* | 72B | $91.87_{0.29}$ | $87.65_{0.37}$ | 4.23 | $95.48_{0.18}$ | $90.75_{0.30}$ | 4.73 |
| | | | Yi-VL | 34B | $0.82_{0.03}$ | $1.61_{0.04}$ | -0.79 | $5.59_{0.04}$ | $7.72_{0.03}$ | -2.13 |
| | Hard | Closed | Claude 3 Opus | - | $37.8_{0.28}$ | $50.0_{0.33}$ | -12.2 | $57.68_{0.8}$ | $70.16_{0.64}$ | -12.48 |
| | | | Claude 3.5 Sonnet | - | $41.74_{1.69}$ | $44.72_{1.78}$ | -2.98 | $56.15_{1.46}$ | $58.54_{1.6}$ | -2.4 |
| | | | Gemini 1.5 Pro | - | $28.07_{1.58}$ | $38.76_{1.68}$ | -10.68 | $51.9_{1.22}$ | $59.62_{1.27}$ | -7.72 |
| | | | GPT-4 Turbo | - | $45.15_{0.28}$ | $48.64_{0.57}$ | -3.5 | $65.72_{0.25}$ | $67.86_{0.2}$ | -2.14 |
| | | | GPT-4o | - | $73.2_{0.16}$ | $\mathbf{82.43_{0.17}}$ | -9.22 | $86.17_{0.21}$ | $\mathbf{92.01_{0.2}}$ | -5.84 |
| | | | GPT-4V | - | $25.83_{0.44}$ | $14.95_{0.3}$ | 10.87 | $44.63_{0.48}$ | $30.08_{0.67}$ | 14.56 |
| | | | Qwen-VL-Max | - | $41.65_{0.32}$ | $52.72_{0.2}$ | -11.07 | $61.18_{0.35}$ | $70.19_{0.37}$ | -9.01 |
| | | | Reka Core | - | $6.71_{0.89}$ | $11.18_{1.15}$ | -4.47 | $25.84_{0.95}$ | $35.83_{1.05}$ | -9.99 |
| | | Open | Cambrian-1* | 34B | $27.20_{0.48}$ | $29.68_{0.50}$ | -2.48 | $50.04_{0.40}$ | $55.66_{0.39}$ | -5.62 |
| | | | CogVLM2 | 19B | $37.98_{0.18}$ | $17.68_{0.06}$ | 20.3 | $59.99_{0.05}$ | $39.69_{0.03}$ | 20.3 |
| | | | DeepSeek-VL | 7B | $1.0_{0.02}$ | $1.75_{0.03}$ | -0.75 | $15.9_{0.08}$ | $17.2_{0.04}$ | -1.3 |
| | | | DeepSeek-VL2* | 28B | $25.06_{0.47}$ | $32.39_{0.51}$ | -7.33 | $45.55_{0.43}$ | $54.18_{0.41}$ | -8.63 |
| | | | DocOwl-1.5-Omni | 8B | $0.04_{0.0}$ | $0.02_{0.0}$ | 0.01 | $7.76_{0.01}$ | $7.74_{0.02}$ | 0.03 |
| | | | Idefics3* | 8B | $0.60_{0.08}$ | $0.37_{0.07}$ | 0.23 | $10.37_{0.15}$ | $9.59_{0.13}$ | 0.79 |
| | | | InternLM-XComposer2-VL | 7B | $0.7_{0.01}$ | $0.92_{0.01}$ | -0.22 | $12.51_{0.02}$ | $13.23_{0.02}$ | -0.72 |
| | | | InternLM-XComposer2-VL-4K | 7B | $0.21_{0.05}$ | $0.18_{0.05}$ | 0.02 | $9.52_{0.12}$ | $9.52_{0.11}$ | -0.00 |
| | | | InternLM-XComposer2.5-VL* | 7B | $0.93_{0.11}$ | $1.11_{0.11}$ | -0.18 | $13.82_{0.16}$ | $14.72_{0.18}$ | -0.89 |
| | Hard | | InternVL-V2* | 40B | $13.10_{0.37}$ | $19.16_{0.44}$ | -6.06 | $33.64_{0.36}$ | $41.35_{0.39}$ | -7.71 |
| | | | InternVL-V2* | 76B | $20.58_{0.39}$ | $20.29_{0.39}$ | 0.29 | $44.59_{0.34}$ | $42.86_{0.34}$ | 1.73 |
| | | | Llama-3.2* | 11B | $14.09_{0.40}$ | $6.92_{0.27}$ | 7.17 | $35.26_{0.36}$ | $26.35_{0.29}$ | 8.90 |
| | | | Llama-3.2* | 90B | $14.91_{0.40}$ | $13.06_{0.37}$ | 1.85 | $35.44_{0.35}$ | $34.44_{0.35}$ | 1.00 |
| | | | MiniCPM-V2.5 | 8B | $1.41_{0.03}$ | $1.96_{0.02}$ | -0.55 | $11.94_{0.02}$ | $13.37_{0.04}$ | -1.43 |
| | | | Monkey | 7B | $1.96_{0.04}$ | $2.43_{0.03}$ | -0.48 | $14.02_{0.03}$ | $14.11_{0.03}$ | -0.09 |
| | | | Pixtral* | 12B | $0.44_{0.08}$ | $0.64_{0.09}$ | -0.20 | $10.99_{0.13}$ | $11.45_{0.15}$ | -0.46 |
| | | | Ovis2* | 34B | $77.43_{0.46}$ | $69.31_{0.49}$ | 8.13 | $87.92_{0.29}$ | $87.02_{0.25}$ | 0.89 |
| | | | Qwen-VL | 7B | $2.0_{0.03}$ | $2.32_{0.03}$ | -0.32 | $15.04_{0.05}$ | $14.27_{0.05}$ | 0.77 |
| | | | Qwen2-VL* | 7B | $74.32_{0.47}$ | $75.20_{0.49}$ | -0.88 | $85.47_{0.30}$ | $87.63_{0.27}$ | -2.15 |
| | | | Qwen2-VL* | 72B | $68.87_{0.52}$ | $71.70_{0.49}$ | -1.83 | $82.78_{0.33}$ | $85.58_{0.28}$ | -2.80 |
| | | | Qwen2.5-VL* | 7B | $\underline{80.47_{0.45}}$ | $73.83_{0.48}$ | 6.65 | $\underline{91.65_{0.20}}$ | $87.85_{0.25}$ | 3.80 |
| | | | Qwen2.5-VL* | 72B | $79.79_{0.45}$ | $67.31_{0.54}$ | 12.48 | $87.91_{0.29}$ | $77.12_{0.42}$ | 10.78 |
| | | | Yi-VL | 34B | $0.07_{0.0}$ | $0.05_{0.0}$ | 0.02 | $4.31_{0.02}$ | $5.89_{0.02}$ | -1.58 |
| Chinese | Easy | Closed | Claude 3 Opus | - | $0.9_{0.3}$ | $1.0_{0.31}$ | -0.1 | $11.5_{0.49}$ | $10.0_{0.49}$ | 1.49 |
| | | | Claude 3.5 Sonnet | - | $1.0_{0.31}$ | $0.8_{0.28}$ | 0.2 | $7.54_{0.54}$ | $7.5_{0.51}$ | 0.03 |
| | | | Gemini 1.5 Pro | - | $1.1_{0.32}$ | $0.5_{0.22}$ | 0.6 | $11.1_{0.56}$ | $11.47_{0.48}$ | -0.37 |
| | | | GPT-4o | - | $14.87_{1.14}$ | $22.46_{1.35}$ | -7.58 | $39.05_{0.99}$ | $48.24_{1.09}$ | -9.19 |
| | | | GPT-4 Turbo | - | $0.2_{0.14}$ | $0.1_{0.1}$ | 0.1 | $8.42_{0.36}$ | $6.97_{0.29}$ | 1.45 |
| | | | Qwen-VL-Max | - | $6.34_{0.08}$ | $9.92_{0.09}$ | -3.58 | $13.45_{0.41}$ | $22.86_{0.46}$ | -9.42 |
| | | | Reka Core | - | $0.0_{0.0}$ | $0.0_{0.0}$ | 0 | $3.43_{0.26}$ | $3.15_{0.2}$ | 0.28 |
| | | Open | CogVLM2-Chinese | 19B | $33.24_{0.04}$ | $30.7_{0.07}$ | 2.54 | $57.57_{0.06}$ | $53.66_{0.04}$ | 3.91 |
| | | | DeepSeek-VL | 7B | $0.0_{0.0}$ | $0.0_{0.0}$ | 0 | $4.08_{0.01}$ | $6.84_{0.01}$ | -2.76 |
| | | | DeepSeek-VL2* | 28B | $3.81_{0.18}$ | $2.95_{0.17}$ | 0.86 | $10.32_{0.20}$ | $10.40_{0.20}$ | -0.07 |
| | | | DocOwl-1.5-Omni | 8B | $0.0_{0.0}$ | $0.0_{0.0}$ | 0 | $1.14_{0.01}$ | $3.38_{0.01}$ | -2.23 |
| | | | InternLM-XComposer2-VL | 7B | $0.27_{0.01}$ | $0.23_{0.01}$ | 0.04 | $12.32_{0.02}$ | $12.28_{0.03}$ | 0.04 |
| | | | InternLM-XComposer2-VL-4K | 7B | $0.46_{0.07}$ | $0.46_{0.07}$ | 0.00 | $12.31_{0.14}$ | $13.37_{0.14}$ | -1.05 |
| | | | InternLM-XComposer2.5-VL* | 7B | $0.46_{0.07}$ | $0.58_{0.08}$ | -0.12 | $12.97_{0.16}$ | $14.99_{0.17}$ | -2.01 |
| | Easy | | InternVL-V2* | 40B | $22.09_{0.41}$ | $17.26_{0.39}$ | 4.84 | $47.62_{0.34}$ | $37.93_{0.35}$ | 9.69 |
| | | | InternVL-V2* | 76B | $18.45_{0.44}$ | $21.09_{0.44}$ | -2.64 | $41.16_{0.37}$ | $44.48_{0.38}$ | -3.32 |
| | | | MiniCPM-V2.5 | 8B | $4.1_{0.02}$ | $5.05_{0.08}$ | -0.95 | $18.03_{0.07}$ | $22.94_{0.04}$ | -4.9 |
| | | | Monkey | 7B | $0.62_{0.01}$ | $1.44_{0.01}$ | -0.82 | $8.34_{0.06}$ | $10.95_{0.03}$ | -2.61 |
| | | | Ovis2* | 34B | $21.72_{0.40}$ | $16.68_{0.36}$ | 5.04 | $39.94_{0.37}$ | $36.43_{0.34}$ | 3.51 |
| | | | Qwen-VL | 7B | $0.04_{0.01}$ | $0.0_{0.0}$ | 0.04 | $1.5_{0.01}$ | $0.34_{0.01}$ | 1.15 |
| | | | Qwen2-VL* | 7B | $59.94_{0.49}$ | $67.48_{0.47}$ | -7.54 | $76.95_{0.32}$ | $82.63_{0.28}$ | -5.67 |
| | | | Qwen2-VL* | 72B | $65.38_{0.46}$ | $74.08_{0.44}$ | -8.70 | $81.14_{0.28}$ | $\mathbf{86.78_{0.25}}$ | -5.64 |
| | | | Qwen2.5-VL* | 7B | $72.38_{0.46}$ | $46.08_{0.49}$ | 26.30 | $84.66_{0.28}$ | $57.03_{0.42}$ | 27.63 |
| | | | Qwen2.5-VL* | 72B | $\underline{75.82_{0.41}}$ | $\underline{72.12_{0.43}}$ | 3.70 | $\mathbf{\underline{86.93_{0.25}}}$ | $83.40_{0.28}$ | 3.54 |
| | | | Yi-VL | 34B | $0.0_{0.0}$ | $0.0_{0.0}$ | 0 | $4.44_{0.01}$ | $1.8_{0.01}$ | 2.64 |
| | Hard | Closed | Claude 3 Opus | - | $0.3_{0.18}$ | $0.1_{0.1}$ | 0.2 | $9.22_{0.38}$ | $8.09_{0.33}$ | 1.13 |
| | | | Claude 3.5 Sonnet | - | $0.2_{0.15}$ | $0.0_{0.0}$ | 0.2 | $4.0_{0.33}$ | $2.37_{0.23}$ | 1.63 |
| | | | Gemini 1.5 Pro | - | $0.7_{0.26}$ | $0.5_{0.23}$ | 0.2 | $11.82_{0.51}$ | $11.75_{0.44}$ | 0.07 |
| | | | GPT-4o | - | $2.2_{0.47}$ | $1.8_{0.4}$ | 0.4 | $22.72_{0.67}$ | $22.89_{0.65}$ | -0.17 |
| | | | GPT-4 Turbo | - | $0.0_{0.0}$ | $0.2_{0.13}$ | -0.2 | $8.58_{0.3}$ | $6.87_{0.28}$ | 1.72 |
| | | | Qwen-VL-Max | - | $0.89_{0.06}$ | $1.38_{0.1}$ | -0.49 | $5.4_{0.19}$ | $12.29_{0.18}$ | -6.89 |
| | | | Reka Core | - | $0.0_{0.0}$ | $0.0_{0.0}$ | 0 | $3.35_{0.23}$ | $2.97_{0.2}$ | 0.38 |
| | | Open | CogVLM2-Chinese | 19B | $1.34_{0.03}$ | $2.67_{0.02}$ | -1.32 | $17.35_{0.03}$ | $19.51_{0.03}$ | -2.16 |
| | | | DeepSeek-VL | 7B | $0.0_{0.0}$ | $0.0_{0.0}$ | 0 | $5.11_{0.01}$ | $7.21_{0.01}$ | -2.1 |
| | | | DeepSeek-VL2* | 28B | $0.08_{0.03}$ | $0.14_{0.04}$ | -0.06 | $4.30_{0.09}$ | $6.86_{0.09}$ | -2.55 |
| | | | DocOwl-1.5-Omni | 8B | $0.0_{0.0}$ | $0.0_{0.0}$ | 0 | $1.37_{0.01}$ | $4.07_{0.02}$ | -2.7 |
| | | | InternLM-XComposer2-VL | 7B | $0.07_{0.01}$ | $0.09_{0.01}$ | -0.02 | $8.97_{0.02}$ | $8.51_{0.01}$ | 0.46 |
| | | | InternLM-XComposer2-VL-4K | 7B | $0.05_{0.02}$ | $0.05_{0.02}$ | 0.00 | $7.67_{0.10}$ | $7.72_{0.10}$ | -0.04 |
| | | | InternLM-XComposer2.5-VL* | 7B | $0.11_{0.04}$ | $0.12_{0.04}$ | -0.01 | $10.95_{0.11}$ | $11.43_{0.12}$ | -0.48 |
| | Hard | | InternVL-V2* | 40B | $0.48_{0.07}$ | $0.74_{0.08}$ | -0.26 | $12.57_{0.14}$ | $13.31_{0.15}$ | -0.74 |
| | | | InternVL-V2* | 76B | $0.56_{0.07}$ | $0.66_{0.08}$ | -0.10 | $15.31_{0.14}$ | $14.58_{0.13}$ | 0.73 |
| | | | MiniCPM-V2.5 | 8B | $0.09_{0.0}$ | $0.08_{0.0}$ | 0.01 | $7.39_{0.02}$ | $7.89_{0.01}$ | -0.5 |
| | | | Monkey | 7B | $0.12_{0.01}$ | $0.07_{0.0}$ | 0.05 | $6.30_{0.01}$ | $6.68_{0.03}$ | -0.32 |
| | | | Ovis2* | 34B | $3.73_{0.19}$ | $2.66_{0.15}$ | 1.07 | $20.62_{0.24}$ | $18.72_{0.21}$ | 1.91 |
| | | | Qwen-VL | 7B | $0.01_{0.0}$ | $0.01_{0.0}$ | 0 | $1.17_{0.01}$ | $0.11_{0.0}$ | 1.06 |
| | | | Qwen2-VL* | 7B | $18.33_{0.37}$ | $27.58_{0.44}$ | -9.26 | $43.55_{0.34}$ | $54.24_{0.34}$ | -10.69 |
| | | | Qwen2-VL* | 72B | $15.30_{0.36}$ | $27.35_{0.44}$ | -12.05 | $39.71_{0.32}$ | $53.63_{0.35}$ | -13.91 |
| | | | Qwen2.5-VL* | 7B | $19.84_{0.38}$ | $12.11_{0.32}$ | 7.74 | $45.38_{0.35}$ | $31.23_{0.34}$ | 14.14 |
| | | | Qwen2.5-VL* | 72B | $\underline{31.53_{0.47}}$ | $\underline{29.47_{0.44}}$ | 2.06 | $\mathbf{\underline{56.69_{0.35}}}$ | $\underline{54.41_{0.34}}$ | 2.27 |
| | | | Yi-VL | 34B | $0.0_{0.0}$ | $0.0_{0.0}$ | 0 | $4.12_{0.0}$ | $1.81_{0.01}$ | 2.31 |

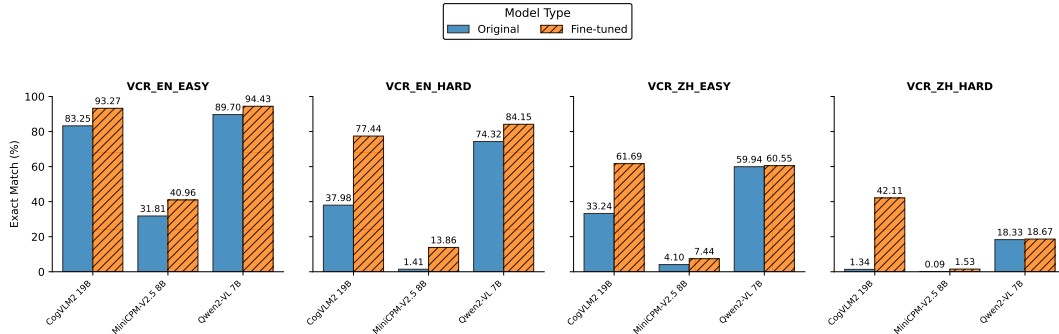

Figure 5: Exact Match performance on VCR before and after LoRA fine-tuning for selected models. The results demonstrate varying degrees of improvement across different tasks and models, highlighting the heterogeneous responses to fine-tuning.

## 4.5 HUMAN EVALUATION

We recruited 7 volunteers to perform human evaluation on a subset of the samples from our datasets. Two out of the seven evaluators are native English speakers, while five are native Chinese speakers who are also fluent in English[4]. All volunteers have earned postgraduate degrees, majoring in one of the following fields: biology, statistics, computer science, and economics. The evaluations were conducted on a voluntary basis, and participants received no rewards.

We gave the volunteers the following instructions: (1) We asked the volunteers to focus on the puzzles. Each example in the hard collection may require 30 seconds to 2 minutes of focused attention, and (2) we asked the volunteers to utilize the context rather than directly brute-force the puzzle.

Every sample is solved by at least 3 volunteers. In English, we release the exact match score in 2 splits: all errors counted (All), and only counting errors not related to dates and person names (Filtered).

The human evaluation results are shown in Table 3. Although current SOTA models suffer from the challenge, fluent speakers can easily achieve more than 90 percent accuracy across difficulties. Please refer to Table 8 to compare all models with human evaluation results using the same test cases.

Table 3: Human evaluation results on the VCR task in terms of exact matches. $N$ is the number of puzzles in each language.

| | $VCR_{EN, EASY}$ (N = 169) | | $VCR_{EN, HARD}$ (N = 169) | | $VCR_{ZH, EASY}$ (N = 188) | | $VCR_{ZH, HARD}$ (N = 188) | |
| | Mean (%) | SD (%) | Mean (%) | SD (%) | Mean (%) | SD (%) | Mean (%) | SD (%) |
|---|---|---|---|---|---|---|---|---|
| All | 96.65 | 0.34 | 91.12 | 1.18 | 98.58 | 0.31 | 91.84 | 0.81 |
| Filtered | 98.62 | 0.34 | 97.63 | 2.13 | 99.47 | 0.00 | 96.63 | 1.11 |

## 5 RELATED WORK

**Visual Question Answering (VQA).** Several datasets have been proposed for visual question answering VQA (Antol et al., 2015; Zhang et al., 2016; Goyal et al., 2017; Mishra et al., 2019b). FVQA (Wang et al., 2018) and OK-VQA(Marino et al., 2019) are datasets about knowledge-based visual question answering and contains questions that necessitate the usage of external knowledge resources. CLEVR (Johnson et al., 2017) is a synthetic VQA dataset that mainly focuses on visual reasoning abilities. Recognizing the need to develop VQA models that can understand text, Text-VQA (Singh et al., 2019; Biten et al., 2019; Mishra et al., 2019a; Wang et al., 2020) aims to read and reason about texts embedded within images in the context of image-question answering. Several datasets (Singh et al., 2019; Biten et al., 2019; Mishra et al., 2019a) have been developed for the Text-VQA task, such as the TextVQA dataset (Singh et al., 2019) and the ST-VQA dataset (Biten et al., 2019)

---

[4]The TOEFL scores of the non-native English-speaking participants range from 102/120 to 112/120.

on natural images, the OCR-VQA dataset (Mishra et al., 2019b) on book or movie covers, the InfographicVQA (Mathew et al., 2022) dataset on infographics, and the DocVQA dataset (Mathew et al., 2021) on document images.

**Vision Language Model.** Vision-language models are designed for tasks that involve understanding and generating content from images and text (Sun et al., 2023; Liu et al., 2023b; Laurençon et al., 2023; 2024a). For example, models have been developed to combine Llama3 with advanced vision-language processing capabilities to handle complex multimodal tasks (Yu et al., 2024; Xu et al., 2024; Hu et al., 2023; Yu et al., 2023; Wang et al., 2023b; Dong et al., 2024a). Qwen-VL (Bai et al., 2023) enhances visual-linguistic representations for more accurate contextual interpretations, while OpenGVLab-InternVL-Chat (Chen et al., 2023; 2024c) merges the InternVL framework with interactive chat capabilities. These studies typically employ a multimodal encoder (Radford et al., 2021; Zhai et al., 2023; Wu et al., 2022) to process multimodal data, which is then mapped to the same input space of the language model. General-purpose models such as the GPT-4 series models (Ouyang et al., 2022; OpenAI et al., 2023), the Claude series models (Anthropic, 2024), the Gemini series models (Team et al., 2024a) and the Reka series models (Team et al., 2024b) have also been adapted for vision-language tasks, demonstrating strong performance in multimodal tasks. Finally, DocLLM (Wang et al., 2023a) specializes in document understanding by integrating visual and textual data to enhance the interpretation and generation of document-related content. These models collectively represent significant advancements in vision-language integration, contributing unique capabilities and enhancements to the understanding and generation of multimodal information.

**Optical Character Recognition (OCR).** OCR (Nagy, 2000) and its subproblems (Howe, 2013; Smith, 1995; Shafait et al., 2008; Frinken et al., 2011) have been well-studied in the literature in the constrained setting. However, classical OCR methods often cannot perform well on images captured in the wild in an unconstrained setting. Many new methods have been developed for advancing scene-text recognition on camera-captured images (Bissacco et al., 2013; Gupta et al., 2016; Huang et al., 2014; Jaderberg et al., 2014; Wang et al., 2012; Shi et al., 2017; Zhou et al., 2017; Lee & Osindero, 2016). In addition to the detection and recognition of OCR tasks, visual question answering has emerged as an important downstream task in the OCR literature. With the development of Text-VQA, new methods for improving the reading abilities in VQA utilizing OCR have been proposed. For example, LoRRA (Singh et al., 2019) extends a VQA model Pythia (Jiang et al., 2018) with an OCR module to better handle Text-VQA tasks. TAP (Yang et al., 2021) incorporates scene texts that are generated from OCR engines during pretraining to further improve Text-VQA capabilities.

## 6 CONCLUSION

In this work, we introduced the VCR task, a novel vision-language challenge aimed at promoting the integration of visual and textual modalities, including text embedded in both natural language tokens and image formats and highly obscured text embedded in the image. We developed a specialized pipeline to create a dataset tailored to this task, utilizing correlated image-text pairs. This task stands out from existing methods by requiring a more profound integration of visual cues and partially obscured text, highlighting its uniqueness and importance in the field.

We conducted extensive evaluations of state-of-the-art vision-language models (VLMs) in both English and Chinese. The results demonstrated significant room for improvement, suggesting that current models have not yet fully exploited the capabilities necessary for VCR. We selected models representing both the highest and average performance tiers for additional fine-tuning with our dataset. Although fine-tuning exhibited potential for enhancing VCR capabilities, it did not consistently result in significant improvements, indicating the complexity and challenges of adapting models to this task.

By introducing the VCR task and its specialized dataset, we aim to advance research in vision-language interaction. The unique challenges of VCR seek to improve model development and training, extending the limits of multimodal AI. VCR provides a controllable testbed for fine-grained analysis of model behavior across languages, difficulty levels, image inclusion, and fine-tuning stages. We invite the community to utilize our dataset and develop innovative strategies to boost the performance of vision-language models.

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

## A    STATISTICS OF **VCR-WIKI** AND **VCR-HIDDEN**

This section provides a comprehensive statistical analysis of both our main dataset (**VCR-WIKI**) and our hidden test set (**VCR-HIDDEN**). Table 4 presents key metrics including split sizes, image dimensions, and text obscuration patterns across languages. Note that the English and Chinese configurations maintain consistent statistical properties between their respective Easy and Hard variants since the two difficulties keeps the exact same dataset splits and covered texts, and only differs in covered area.

Table 4: Basic statistics of **VCR-WIKI** and **VCR-HIDDEN**. Note that each language's Easy and Hard configurations share the same statistics. We report the mean, standard deviation, and the 5[th] and 95[th] percentile ($\eta_{.05}$ and $\eta_{.95}$) for the stacked image height and the number of obscured text spans of **VCR-WIKI**. Unit is in pixels. "Hidden Test" means the hidden test set (**VCR-HIDDEN**).

| | # Train | # Val | # Test | # Hidden Test | $VI + TEI$ Image Height | | | | # Obscured Text Spans | | | |
|---|---|---|---|---|---|---|---|---|---|---|---|---|
| | | | | | Mean | SD | $\eta_{.5}$ | $\eta_{.95}$ | Mean | SD | $\eta_{.5}$ | $\eta_{.95}$ |
| English | 2095733 | 5000 | 5000 | 100 | 375.52 | 106.01 | 253 | 564 | 1.62 | 0.63 | 1 | 3 |
| Chinese | 336448 | 5000 | 5000 | 100 | 360.44 | 102.76 | 239 | 562 | 2.06 | 0.94 | 1 | 4 |

## B    DETAILS ABOUT THE EVALUATED MODELS

This section provides comprehensive specifications of all vision-language models (VLMs) evaluated in this study. We assess a diverse range of models spanning both proprietary and open-source ecosystems, representing the state-of-the-art in VLM capabilities as of October 2024 for **VCR-WIKI** and February 2025 for **VCR-HIDDEN**. Our evaluation includes models with varying parameter sizes, architectural designs, and training paradigms to ensure a thorough assessment of frontier visual-textual reasoning abilities. The models are categorized below as either closed-source (proprietary) or open-source, with detailed specifications and links to the models provided in Table 5.

**Closed-source Models.**    We evaluate several most advanced proprietary models with either their official APIs or APIs provided by OpenRouter. The evaluated models include o1 (o1-2024-12-17), GPT-4o (gpt-4o-2024-0513), GPT-4 Turbo (gpt-4-turbo-2024-04-09), GPT-4V (gpt-4-1106-vision-preview) (Jaech et al., 2024; Ouyang et al., 2022; OpenAI et al., 2023), Claude 3 Opus (claude-3-opus-20240229), Claude 3.5 Sonnet (claude-3-5-sonnet-20240620), Claude 3.7 Sonnet (claude-3-7-sonnet-20250219) (Anthropic, 2024), Gemini 1.5 Flash (gemini-1.5-flash-8b), Gemini 1.5 Pro (gemini-1.5-pro-001), Gemini 2 Flash Lite (gemini-2.0-flash-lite), Gemini 2 Flash (gemini-2.0-flash) (Team et al., 2024a), Reka Core (reka-core-20240501) (Team et al., 2024b), and Qwen-VL-Max (tested in May 2024) (Bai et al., 2023).

**Open-source Models.**    We evaluate open-source model families with the best performance on the OpenVLM Leaderboard as of May 2024 and state-of-the-art Chinese VLM models. The evaluated models include Cambrian-1 (Tong et al., 2024), CogVLM2-19B (Hong et al., 2024), DeepSeek-VL-7B-Chat (Lu et al., 2024), DeepSeek-VL2 Series (Wu et al., 2024b), DocOwl-1.5-Omni (Hu et al., 2024a), GLM-4v (GLM et al., 2024), Idefics2-8B (Laurençon et al., 2024b), Idefics3-8B (Laurençon et al., 2024a), InternVL 1.5, InternVL 2, InternVL 2.5 (Chen et al., 2024c;b), InternLM-XComposer2-VL-7B (Dong et al., 2024a), InternLM-XComposer2.5-VL (Zhang et al., 2024), InternLM-XComposer2-VL-4K (Dong et al., 2024b), Llama-3.2 (Dubey et al., 2024), MiniCPM-V2.5, MiniCPM-V2.6 (Hu et al., 2024b), MiniMax-VL-01 (MiniMax et al., 2025), Molmo Series (Deitke et al., 2024), Monkey (Liu et al., 2024b; Li et al., 2023b), Phi 3.5-vision (Abdin et al., 2024a), Phi-4-multimodal (Abdin et al., 2024b), Pixtral (Mistral, 2024), Qwen-VL-Chat (Bai et al., 2023), QVQ-preview (Team, 2024), Qwen2-VL Series (Wang et al., 2024), Qwen2.5-VL Series (Bai et al., 2025) and Yi-VL (01.AI et al., 2024). Out of these models, Cambrian-1, Idefics3 and Llama-3.2 are English-only models, and CogVLM2-Llama3-19B-Chat has its Chinese variant, CogVLM2-Llama3-19B-Chinese-Chat.

Table 5: Model specifications

| Model name | Model size | Open-sourced |
|---|---|---|
| Claude 3 Opus, 3.5 Sonnet, 3.7 Sonnet | - | ✗ |
| Gemini 1.5 Flash, 1.5 Pro, 2.0 Flash, 2.0 Flash Lite | - | ✗ |
| GPT-4 Turbo, 4V, 4o, 4o-mini, o1 | - | ✗ |
| Qwen-VL-Max | - | ✗ |
| Reka Core | - | ✗ |
| Cambrian-1 Series [5] | 8B, 13B, 34B | ✓ |
| CogVLM2 English, Chinese [6] | 19B | ✓ |
| DeepSeek-VL Series [7] | 1.3B, 7B | ✓ |
| DeepSeek-VL2 Series [8] | 16B, 28B | ✓ |
| DocOwl-1.5-Omni[9] | 8B | ✓ |
| GLM-4V [10], | 9B | ✓ |
| Idefics 2 [11], 3 [12] | 8B, 8B | ✓ |
| InternLM-XComposer2-VL, 4KHD [13] | 7B, 7B | ✓ |
| InternLM-XComposer2.5-VL [14] | 7B | ✓ |
| InternVL-V1.5 [15] | 26B | ✓ |
| InternVL-V2 Series [16] | 1B, 2B, 4B, 8B, 26B, 40B | ✓ |
| InternVL-V2.5 Series [17] | 1B, 2B, 4B, 8B, 26B, 38B | ✓ |
| Llama-3.2-Vision Series [18] | 11B, 90B | ✓ |
| MiniCPM-V2.5, V2.6 [19] | 8B, 8B | ✓ |
| MiniMax-VL-01 [20] | 456B | ✓ |
| Molmo Series[21] | 1B, 7B, 7B, 72B | ✓ |
| Monkey[22] | 7B | ✓ |
| Ovis1.6-Gemma2 Series [23] | 9B, 27B | ✓ |
| Ovis2 Series [24] | 1B, 2B, 4B, 8B, 16B, 34B | ✓ |
| Phi-3.5-vision [25] | 4B | ✓ |
| Phi-4-multimodal [26] | 6B | ✓ |
| Pixtral [27] | 12B | ✓ |
| Qwen-VL [28] | 7B | ✓ |
| Qwen2-VL Series [29] | 2B, 7B, 72B | ✓ |
| Qwen2.5-VL Series [30] | 3B, 7B, 72B | ✓ |
| QVQ [31] | 72B | ✓ |
| Yi-VL[32] | 6B, 34B | ✓ |

---

[5]https://huggingface.co/collections/nyu-visionx/cambrian-1-models-666fa7116d5420e514b0f23c
[6]https://huggingface.co/collections/THUDM/cogvlm2-6645f36a29948b67dc4eef75
[7]https://huggingface.co/collections/deepseek-ai/deepseek-vl-65f295948133d9cf92b706d3
[8]https://huggingface.co/collections/deepseek-ai/deepseek-vl2-675c22accc456d3beb4613ab
[9]https://huggingface.co/mPLUG/DocOwl1.5-Omni
[10]https://huggingface.co/THUDM/glm-4v-9b
[11]https://huggingface.co/HuggingFaceM4/idefics2-8b
[12]https://huggingface.co/HuggingFaceM4/Idefics3-8B
[13]https://huggingface.co/collections/internlm/internlm-xcomposer2-65b3706bf5d76208998e7477
[14]https://huggingface.co/internlm/internlm-xcomposer2d5-7b
[15]https://huggingface.co/OpenGVLab/InternVL-Chat-V1-5
[16]https://huggingface.co/collections/OpenGVLab/internvl20-667d3961ab5eb12c7ed1463e
[17]https://huggingface.co/collections/OpenGVLab/internvl25-673e1019b66e2218f68d7c1c
[18]https://huggingface.co/collections/meta-llama/llama-32-66f448ffc8c32f949b04c8cf
[19]https://huggingface.co/collections/openbmb/minicpm-65d48bf958302b9fd25b698f
[20]https://huggingface.co/MiniMaxAI/MiniMax-VL-01
[21]https://huggingface.co/collections/allenai/molmo-66f379e6fe3b8ef090a8ca19
[22]https://huggingface.co/echo840/Monkey-Chat
[23]https://huggingface.co/collections/AIDC-AI/ovis16-66eadbe52f79fb99cc122c08
[24]https://huggingface.co/collections/AIDC-AI/ovis2-67ab36c7e497429034874464
[25]https://huggingface.co/microsoft/Phi-3.5-vision-instruct
[26]https://huggingface.co/microsoft/Phi-4-multimodal-instruct
[27]https://huggingface.co/mistralai/Pixtral-12B-2409

# C HIDDEN TEST RESULTS ON VCR-HIDDEN

Table 6: Performance of vision language models on **VCR-HIDDEN** in English. We label the best result for each setting and metric with **bold fonts**. A superscript of * marks that the model was released after the initial public release of the **VCR-WIKI** dataset (June 10, 2024). Subscripts show bootstrapped standard deviation.

| Open/closed source | Model name | Model size | Exact match (%) ↑ | | | Jaccard index (%) ↑ | | |
|---|---|---|---|---|---|---|---|---|
| | | | $VI + TEI$ | $TEI$ | $\Delta$ | $VI + TEI$ | $TEI$ | $\Delta$ |
| Closed | Claude 3.5 Sonnet | - | $75.50_{2.99}$ | $78.50_{2.88}$ | -3.00 | $85.12_{1.96}$ | $86.63_{2.08}$ | -1.51 |
| | Claude 3.7 Sonnet* | - | $74.00_{2.99}$ | $58.50_{3.38}$ | 15.50 | $81.53_{2.34}$ | $61.47_{3.26}$ | 20.06 |
| | GPT-4o | - | $41.50_{3.43}$ | $45.50_{3.55}$ | -4.00 | $48.09_{3.39}$ | $51.51_{3.30}$ | -3.43 |
| | GPT-4o Mini | - | $60.50_{3.48}$ | $69.00_{3.23}$ | -8.50 | $71.02_{2.86}$ | $81.19_{2.11}$ | -10.17 |
| | o1* | - | $26.50_{3.17}$ | $18.50_{2.64}$ | 8.00 | $31.17_{3.11}$ | $22.06_{2.79}$ | 9.11 |
| | Gemini 2.0 Flash* | - | $48.50_{3.55}$ | $42.00_{3.55}$ | 6.50 | $66.35_{2.76}$ | $61.16_{2.76}$ | 5.20 |
| | Gemini 2.0 Flash Lite* | - | $39.00_{3.33}$ | $32.00_{3.27}$ | 7.00 | $51.92_{2.97}$ | $45.29_{3.18}$ | 6.63 |
| | Gemini 1.5 Flash | 8B | $37.00_{3.35}$ | $44.00_{3.52}$ | -7.00 | $46.74_{3.10}$ | $55.55_{3.14}$ | -8.81 |
| | Gemini 1.5 Pro | - | $37.50_{3.43}$ | $47.00_{3.51}$ | -9.50 | $53.52_{2.87}$ | $62.26_{2.76}$ | -8.75 |
| | Grok 2 Vision* | - | $16.50_{2.43}$ | $26.00_{3.15}$ | -9.50 | $36.05_{2.47}$ | $43.83_{2.79}$ | -7.78 |
| Open | Cambrian-1* | 13B | $27.50_{3.21}$ | $56.50_{3.47}$ | -29.00 | $41.34_{2.74}$ | $71.41_{2.64}$ | -30.07 |
| | Cambrian-1* | 34B | $39.50_{3.46}$ | $50.00_{3.49}$ | -10.50 | $57.15_{2.86}$ | $68.26_{2.63}$ | -11.11 |
| | CogVLM2 | 19B | $53.00_{3.72}$ | $51.00_{3.56}$ | 2.00 | $64.97_{2.76}$ | $61.01_{2.98}$ | 3.97 |
| | DeepSeek-VL2-Small* | 16B | $5.00_{1.53}$ | $21.50_{2.90}$ | -16.50 | $6.56_{1.62}$ | $31.30_{2.96}$ | -24.74 |
| | DeepSeek-VL2* | 28B | $23.50_{2.93}$ | $32.00_{3.31}$ | -8.50 | $31.75_{2.94}$ | $45.50_{2.90}$ | -13.75 |
| | GLM-4v | 9B | $44.50_{3.48}$ | $52.50_{3.63}$ | -8.00 | $59.03_{2.88}$ | $68.24_{2.63}$ | -9.21 |
| | Idefics2 | 8B | $9.50_{2.05}$ | $23.00_{2.97}$ | -13.50 | $20.45_{2.12}$ | $38.45_{2.81}$ | -18.00 |
| | Idefics3* | 8B | $18.00_{2.78}$ | $24.50_{3.02}$ | -6.50 | $33.96_{2.49}$ | $38.47_{2.84}$ | -4.51 |
| | InternLM-XComposer2-VL | 7B | $9.50_{2.11}$ | $7.50_{1.86}$ | 2.00 | $23.49_{2.14}$ | $23.31_{2.06}$ | 0.17 |
| | InternLM-XComposer2-VL-4K | 7B | $18.50_{2.78}$ | $19.00_{2.78}$ | -0.50 | $34.12_{2.59}$ | $34.78_{2.62}$ | -0.66 |
| | InternLM-XComposer2.5-VL* | 7B | $14.50_{2.54}$ | $12.50_{2.25}$ | 2.00 | $29.93_{2.55}$ | $28.32_{2.31}$ | 1.61 |
| | InternVL1.5 | 26B | $29.50_{3.19}$ | $38.50_{3.28}$ | -9.00 | $46.58_{2.79}$ | $49.69_{3.06}$ | -3.12 |
| | InternVL2* | 1B | $17.00_{2.68}$ | $13.00_{2.34}$ | 4.00 | $36.50_{2.58}$ | $28.52_{2.34}$ | 7.98 |
| | InternVL2* | 2B | $16.50_{2.59}$ | $13.50_{2.48}$ | 3.00 | $35.97_{2.44}$ | $29.76_{2.48}$ | 6.20 |
| | InternVL2* | 4B | $17.00_{2.56}$ | $15.50_{2.51}$ | 1.50 | $34.46_{2.46}$ | $32.42_{2.48}$ | 2.05 |
| | InternVL2* | 8B | $22.00_{2.99}$ | $15.50_{2.54}$ | 6.50 | $39.44_{2.84}$ | $31.80_{2.65}$ | 7.65 |
| | InternVL2* | 26B | $48.00_{3.42}$ | $46.50_{3.53}$ | 1.50 | $56.73_{3.23}$ | $54.97_{3.06}$ | 1.75 |
| | InternVL2* | 40B | $50.50_{3.48}$ | $44.50_{3.53}$ | 6.00 | $59.18_{3.16}$ | $55.99_{3.13}$ | 3.19 |
| | InternVL2* | 76B | $50.50_{3.52}$ | $49.50_{3.60}$ | 1.00 | $62.93_{2.92}$ | $62.07_{3.07}$ | 0.86 |
| | InternVL2.5* | 1B | $65.00_{3.43}$ | $56.50_{3.30}$ | 8.50 | $79.53_{2.13}$ | $70.12_{2.63}$ | 9.41 |
| | InternVL2.5* | 2B | $73.00_{3.19}$ | $62.00_{3.36}$ | 11.00 | $83.36_{2.15}$ | $74.61_{2.70}$ | 8.74 |
| | InternVL2.5* | 4B | $74.00_{3.14}$ | $67.50_{3.34}$ | 6.50 | $84.62_{2.05}$ | $78.66_{2.39}$ | 5.95 |
| | InternVL2.5* | 8B | $74.50_{3.00}$ | $63.00_{3.47}$ | 11.50 | $84.65_{2.14}$ | $74.74_{2.56}$ | 9.91 |
| | InternVL2.5* | 26B | $\mathbf{88.00_{2.31}}$ | $84.00_{2.62}$ | 4.00 | $\mathbf{94.60_{1.12}}$ | $90.17_{1.69}$ | 4.43 |
| | InternVL2.5* | 38B | $79.50_{2.76}$ | $78.50_{2.94}$ | 1.00 | $89.72_{1.48}$ | $87.05_{1.91}$ | 2.66 |
| | InternVL2.5* | 78B | $83.50_{2.70}$ | $79.50_{2.83}$ | 4.00 | $91.47_{1.39}$ | $87.60_{1.88}$ | 3.87 |
| | Llama-3.2-Vision* | 11B | $46.50_{3.53}$ | $35.00_{3.33}$ | 11.50 | $60.12_{2.85}$ | $50.81_{2.99}$ | 9.31 |
| | Llama-3.2-Vision* | 90B | $51.00_{3.52}$ | $33.50_{3.20}$ | 17.50 | $63.21_{2.88}$ | $50.55_{2.86}$ | 12.66 |
| | MiniCPM-V2.5 | 8B | $15.00_{2.42}$ | $12.50_{2.34}$ | 2.50 | $27.65_{2.60}$ | $25.05_{2.33}$ | 2.60 |
| | MiniCPM-V2.6* | 8B | $42.50_{3.52}$ | $41.00_{3.47}$ | 1.50 | $53.30_{3.12}$ | $53.09_{3.11}$ | 0.21 |
| | MiniMax-VL-01* | 456B | $48.50_{3.52}$ | $51.50_{3.60}$ | -3.00 | $62.20_{2.74}$ | $63.08_{2.93}$ | -0.88 |
| | Molmo* | 72B | $27.00_{3.18}$ | $32.50_{3.22}$ | -5.50 | $42.66_{2.85}$ | $49.30_{2.90}$ | -6.64 |
| | MolmoE* | 1B | $0.00_{0.00}$ | $0.00_{0.00}$ | 0.00 | $5.83_{0.46}$ | $9.95_{0.71}$ | -4.12 |
| | Molmo-D* | 7B | $22.50_{2.94}$ | $16.00_{2.62}$ | 6.50 | $40.03_{2.78}$ | $34.52_{2.49}$ | 5.51 |
| | Molmo-O* | 7B | $20.50_{2.84}$ | $18.50_{2.78}$ | 2.00 | $40.52_{2.61}$ | $41.05_{2.40}$ | -0.53 |
| | Ovis1.6-Gemma2* | 9B | $41.00_{3.42}$ | $37.50_{3.34}$ | 3.50 | $54.67_{3.11}$ | $51.10_{3.11}$ | 3.58 |
| | Ovis1.6-Gemma2* | 27B | $38.00_{3.38}$ | $42.50_{3.67}$ | -4.50 | $52.45_{2.89}$ | $54.81_{3.06}$ | -2.36 |
| | Ovis2* | 1B | $53.50_{3.60}$ | $56.00_{3.61}$ | -2.50 | $68.30_{2.81}$ | $74.48_{2.28}$ | -6.17 |
| | Ovis2* | 2B | $55.00_{3.62}$ | $51.50_{3.47}$ | 3.50 | $69.13_{2.71}$ | $68.80_{2.63}$ | 0.33 |
| | Ovis2* | 4B | $71.50_{3.05}$ | $65.00_{3.44}$ | 6.50 | $80.37_{2.35}$ | $76.49_{2.52}$ | 3.88 |
| | Ovis2* | 8B | $59.50_{3.38}$ | $61.50_{3.75}$ | -2.00 | $72.13_{2.58}$ | $72.33_{2.70}$ | -0.20 |
| | Ovis2* | 16B | $47.00_{3.62}$ | $49.00_{3.48}$ | -2.00 | $58.93_{3.03}$ | $62.20_{2.88}$ | -3.28 |
| | Ovis2* | 34B | $62.00_{3.52}$ | $60.50_{3.43}$ | 1.50 | $73.30_{2.70}$ | $71.34_{2.85}$ | 1.97 |
| | Phi-3.5-Vision* | 4B | $12.00_{2.34}$ | $15.00_{2.64}$ | -3.00 | $25.50_{2.35}$ | $27.82_{2.48}$ | -2.32 |
| | Phi-4-Multimodal* | 6B | $6.00_{1.67}$ | $16.00_{2.67}$ | -10.00 | $17.47_{1.85}$ | $29.17_{2.49}$ | -11.69 |
| | Pixtral* | 12B | $27.50_{3.10}$ | $19.00_{2.74}$ | 8.50 | $40.98_{2.89}$ | $36.50_{2.67}$ | 4.48 |
| | QvQ-Preview* | 72B | $66.00_{3.40}$ | $62.50_{3.25}$ | 3.50 | $75.76_{2.66}$ | $72.57_{2.73}$ | 3.19 |
| | Qwen-VL | 7B | $15.50_{2.44}$ | $12.50_{2.36}$ | 3.00 | $33.74_{2.35}$ | $28.27_{2.33}$ | 5.47 |
| | Qwen2-VL* | 2B | $78.50_{3.08}$ | $80.00_{2.91}$ | -1.50 | $86.28_{2.04}$ | $87.03_{1.90}$ | -0.75 |
| | Qwen2-VL* | 7B | $80.50_{2.78}$ | $83.50_{2.57}$ | -3.00 | $86.42_{2.07}$ | $88.75_{1.93}$ | -2.32 |
| | Qwen2-VL* | 72B | $85.50_{2.60}$ | $\mathbf{84.50_{2.44}}$ | 1.00 | $90.62_{1.67}$ | $89.77_{1.83}$ | 0.85 |
| | Qwen2.5-VL* | 3B | $65.50_{3.47}$ | $66.00_{3.27}$ | -0.50 | $73.32_{2.67}$ | $76.42_{2.58}$ | -3.10 |
| | Qwen2.5-VL* | 7B | $58.00_{3.42}$ | $74.00_{3.00}$ | -16.00 | $65.75_{3.10}$ | $83.90_{2.07}$ | -18.15 |
| | Qwen2.5-VL* | 72B | $86.50_{2.43}$ | $79.50_{2.88}$ | 7.00 | $92.28_{1.52}$ | $86.43_{2.01}$ | 5.86 |

---

[28] https://huggingface.co/Qwen/Qwen-VL-Chat

[29] https://huggingface.co/collections/Qwen/qwen2-vl-66cee7455501d7126940800d

[30] https://huggingface.co/collections/Qwen/qwen25-vl-6795ffac22b334a837c0f9a5

[31] https://huggingface.co/Qwen/QVQ-72B-Preview

[32] https://huggingface.co/collections/01-ai/yi-vl-663f557228538eae745769f3

Table 7: Performance of vision language models on **VCR-HIDDEN** in Chinese. We label the best result for each setting and metric with **bold fonts**. A superscript of * marks that the model was released after the initial public release of the **VCR-WIKI** dataset (June 10, 2024). Subscripts show bootstrapped standard deviation.

| Open/closed source | Model name | Model size | Exact match (%) ↑ | | | Jaccard index (%) ↑ | | |
|---|---|---|---|---|---|---|---|---|
| | | | $VI+TEI$ | $TEI$ | $\Delta$ | $VI+TEI$ | $TEI$ | $\Delta$ |
| Closed | Claude 3.5 Sonnet | - | $1.50_{0.82}$ | $1.00_{0.67}$ | 0.50 | $14.17_{1.16}$ | $13.93_{1.20}$ | 0.24 |
| | Claude 3.7 Sonnet* | - | $3.00_{1.16}$ | $0.50_{0.50}$ | 2.50 | $9.93_{1.44}$ | $0.92_{0.52}$ | 9.02 |
| | GPT-4o | - | $1.50_{0.89}$ | $0.50_{0.50}$ | 1.00 | $3.78_{1.01}$ | $2.22_{0.66}$ | 1.56 |
| | o1* | - | $0.00_{0.00}$ | $0.00_{0.00}$ | 0.00 | $0.00_{0.00}$ | $0.00_{0.00}$ | 0.00 |
| | Gemini 2.0 Flash* | - | $1.00_{0.70}$ | $1.00_{0.69}$ | 0.00 | $12.36_{1.11}$ | $11.77_{1.09}$ | 0.59 |
| | Gemini 2.0 Flash Lite* | - | $0.50_{0.52}$ | $0.50_{0.49}$ | 0.00 | $7.47_{0.75}$ | $6.97_{0.88}$ | 0.50 |
| | Gemini 1.5 Flash | 8B | $0.50_{0.52}$ | $0.50_{0.51}$ | 0.00 | $8.51_{0.94}$ | $9.90_{0.93}$ | -1.39 |
| | Gemini 1.5 Pro | - | $1.00_{0.70}$ | $1.00_{0.70}$ | 0.00 | $12.15_{1.12}$ | $10.95_{1.04}$ | 1.21 |
| | Grok 2 Vision* | - | $0.00_{0.00}$ | $0.00_{0.00}$ | 0.00 | $6.27_{0.52}$ | $5.33_{0.43}$ | 0.94 |
| Open | Cambrian-1* | 13B | $0.00_{0.00}$ | $0.00_{0.00}$ | 0.00 | $4.06_{0.47}$ | $6.83_{0.49}$ | -2.76 |
| | Cambrian-1* | 34B | $0.00_{0.00}$ | $0.00_{0.00}$ | 0.00 | $1.29_{0.27}$ | $4.91_{0.44}$ | -3.62 |
| | CogVLM2-Chinese | 19B | $3.00_{1.17}$ | $1.50_{0.82}$ | 1.50 | $16.05_{1.47}$ | $13.15_{1.21}$ | 2.90 |
| | DeepSeek-VL2-Small* | 16B | $0.00_{0.00}$ | $0.00_{0.00}$ | 0.00 | $1.30_{0.39}$ | $2.59_{0.54}$ | -1.28 |
| | DeepSeek-VL2* | 28B | $0.00_{0.00}$ | $0.00_{0.00}$ | 0.00 | $6.82_{0.64}$ | $7.09_{0.73}$ | -0.26 |
| | GLM-4v | 9B | $1.50_{0.84}$ | $2.00_{0.98}$ | -0.50 | $14.65_{1.38}$ | $9.78_{1.12}$ | 4.88 |
| | Idefics2 | 8B | $0.00_{0.00}$ | $0.00_{0.00}$ | 0.00 | $1.54_{0.33}$ | $2.38_{0.37}$ | -0.83 |
| | Idefics3* | 8B | $0.00_{0.00}$ | $0.00_{0.00}$ | 0.00 | $3.68_{0.47}$ | $3.04_{0.41}$ | 0.64 |
| | InternLM-XComposer2-VL | 7B | $0.00_{0.00}$ | $0.00_{0.00}$ | 0.00 | $8.34_{0.73}$ | $8.04_{0.62}$ | 0.30 |
| | InternLM-XComposer2-VL-4K | 7B | $1.00_{0.72}$ | $1.00_{0.70}$ | 0.00 | $8.97_{0.99}$ | $8.37_{0.93}$ | 0.59 |
| | InternLM-XComposer2.5-VL* | 7B | $0.50_{0.50}$ | $1.50_{0.84}$ | -1.00 | $9.53_{0.83}$ | $11.35_{1.07}$ | -1.81 |
| | InternVL1.5 | 26B | $0.50_{0.51}$ | $1.00_{0.71}$ | -0.50 | $10.51_{0.92}$ | $9.93_{1.08}$ | 0.58 |
| | InternVL2* | 1B | $0.50_{0.50}$ | $0.00_{0.00}$ | 0.50 | $10.40_{0.84}$ | $8.35_{0.74}$ | 2.06 |
| | InternVL2* | 2B | $2.00_{0.97}$ | $0.50_{0.48}$ | 1.50 | $11.29_{1.16}$ | $7.98_{0.93}$ | 3.31 |
| | InternVL2* | 4B | $1.00_{0.69}$ | $0.50_{0.50}$ | 0.50 | $10.11_{0.95}$ | $9.09_{0.87}$ | 1.02 |
| | InternVL2* | 8B | $1.50_{0.83}$ | $1.00_{0.69}$ | 0.50 | $10.27_{1.04}$ | $8.49_{1.02}$ | 1.78 |
| | InternVL2* | 26B | $0.50_{0.50}$ | $0.50_{0.48}$ | 0.00 | $10.92_{0.95}$ | $9.09_{0.94}$ | 1.83 |
| | InternVL2* | 40B | $2.00_{0.98}$ | $0.00_{0.00}$ | 2.00 | $13.12_{1.23}$ | $11.30_{0.95}$ | 1.82 |
| | InternVL2* | 76B | $0.50_{0.50}$ | $1.00_{0.69}$ | -0.50 | $12.66_{1.04}$ | $11.88_{1.02}$ | 0.78 |
| | InternVL2.5* | 1B | $30.50_{3.29}$ | $20.50_{2.93}$ | 10.00 | $54.10_{2.63}$ | $42.89_{2.54}$ | 11.21 |
| | InternVL2.5* | 2B | $34.50_{3.40}$ | $16.00_{2.64}$ | 18.50 | $57.25_{2.57}$ | $39.81_{2.35}$ | 17.44 |
| | InternVL2.5* | 4B | $35.50_{3.31}$ | $17.00_{2.65}$ | 18.50 | $59.25_{2.65}$ | $41.38_{2.44}$ | 17.87 |
| | InternVL2.5* | 8B | $29.50_{3.29}$ | $19.00_{2.77}$ | 10.50 | $55.19_{2.43}$ | $40.27_{2.35}$ | 14.93 |
| | InternVL2.5* | 26B | $31.00_{3.33}$ | $19.50_{2.76}$ | 11.50 | $54.71_{2.54}$ | $42.13_{2.46}$ | 12.58 |
| | InternVL2.5* | 38B | $12.50_{2.36}$ | $7.00_{1.77}$ | 5.50 | $36.77_{2.20}$ | $28.40_{1.91}$ | 8.37 |
| | InternVL2.5* | 78B | $21.00_{2.91}$ | $11.50_{2.30}$ | 9.50 | $46.23_{2.52}$ | $34.80_{2.12}$ | 11.43 |
| | Llama-3.2-Vision* | 11B | $0.00_{0.00}$ | $0.00_{0.00}$ | 0.00 | $4.82_{0.50}$ | $7.23_{0.55}$ | -2.41 |
| | Llama-3.2-Vision* | 90B | $0.00_{0.00}$ | $0.00_{0.00}$ | 0.00 | $11.38_{0.81}$ | $9.77_{0.71}$ | 1.61 |
| | MiniCPM-V2.5 | 8B | $0.00_{0.00}$ | $0.50_{0.50}$ | -0.50 | $7.62_{0.62}$ | $5.00_{0.76}$ | 2.61 |
| | MiniCPM-V2.6* | 8B | $1.50_{0.85}$ | $1.50_{0.85}$ | 0.00 | $9.20_{1.13}$ | $9.97_{1.06}$ | -0.77 |
| | MiniMax-VL-01* | 456B | $2.00_{0.97}$ | $0.50_{0.50}$ | 1.50 | $16.12_{1.34}$ | $13.40_{1.04}$ | 2.72 |
| | Molmo* | 72B | $0.00_{0.00}$ | $0.00_{0.00}$ | 0.00 | $5.65_{0.54}$ | $4.52_{0.43}$ | 1.13 |
| | MolmoE* | 1B | $0.00_{0.00}$ | $0.00_{0.00}$ | 0.00 | $5.14_{0.45}$ | $4.13_{0.45}$ | 1.01 |
| | Molmo-D* | 7B | $0.00_{0.00}$ | $0.00_{0.00}$ | 0.00 | $3.02_{0.35}$ | $4.87_{0.41}$ | -1.85 |
| | Molmo-O* | 7B | $0.00_{0.00}$ | $0.00_{0.00}$ | 0.00 | $4.25_{0.44}$ | $4.09_{0.40}$ | 0.15 |
| | Ovis1.6-Gemma2* | 9B | $1.00_{0.72}$ | $0.50_{0.49}$ | 0.50 | $6.50_{0.90}$ | $8.95_{0.82}$ | -2.45 |
| | Ovis1.6-Gemma2* | 27B | $1.50_{0.87}$ | $1.00_{0.70}$ | 0.50 | $10.37_{1.08}$ | $9.51_{0.96}$ | 0.86 |
| | Ovis2* | 1B | $19.00_{2.88}$ | $15.00_{2.53}$ | 4.00 | $41.79_{2.50}$ | $35.21_{2.43}$ | 6.59 |
| | Ovis2* | 2B | $24.00_{2.98}$ | $15.00_{2.45}$ | 9.00 | $44.65_{2.63}$ | $36.37_{2.26}$ | 8.27 |
| | Ovis2* | 4B | $6.00_{1.66}$ | $4.50_{1.47}$ | 1.50 | $24.70_{1.90}$ | $20.47_{1.70}$ | 4.23 |
| | Ovis2* | 8B | $2.50_{1.07}$ | $0.50_{0.49}$ | 2.00 | $17.65_{1.51}$ | $13.04_{1.01}$ | 4.61 |
| | Ovis2* | 16B | $1.00_{0.70}$ | $1.00_{0.68}$ | 0.00 | $12.72_{1.11}$ | $12.24_{1.05}$ | 0.48 |
| | Ovis2* | 34B | $2.00_{1.00}$ | $1.00_{0.73}$ | 1.00 | $14.12_{1.39}$ | $13.49_{1.28}$ | 0.62 |
| | Phi-3.5-Vision* | 4B | $0.00_{0.00}$ | $0.00_{0.00}$ | 0.00 | $0.00_{0.00}$ | $0.11_{0.08}$ | -0.11 |
| | Phi-4-Multimodal* | 6B | $0.00_{0.00}$ | $0.00_{0.00}$ | 0.00 | $0.06_{0.06}$ | $0.36_{0.19}$ | -0.30 |
| | Pixtral* | 12B | $0.00_{0.00}$ | $0.00_{0.00}$ | 0.00 | $5.47_{0.44}$ | $5.65_{0.45}$ | -0.18 |
| | QvQ-Preview* | 72B | $18.50_{2.75}$ | $23.50_{3.01}$ | -5.00 | $24.71_{2.73}$ | $34.85_{2.90}$ | -10.13 |
| | Qwen-VL | 7B | $0.00_{0.00}$ | $0.00_{0.00}$ | 0.00 | $0.00_{0.00}$ | $0.00_{0.00}$ | 0.00 |
| | Qwen2-VL* | 2B | $30.00_{3.13}$ | $37.50_{3.41}$ | -7.50 | $47.72_{2.73}$ | $55.75_{2.78}$ | -8.04 |
| | Qwen2-VL* | 7B | $\mathbf{42.50_{3.29}}$ | $\mathbf{45.50_{3.52}}$ | -3.00 | $\mathbf{60.55_{2.80}}$ | $\mathbf{68.09_{2.42}}$ | -7.54 |
| | Qwen2-VL* | 72B | $38.00_{3.48}$ | $41.50_{3.46}$ | -3.50 | $58.81_{2.66}$ | $61.93_{2.60}$ | -3.12 |
| | Qwen2.5-VL* | 3B | $8.50_{2.00}$ | $10.00_{2.12}$ | -1.50 | $24.24_{2.17}$ | $30.77_{2.27}$ | -6.53 |
| | Qwen2.5-VL* | 7B | $17.00_{2.63}$ | $21.50_{2.88}$ | -4.50 | $32.48_{2.69}$ | $42.54_{2.63}$ | -10.06 |
| | Qwen2.5-VL* | 72B | $33.50_{3.35}$ | $28.00_{3.21}$ | 5.50 | $52.65_{2.78}$ | $46.83_{2.72}$ | 5.82 |

# D  ADDITIONAL EVALUATION RESULTS ON FIRST 100 AND 500 TEST CASES

Table 8: Results of various open-source and closed-source vision language models on the VCR task using the first 100 test cases. FT = fine-tuned on 16,000 samples from the **VCR-WIKI** training set. We label the best result of each setting and metric with **bold fonts**, and the best open-souce model with underline. Subscripts are standard deviations obtained from Bootstrap. For more latest models' results, please visit our GitHub repository.

| Language | Mode | Open/closed source | Model name | Model size | Exact match (%) ↑ $VI+TEI$ | Exact match (%) ↑ $TEI$ | Exact match (%) ↑ Δ | Jaccard index (%) ↑ $VI+TEI$ | Jaccard index (%) ↑ $TEI$ | Jaccard index (%) ↑ Δ |
|---|---|---|---|---|---|---|---|---|---|---|
| English | Easy | Closed | Claude 3 Opus | - | $62.0_{0.76}$ | $82.0_{0.63}$ | -20 | $78.06_{0.24}$ | $91.12_{0.13}$ | -13.06 |
| | | | Claude 3.5 Sonnet | - | $70.41_{3.46}$ | $75.15_{3.36}$ | -4.73 | $78.1_{2.85}$ | $86.5_{2.18}$ | -8.4 |
| | | | Gemini 1.5 Pro | - | $71.01_{3.4}$ | $86.98_{2.67}$ | -15.98 | $82.89_{2.27}$ | $94.21_{1.32}$ | -11.32 |
| | | | GPT-4 Turbo | - | $78.47_{0.22}$ | $86.6_{0.79}$ | -8.13 | $88.08_{5.25}$ | $94.15_{0.2}$ | -6.07 |
| | | | GPT-4o | - | $90.91_{0.36}$ | $95.69_{0.23}$ | -4.78 | $96.77_{0.16}$ | $98.45_{0.06}$ | -1.68 |
| | | | GPT-4V | - | $25.36_{0.5}$ | $18.18_{0.54}$ | 7.18 | $35.64_{0.22}$ | $28.49_{0.23}$ | 7.15 |
| | | | Qwen-VL-Max | - | $82.3_{0.19}$ | $88.04_{0.43}$ | -5.74 | $89.73_{0.32}$ | $92.55_{0.17}$ | -2.82 |
| | | | Reka Core | - | $65.68_{3.78}$ | $78.11_{3.19}$ | -12.43 | $83.14_{2.04}$ | $90.43_{1.49}$ | -7.29 |
| | | Open | Cambrian-1 | 34B | $78.11_{3.16}$ | $82.84_{2.86}$ | -4.73 | $87.88_{1.97}$ | $93.12_{1.26}$ | -5.24 |
| | | | CogVLM2 | 19B | $86.39_{0.66}$ | $84.62_{0.92}$ | 1.78 | $91.39_{0.11}$ | $91.63_{0.11}$ | -0.24 |
| | | | CogVLM2-FT | 19B | $94.08_{0.2}$ | $94.67_{0.26}$ | -0.59 | $98.03_{0.07}$ | $98.22_{0.03}$ | -0.2 |
| | | | DeepSeek-VL | 7B | $36.09_{1.36}$ | $44.97_{0.79}$ | -8.88 | $57.81_{0.18}$ | $61.83_{0.33}$ | -4.01 |
| | | | DeepSeek-VL2 | 28B | $56.21_{3.84}$ | $69.23_{3.58}$ | -13.02 | $68.35_{0.07}$ | $79.82_{2.55}$ | -11.46 |
| | | | DocOwl-1.5-Omni | 8B | $0.59_{0.14}$ | $1.18_{0.14}$ | -0.59 | $12.69_{0.04}$ | $13.3_{0.06}$ | -0.61 |
| | | | Idefics3 | 8B | $26.63_{3.35}$ | $32.54_{3.63}$ | -5.92 | $48.83_{2.95}$ | $55.11_{2.78}$ | -6.29 |
| | | | InternLM-XComposer2-VL | 7B | $47.93_{0.69}$ | $47.34_{0.57}$ | 0.59 | $73.88_{0.22}$ | $74.58_{0.16}$ | -0.7 |
| | | | InternLM-XComposer2-VL-4K | 7B | $4.11_{1.54}$ | $3.55_{1.49}$ | 0.59 | $21.91_{1.81}$ | $21.85_{1.86}$ | 0.06 |
| | | | InternLM-XComposer2.5-VL | 7B | $45.56_{3.83}$ | $28.99_{3.50}$ | 16.57 | $67.70_{2.79}$ | $54.25_{2.70}$ | 13.45 |
| | | | InternVL-V2 | 40B | $86.39_{2.56}$ | $86.98_{2.60}$ | -0.59 | $93.51_{1.40}$ | $94.35_{1.24}$ | -0.84 |
| | | | InternVL-V2 | 76B | $88.17_{2.48}$ | $92.31_{2.05}$ | -4.14 | $94.22_{1.37}$ | $97.04_{0.89}$ | -2.82 |
| | | | Llama-3.2 | 11B | $79.88_{2.96}$ | $68.64_{3.75}$ | 11.24 | $90.88_{1.56}$ | $82.91_{2.11}$ | 7.97 |
| | | | Llama-3.2 | 90B | $79.29_{3.14}$ | $71.01_{3.36}$ | 8.28 | $87.81_{2.00}$ | $83.17_{2.30}$ | 4.64 |
| | | | MiniCPM-V2.5 | 8B | $30.18_{0.66}$ | $36.09_{0.34}$ | -5.92 | $53.1_{0.18}$ | $59.06_{0.14}$ | -5.96 |
| | | | MiniCPM-V2.5-FT | 8B | $39.05_{0.69}$ | $46.75_{0.59}$ | -7.69 | $63.05_{0.28}$ | $69.89_{0.33}$ | -6.84 |
| | | | Monkey | 7B | $46.75_{0.44}$ | $48.52_{0.41}$ | -1.78 | $67.82_{0.22}$ | $68.59_{0.13}$ | -0.76 |
| | | | Ovis2 | 34B | $75.74_{3.35}$ | $71.01_{3.36}$ | 4.73 | $83.91_{2.39}$ | $86.46_{1.83}$ | -2.54 |
| | | | Pixtral | 34B | $14.79_{2.65}$ | $13.02_{2.62}$ | 1.78 | $39.00_{0.49}$ | $33.16_{2.53}$ | 5.84 |
| | | | Qwen-VL | 7B | $47.34_{0.44}$ | $46.75_{0.57}$ | 0.59 | $69.02_{0.35}$ | $69.19_{0.37}$ | -0.17 |
| | | | Qwen2-VL | 7B | $90.53_{2.25}$ | **$96.45_{1.39}$** | -5.92 | $94.28_{1.54}$ | **$98.82_{0.49}$** | -4.54 |
| | | | Qwen2-VL | 72B | $94.08_{1.80}$ | $95.27_{1.67}$ | -1.18 | $96.37_{1.23}$ | $97.49_{0.98}$ | -1.12 |
| | | | Qwen2.5-VL | 7B | **$95.86_{1.53}$** | $94.67_{1.76}$ | 1.18 | **$98.52_{0.55}$** | $96.51_{1.25}$ | 2.01 |
| | | | Qwen2.5-VL | 72B | $93.49_{1.91}$ | $90.53_{2.27}$ | 2.96 | $95.54_{1.41}$ | $91.80_{2.05}$ | 3.74 |
| | | | Yi-VL | 34B | $1.78_{0.16}$ | $1.18_{0.11}$ | 0.59 | $6.21_{0.06}$ | $7.5_{0.08}$ | -1.3 |
| | Hard | Closed | Claude 3 Opus | - | $34.0_{1.12}$ | $51.0_{0.5}$ | -17 | $57.02_{0.24}$ | $70.32_{0.15}$ | -13.31 |
| | | | Claude 3.5 Sonnet | - | $46.75_{3.58}$ | $43.2_{3.83}$ | 3.55 | $57.74_{3.33}$ | $54.13_{3.51}$ | 3.61 |
| | | | Gemini 1.5 Pro | - | $33.73_{3.69}$ | $43.79_{3.74}$ | -10.06 | $57.09_{2.67}$ | $62.34_{2.76}$ | -5.25 |
| | | | GPT-4 Turbo | - | $53.11_{0.46}$ | $57.42_{0.5}$ | -4.31 | $71.75_{0.19}$ | $73.82_{0.24}$ | -2.07 |
| | | | GPT-4o | - | $74.16_{0.31}$ | **$84.69_{0.31}$** | -10.53 | $86.99_{0.09}$ | **$93.19_{0.07}$** | -6.21 |
| | | | GPT-4V | - | $28.71_{0.49}$ | $16.27_{0.73}$ | 12.44 | $49.89_{0.15}$ | $33.64_{0.16}$ | 16.25 |
| | | | Qwen-VL-Max | - | $40.67_{0.38}$ | $55.02_{0.46}$ | -14.35 | $61.8_{0.19}$ | $72.46_{0.15}$ | -10.66 |
| | | | Reka Core | - | $7.1_{2.01}$ | $10.65_{2.38}$ | -3.55 | $25.49_{1.99}$ | $36.78_{2.19}$ | -11.29 |
| | | Open | Cambrian-1 | 34B | $27.81_{3.29}$ | $29.59_{3.54}$ | -1.78 | $51.39_{2.79}$ | $54.00_{2.76}$ | -2.61 |
| | | | CogVLM2 | 19B | $44.97_{0.83}$ | $21.3_{0.47}$ | 23.67 | $65.39_{0.2}$ | $43.86_{0.27}$ | 21.53 |
| | | | CogVLM2-FT | 19B | $75.74_{0.72}$ | $67.46_{0.64}$ | 8.28 | **$90.6_{0.13}$** | $84.26_{0.08}$ | 6.34 |
| | | | DeepSeek-VL | 7B | $0.59_{0.09}$ | $1.78_{0.17}$ | -1.18 | $16.71_{0.11}$ | $18.09_{0.13}$ | -1.38 |
| | | | DeepSeek-VL2 | 28B | $32.54_{3.57}$ | $42.60_{3.80}$ | -10.06 | $50.82_{3.04}$ | $58.42_{3.06}$ | -7.60 |
| | | | DocOwl-1.5-Omni | 8B | $0.0_{0.0}$ | $0.0_{0.0}$ | 0 | $7.89_{0.05}$ | $8.28_{0.05}$ | -0.4 |
| | | | Idefics3 | 8B | $1.18_{0.87}$ | $0.59_{0.61}$ | 0.59 | $11.62_{1.20}$ | $10.28_{1.00}$ | 1.34 |
| | | | InternLM-XComposer2-VL | 7B | $0.0_{0.0}$ | $0.59_{0.09}$ | -0.59 | $12.69_{0.08}$ | $14.05_{0.11}$ | -1.35 |
| | | | InternLM-XComposer2-VL-4K | 7B | $0.00_{0.00}$ | $0.59_{0.59}$ | -0.59 | $9.67_{0.90}$ | $8.83_{0.95}$ | 0.84 |
| | | | InternLM-XComposer2.5-VL | 7B | $0.59_{0.58}$ | $1.78_{1.01}$ | -1.18 | $14.09_{1.04}$ | $16.57_{1.25}$ | -2.48 |
| | | | InternVL-V2 | 40B | $12.43_{2.54}$ | $16.57_{2.89}$ | -4.14 | $33.74_{2.40}$ | $39.51_{2.69}$ | -5.76 |
| | | | InternVL-V2 | 76B | $22.34_{3.10}$ | $23.94_{3.19}$ | -1.60 | $46.64_{2.46}$ | $48.01_{2.51}$ | -1.36 |
| | | | Llama-3.2 | 11B | $10.65_{2.33}$ | $7.69_{2.04}$ | 2.96 | $33.50_{2.28}$ | $26.80_{2.03}$ | 6.70 |
| | | | Llama-3.2 | 90B | $13.02_{2.59}$ | $15.38_{2.75}$ | -2.37 | $36.80_{2.36}$ | $39.85_{2.52}$ | -3.06 |
| | | | MiniCPM-V2.5 | 8B | $1.18_{0.12}$ | $1.78_{0.12}$ | -0.59 | $12.02_{0.12}$ | $12.41_{0.07}$ | -0.39 |
| | | | MiniCPM-V2.5-FT | 8B | $10.06_{0.43}$ | $13.02_{0.54}$ | -2.96 | $34.67_{0.2}$ | $36.43_{0.19}$ | -1.76 |
| | | | Monkey | 7B | $1.18_{0.22}$ | $3.55_{0.18}$ | -2.37 | $12.66_{0.21}$ | $15.97_{0.08}$ | -3.31 |
| | | | Ovis2 | 34B | $74.56_{3.32}$ | $73.96_{3.30}$ | 0.59 | $86.86_{2.00}$ | $90.12_{1.38}$ | -3.25 |
| | | | Pixtral | 12B | $0.00_{0.00}$ | $0.59_{0.61}$ | -0.59 | $9.90_{0.79}$ | $12.56_{1.09}$ | -2.66 |
| | | | Qwen-VL | 7B | $1.78_{0.21}$ | $2.96_{0.12}$ | -1.18 | $15.7_{0.14}$ | $15.06_{0.19}$ | 0.63 |
| | | | Qwen2-VL | 7B | $75.74_{4.32}$ | $73.96_{3.55}$ | 1.78 | $85.91_{2.17}$ | $85.83_{2.04}$ | 0.08 |
| | | | Qwen2-VL | 72B | $71.60_{3.61}$ | $72.19_{3.60}$ | -0.59 | $84.52_{2.18}$ | $86.17_{1.89}$ | -1.64 |
| | | | Qwen2.5-VL | 7B | **$82.84_{2.91}$** | $75.74_{3.30}$ | 7.10 | $93.38_{1.21}$ | $88.75_{1.73}$ | 4.64 |
| | | | Qwen2.5-VL | 72B | $81.07_{2.98}$ | $74.56_{3.29}$ | 6.51 | $89.03_{2.01}$ | $83.21_{2.52}$ | 5.82 |
| | | | Yi-VL | 34B | $0.59_{0.09}$ | $0.0_{0.0}$ | 0.59 | $4.39_{0.07}$ | $5.49_{0.08}$ | -1.1 |
| Chinese | Easy | Closed | Claude 3 Opus | - | $0.53_{0.51}$ | $0.53_{0.55}$ | 0 | $11.34_{1.07}$ | $9.14_{0.93}$ | 2.2 |
| | | | Claude 3.5 Sonnet | - | $1.6_{0.91}$ | $2.13_{1.05}$ | -0.53 | $8.07_{1.29}$ | $9.9_{1.48}$ | -1.84 |
| | | | Gemini 1.5 Pro | - | $0.53_{0.56}$ | $0.0_{0.0}$ | 0.53 | $12.94_{1.26}$ | $12.77_{1.17}$ | 0.16 |
| | | | GPT-4o | - | $14.89_{2.51}$ | $21.81_{2.98}$ | -6.91 | $38.57_{2.46}$ | $48.29_{2.43}$ | -9.72 |
| | | | GPT-4 Turbo | - | $0.53_{0.55}$ | $0.0_{0.0}$ | 0.53 | $11.09_{1.05}$ | $7.51_{0.65}$ | 3.58 |
| | | | Qwen-VL-Max | - | $5.93_{0.19}$ | $8.7_{0.37}$ | -2.77 | $13.53_{0.11}$ | $18.5_{0.1}$ | -4.97 |
| | | | Reka Core | - | $0.0_{0.0}$ | $0.0_{0.0}$ | 0 | $3.04_{0.53}$ | $2.42_{0.45}$ | 0.61 |
| | | Open | CogVLM2-Chinese | 19B | $34.57_{0.66}$ | $34.04_{1.01}$ | 0.53 | $58.78_{0.13}$ | $57.26_{0.12}$ | 1.52 |
| | | | CogVLM2-Chinese-FT | 19B | $66.49_{0.74}$ | $67.55_{0.73}$ | -1.06 | $79.48_{0.17}$ | $81.78_{0.09}$ | -2.3 |
| | | | DeepSeek-VL | 7B | $0.0_{0.0}$ | $0.0_{0.0}$ | 0 | $3.99_{0.07}$ | $6.71_{0.02}$ | -2.72 |
| | | | DeepSeek-VL2 | 28B | $5.85_{1.71}$ | $3.72_{1.37}$ | 2.13 | $12.57_{1.86}$ | $11.37_{1.59}$ | 1.19 |
| | | | DocOwl-1.5-Omni | 8B | $0.0_{0.0}$ | $0.0_{0.0}$ | 0 | $1.23_{0.04}$ | $2.97_{0.02}$ | -1.75 |
| | | | InternLM-XComposer2-VL | 7B | $1.06_{0.09}$ | $0.53_{0.07}$ | 0.53 | $13.1_{0.03}$ | $13.26_{0.03}$ | -0.16 |
| | | | InternLM-XComposer2-VL-4K | 7B | $0.00_{0.00}$ | $0.00_{0.00}$ | 0.00 | $13.49_{1.01}$ | $14.56_{0.98}$ | -1.08 |
| | | | InternLM-XComposer2.5-VL | 7B | $0.00_{0.00}$ | $1.60_{0.91}$ | -1.60 | $11.94_{0.88}$ | $16.12_{1.24}$ | -4.18 |
| | | | InternVL-V2 | 40B | $26.06_{3.17}$ | $19.15_{2.88}$ | 6.91 | $48.98_{2.61}$ | $41.25_{2.57}$ | 7.72 |
| | | | InternVL-V2 | 76B | $17.16_{2.80}$ | $19.53_{3.13}$ | -2.37 | $40.71_{2.47}$ | $44.86_{2.60}$ | -4.15 |
| | | | MiniCPM-V2.5 | 8B | $4.79_{0.16}$ | $7.45_{0.35}$ | -2.66 | $20.58_{0.11}$ | $25.38_{0.13}$ | -4.81 |
| | | | MiniCPM-V2.5-FT | 8B | $6.91_{0.33}$ | $7.98_{0.4}$ | -1.06 | $30.8_{0.07}$ | $31.46_{0.52}$ | -0.66 |
| | | | Monkey | 7B | $1.06_{0.12}$ | $0.53_{0.06}$ | 0.53 | $9.23_{0.08}$ | $12.29_{0.13}$ | -3.06 |
| | | | Ovis2 | 34B | $23.40_{3.13}$ | $17.55_{2.72}$ | 5.85 | $41.17_{2.84}$ | $36.41_{2.62}$ | 4.76 |
| | | | Qwen-VL | 7B | $0.0_{0.0}$ | $0.0_{0.0}$ | 0 | $1.41_{0.02}$ | $0.66_{0.03}$ | 0.76 |
| | | | Qwen2-VL | 7B | $67.55_{3.34}$ | $73.40_{3.21}$ | -5.85 | $84.63_{1.80}$ | $87.12_{1.76}$ | -2.49 |
| | | | Qwen2-VL | 72B | $70.74_{3.46}$ | $78.72_{3.00}$ | -7.98 | $85.14_{1.87}$ | $90.41_{1.40}$ | -5.26 |
| | | | Qwen2.5-VL | 7B | $73.40_{3.32}$ | $53.19_{3.60}$ | 20.21 | $86.14_{1.81}$ | $62.55_{3.20}$ | 23.58 |
| | | | Qwen2.5-VL | 72B | **$77.13_{3.02}$** | **$81.91_{2.87}$** | -4.79 | **$88.81_{1.65}$** | **$90.67_{1.54}$** | -1.87 |
| | | | Yi-VL | 34B | $0.0_{0.0}$ | $0.0_{0.0}$ | 0 | $4.53_{0.03}$ | $1.84_{0.05}$ | 2.69 |
| | Hard | Closed | Claude 3 Opus | - | $1.06_{0.77}$ | $0.53_{0.54}$ | 0.53 | $9.23_{1.04}$ | $7.77_{0.83}$ | 1.45 |
| | | | Claude 3.5 Sonnet | - | $0.53_{0.51}$ | $0.0_{0.0}$ | 0.53 | $4.11_{0.84}$ | $3.32_{0.71}$ | 0.79 |
| | | | Gemini 1.5 Pro | - | $1.06_{0.71}$ | $1.06_{0.77}$ | 0 | $11.58_{1.14}$ | $13.34_{1.2}$ | -1.76 |
| | | | GPT-4o | - | $2.66_{1.16}$ | $1.6_{0.92}$ | 1.06 | $23.69_{1.65}$ | $23.69_{1.48}$ | 0 |
| | | | GPT-4 Turbo | - | $0.0_{0.0}$ | $0.53_{0.53}$ | -0.53 | $8.51_{0.7}$ | $8.02_{0.58}$ | 0.49 |
| | | | Qwen-VL-Max | - | $1.19_{0.12}$ | $1.98_{0.09}$ | -0.79 | $6.19_{0.1}$ | $11.09_{0.11}$ | -4.9 |
| | | | Reka Core | - | $0.0_{0.0}$ | $0.0_{0.0}$ | 0 | $3.22_{0.51}$ | $3.62_{0.57}$ | -0.4 |
| | | Open | CogVLM2-Chinese | 19B | $3.19_{0.19}$ | $3.19_{0.32}$ | 0 | $18.33_{0.14}$ | $21.38_{0.09}$ | -3.05 |
| | | | CogVLM2-Chinese-FT | 19B | **$46.81_{0.32}$** | **$46.28_{0.49}$** | 0.53 | **$66.85_{0.39}$** | **$69.79_{0.12}$** | -2.95 |
| | | | DeepSeek-VL | 7B | $0.0_{0.0}$ | $0.0_{0.0}$ | 0 | $5.22_{0.04}$ | $7.45_{0.06}$ | -2.23 |
| | | | DeepSeek-VL2 | 28B | $0.00_{0.00}$ | $0.53_{0.52}$ | -0.53 | $4.78_{0.68}$ | $7.67_{0.82}$ | -2.89 |
| | | | DocOwl-1.5-Omni | 8B | $0.0_{0.0}$ | $0.0_{0.0}$ | 0 | $1.35_{0.02}$ | $3.57_{0.04}$ | -2.23 |
| | | | InternLM-XComposer2-VL | 7B | $0.0_{0.0}$ | $0.0_{0.0}$ | 0 | $8.17_{0.03}$ | $7.99_{0.03}$ | 0.18 |
| | | | InternLM-XComposer2-VL-4K | 7B | $0.00_{0.00}$ | $0.00_{0.00}$ | 0.00 | $7.73_{0.68}$ | $8.12_{0.79}$ | -0.39 |
| | | | InternLM-XComposer2.5-VL | 7B | $0.00_{0.00}$ | $0.00_{0.00}$ | 0.00 | $10.87_{0.82}$ | $10.54_{0.84}$ | 0.32 |
| | | | InternVL-V2 | 40B | $0.53_{0.50}$ | $1.06_{0.72}$ | -0.53 | $12.26_{1.01}$ | $13.58_{1.20}$ | -1.32 |
| | | | InternVL-V2 | 76B | $0.53_{0.53}$ | $0.53_{0.53}$ | 0.00 | $14.58_{0.96}$ | $14.32_{1.12}$ | 0.26 |
| | | | MiniCPM-V2.5 | 8B | $0.53_{0.07}$ | $0.53_{0.07}$ | 0 | $7.28_{0.06}$ | $7.71_{0.06}$ | -0.43 |
| | | | MiniCPM-V2.5-FT | 8B | $1.06_{0.08}$ | $2.13_{0.19}$ | -1.06 | $18.46_{0.1}$ | $16.42_{0.22}$ | 2.03 |
| | | | Monkey | 7B | $0.0_{0.0}$ | $0.0_{0.0}$ | 0 | $6.15_{0.11}$ | $6.62_{0.11}$ | -0.47 |
| | | | Ovis2 | 34B | $4.79_{1.54}$ | $3.72_{1.39}$ | 1.06 | $23.02_{1.93}$ | $21.14_{1.83}$ | 1.88 |
| | | | Qwen-VL | 7B | $0.0_{0.0}$ | $0.0_{0.0}$ | 0 | $1.1_{0.04}$ | $0.06_{0.01}$ | 1.04 |
| | | | Qwen2-VL | 7B | $17.55_{2.81}$ | $27.66_{3.21}$ | -10.11 | $43.87_{2.48}$ | $51.99_{2.61}$ | -8.12 |
| | | | Qwen2-VL | 72B | $15.96_{2.69}$ | $25.00_{3.29}$ | -9.04 | $39.42_{2.38}$ | $52.40_{2.49}$ | -12.98 |
| | | | Qwen2.5-VL | 7B | $19.15_{2.90}$ | $13.30_{2.37}$ | 5.85 | $44.06_{2.52}$ | $30.95_{2.59}$ | 13.10 |
| | | | Qwen2.5-VL | 72B | $30.85_{3.32}$ | $30.85_{3.46}$ | 0.00 | $57.64_{2.52}$ | $57.58_{2.52}$ | 0.07 |
| | | | Yi-VL | 34B | $0.0_{0.0}$ | $0.0_{0.0}$ | 0 | $4.17_{0.04}$ | $2.02_{0.04}$ | 2.15 |

Table 9: Results of various open-source and closed-source vision language models on the VCR task using the first 500 test cases. FT = fine-tuned on 16,000 samples from the **VCR-WIKI** training set. We label the best result of each setting and metric with **bold fonts**, and the best open-souce model with underline. Subscripts are standard deviations obtained from Bootstrap. For more latest models' results, please visit our GitHub repository.

| Language | Mode | Open/closed source | Model name | Model size | Exact match (%) ↑ | | | Jaccard index (%) ↑ | | |
|---|---|---|---|---|---|---|---|---|---|---|
| | | | | | $VI+TEI$ | $TEI$ | Δ | $VI+TEI$ | $TEI$ | Δ |
| English | Easy | Closed | Claude 3 Opus | - | $62.0_{0.13}$ | $77.0_{0.5}$ | -15 | $77.67_{0.32}$ | $88.41_{0.39}$ | -10.74 |
| | | | Claude 3.5 Sonnet | - | $63.85_{1.71}$ | $72.8_{1.56}$ | -8.94 | $74.65_{1.33}$ | $83.48_{1.14}$ | -8.83 |
| | | | Gemini 1.5 Pro | - | $62.73_{1.66}$ | $82.98_{1.3}$ | -20.25 | $77.71_{1.21}$ | $91.56_{0.76}$ | -13.85 |
| | | | GPT-4 Turbo | - | $78.74_{0.13}$ | $81.94_{0.25}$ | -3.2 | $88.54_{0.24}$ | $92.18_{0.3}$ | -3.65 |
| | | | GPT-4o | - | $91.55_{0.29}$ | $94.56_{0.13}$ | -3.01 | $96.44_{0.11}$ | $97.76_{0.06}$ | -1.32 |
| | | | GPT-4V | - | $52.04_{0.24}$ | $37.86_{0.22}$ | 14.17 | $65.36_{0.39}$ | $54.13_{0.41}$ | 11.23 |
| | | | Qwen-VL-Max | - | $76.8_{0.5}$ | $85.53_{0.19}$ | -8.74 | $85.71_{0.28}$ | $91.45_{0.29}$ | -5.74 |
| | | | Reka Core | - | $66.46_{1.64}$ | $78.51_{1.42}$ | -12.05 | $84.23_{0.86}$ | $90.45_{0.7}$ | -6.22 |
| | | Open | Cambrian-1 | 34B | $76.89_{1.52}$ | $80.25_{1.36}$ | -3.35 | $87.66_{0.90}$ | $92.42_{0.60}$ | -4.76 |
| | | | CogVLM2 | 19B | $83.11_{0.28}$ | $79.63_{0.33}$ | 3.48 | $89.43_{0.27}$ | $88.65_{0.26}$ | 0.79 |
| | | | CogVLM2-FT | 19B | $92.8_{0.06}$ | $92.67_{0.13}$ | 0.12 | $97.51_{0.24}$ | $97.45_{0.07}$ | 0.06 |
| | | | DeepSeek-VL | 7B | $37.76_{0.42}$ | $45.47_{0.21}$ | -7.7 | $59.07_{0.43}$ | $64.26_{0.57}$ | -5.2 |
| | | | DeepSeek-VL2 | 28B | $41.37_{1.73}$ | $52.92_{1.72}$ | -11.55 | $51.29_{1.59}$ | $60.74_{1.65}$ | -9.44 |
| | | | DocOwl-1.5-Omni | 8B | $0.62_{0.06}$ | $1.86_{0.06}$ | -1.24 | $12.65_{0.3}$ | $14.09_{0.12}$ | -1.44 |
| | | | Idefics3 | 8B | $26.71_{1.57}$ | $29.81_{1.55}$ | -3.11 | $46.91_{1.40}$ | $51.84_{1.30}$ | -4.93 |
| | | | InternLM-XComposer2-VL | 7B | $46.09_{0.35}$ | $46.34_{0.25}$ | -0.25 | $71.11_{0.2}$ | $71.76_{0.67}$ | -0.65 |
| | | | InternLM-XComposer2-VL-4K | 7B | $5.22_{0.80}$ | $3.23_{0.63}$ | 1.99 | $22.70_{0.89}$ | $18.67_{0.79}$ | 4.03 |
| | | | InternLM-XComposer2.5-VL | 7B | $42.48_{1.73}$ | $25.84_{1.53}$ | 16.65 | $63.03_{1.32}$ | $50.75_{1.21}$ | 12.28 |
| | | | InternVL-V2 | 40B | $84.84_{1.21}$ | $87.08_{1.19}$ | -2.24 | $93.13_{0.09}$ | $94.83_{0.50}$ | -1.71 |
| | | | InternVL-V2 | 76B | $81.24_{1.40}$ | $90.06_{1.06}$ | -8.82 | $90.64_{0.77}$ | $96.06_{0.46}$ | -5.42 |
| | | | Llama-3.2 | 11B | $79.25_{1.40}$ | $66.46_{1.63}$ | 12.80 | $89.98_{0.77}$ | $80.91_{1.06}$ | 9.08 |
| | | | Llama-3.2 | 90B | $80.87_{1.37}$ | $71.55_{1.54}$ | 9.32 | $89.63_{0.85}$ | $84.34_{0.98}$ | 5.29 |
| | | | MiniCPM-V2.5 | 8B | $32.8_{0.16}$ | $36.77_{0.25}$ | -3.98 | $52.56_{0.25}$ | $60.89_{0.19}$ | -8.32 |
| | | | MiniCPM-V2.5-FT | 8B | $42.36_{0.3}$ | $45.34_{0.35}$ | -2.98 | $65.39_{0.6}$ | $67.85_{0.43}$ | -2.46 |
| | | | Monkey | 7B | $47.2_{0.2}$ | $54.16_{0.41}$ | -6.96 | $65.7_{0.4}$ | $71.17_{0.72}$ | -5.47 |
| | | | Ovis2 | 34B | $73.91_{1.52}$ | $69.94_{1.59}$ | 3.98 | $83.41_{1.08}$ | $84.99_{0.95}$ | -1.58 |
| | | | Pixtral | 12B | $16.65_{1.31}$ | $11.80_{1.13}$ | 4.84 | $39.81_{1.16}$ | $31.47_{1.11}$ | 8.34 |
| | | | Qwen-VL | 7B | $45.47_{0.35}$ | $52.17_{0.33}$ | -6.71 | $66.81_{0.74}$ | $71.73_{0.59}$ | -4.93 |
| | | | Qwen2-VL | 7B | $90.06_{1.07}$ | $94.53_{0.82}$ | -4.47 | $93.77_{0.76}$ | **$97.80_{0.34}$** | -4.03 |
| | | | Qwen2-VL | 72B | $92.42_{0.92}$ | $94.66_{0.78}$ | -2.24 | $94.77_{0.72}$ | $97.48_{0.42}$ | -2.71 |
| | | | Qwen2.5-VL | 7B | **$94.91_{0.79}$** | **$93.79_{0.85}$** | 1.12 | **$98.17_{0.29}$** | $96.95_{0.47}$ | 1.22 |
| | | | Qwen2.5-VL | 72B | $92.67_{0.91}$ | $90.19_{1.08}$ | 2.48 | $95.55_{0.65}$ | $92.52_{0.86}$ | 3.02 |
| | | | Yi-VL | 34B | $0.87_{0.06}$ | $1.24_{0.04}$ | -0.37 | $5.61_{0.28}$ | $7.63_{0.42}$ | -2.02 |
| | Hard | Closed | Claude 3 Opus | - | $37.8_{0.28}$ | $50.0_{0.33}$ | -12.2 | $57.68_{0.8}$ | $70.16_{0.64}$ | -12.48 |
| | | | Claude 3.5 Sonnet | - | $41.74_{1.69}$ | $44.72_{1.78}$ | -2.98 | $56.15_{1.46}$ | $58.54_{1.6}$ | -2.4 |
| | | | Gemini 1.5 Pro | - | $28.07_{1.58}$ | $38.76_{1.68}$ | -10.68 | $51.9_{1.22}$ | $59.62_{1.27}$ | -7.72 |
| | | | GPT-4 Turbo | - | $45.15_{0.28}$ | $48.64_{0.57}$ | -3.5 | $65.72_{0.25}$ | $67.86_{0.2}$ | -2.14 |
| | | | GPT-4o | - | $73.2_{0.16}$ | **$82.43_{0.17}$** | -9.22 | $86.17_{0.21}$ | **$92.01_{0.2}$** | -5.84 |
| | | | GPT-4V | - | $25.83_{0.44}$ | $14.95_{0.3}$ | 10.87 | $44.63_{0.48}$ | $30.08_{0.67}$ | 14.56 |
| | | | Qwen-VL-Max | - | $41.65_{0.32}$ | $52.72_{0.2}$ | -11.07 | $61.18_{0.35}$ | $70.19_{0.37}$ | -9.01 |
| | | | Reka Core | - | $6.71_{0.89}$ | $11.18_{1.15}$ | -4.47 | $25.84_{0.95}$ | $35.83_{1.05}$ | -9.99 |
| | | Open | Cambrian-1 | 34B | $27.20_{1.59}$ | $30.19_{1.55}$ | -2.98 | $49.96_{1.36}$ | $55.93_{1.23}$ | -5.97 |
| | | | CogVLM2 | 19B | $41.74_{0.25}$ | $16.77_{0.22}$ | 24.97 | $62.56_{0.33}$ | $38.41_{0.44}$ | 24.15 |
| | | | CogVLM2-FT | 19B | $75.9_{0.13}$ | $65.22_{0.18}$ | 10.68 | $89.75_{0.14}$ | $82.71_{0.27}$ | 7.04 |
| | | | DeepSeek-VL | 7B | $0.75_{0.02}$ | $1.61_{0.01}$ | -0.87 | $15.8_{0.29}$ | $17.18_{0.41}$ | -1.38 |
| | | | DeepSeek-VL2 | 28B | $26.96_{1.58}$ | $36.02_{1.70}$ | -9.07 | $47.49_{1.42}$ | $56.79_{1.32}$ | -9.30 |
| | | | DocOwl-1.5-Omni | 8B | $0.0_{0.0}$ | $0.0_{0.0}$ | 0 | $7.34_{0.06}$ | $7.61_{0.16}$ | -0.27 |
| | | | Idefics3 | 8B | $0.75_{0.30}$ | $0.50_{0.25}$ | 0.25 | $10.44_{0.49}$ | $9.17_{0.43}$ | 1.27 |
| | | | InternLM-XComposer2-VL | 7B | $0.5_{0.04}$ | $0.37_{0.05}$ | 0.12 | $12.38_{0.13}$ | $13.22_{0.11}$ | -0.83 |
| | | | InternLM-XComposer2-VL-4K | 7B | $0.00_{0.00}$ | $0.12_{0.12}$ | -0.12 | $9.55_{0.38}$ | $9.18_{0.38}$ | 0.37 |
| | | | InternLM-XComposer2.5-VL | 7B | $0.75_{0.31}$ | $1.24_{0.39}$ | -0.50 | $13.67_{0.51}$ | $14.92_{0.56}$ | -1.25 |
| | | | InternVL-V2 | 40B | $14.16_{1.22}$ | $18.51_{1.36}$ | -4.35 | $35.01_{1.16}$ | $41.02_{1.22}$ | -6.02 |
| | | | InternVL-V2 | 76B | $20.76_{1.29}$ | $19.36_{1.24}$ | 1.40 | $45.41_{1.14}$ | $43.65_{1.10}$ | 1.76 |
| | | | Llama-3.2 | 11B | $13.91_{1.25}$ | $7.33_{0.94}$ | 6.58 | $35.78_{1.11}$ | $26.14_{0.94}$ | 9.64 |
| | | | Llama-3.2 | 90B | $15.16_{1.28}$ | $12.17_{1.13}$ | 2.98 | $37.57_{1.13}$ | $35.14_{1.04}$ | 2.43 |
| | | | MiniCPM-V2.5 | 8B | $1.74_{0.08}$ | $1.61_{0.08}$ | 0.12 | $11.55_{0.12}$ | $11.69_{0.18}$ | -0.15 |
| | | | MiniCPM-V2.5-FT | 8B | $11.43_{0.11}$ | $14.29_{0.16}$ | -2.86 | $35.13_{0.19}$ | $36.65_{0.68}$ | -1.52 |
| | | | Monkey | 7B | $1.37_{0.05}$ | $2.24_{0.15}$ | -0.87 | $13.16_{0.18}$ | $14.45_{0.24}$ | -1.29 |
| | | | Ovis2 | 34B | $77.14_{1.44}$ | $70.31_{1.65}$ | 6.83 | $87.97_{0.92}$ | $87.80_{0.75}$ | 0.17 |
| | | | Pixtral | 12B | $0.25_{0.19}$ | $0.62_{0.28}$ | -0.37 | $10.04_{0.41}$ | $11.21_{0.45}$ | -1.17 |
| | | | Qwen-VL | 7B | $1.61_{0.03}$ | $1.74_{0.03}$ | -0.12 | $15.28_{0.13}$ | $14.43_{0.54}$ | 0.85 |
| | | | Qwen2-VL | 7B | $76.27_{1.49}$ | $75.65_{1.45}$ | 0.62 | $86.56_{1.00}$ | $86.77_{0.93}$ | -0.22 |
| | | | Qwen2-VL | 72B | $70.56_{1.68}$ | $73.42_{1.53}$ | -2.86 | $82.94_{1.04}$ | $86.69_{0.87}$ | -3.74 |
| | | | Qwen2.5-VL | 7B | $80.12_{1.38}$ | $74.04_{1.51}$ | 6.09 | **$91.68_{0.66}$** | $87.85_{0.81}$ | 3.83 |
| | | | Qwen2.5-VL | 72B | $80.50_{1.41}$ | $70.06_{1.62}$ | 10.43 | $88.91_{0.88}$ | $79.05_{1.25}$ | 9.86 |
| | | | Yi-VL | 34B | $0.12_{0.01}$ | $0.0_{0.0}$ | 0.12 | $4.31_{0.08}$ | $5.45_{0.13}$ | -1.14 |
| Chinese | Easy | Closed | Claude 3 Opus | - | $0.9_{0.3}$ | $1.0_{0.31}$ | -0.1 | $11.5_{0.49}$ | $10.0_{0.49}$ | 1.49 |
| | | | Claude 3.5 Sonnet | - | $1.0_{0.31}$ | $0.8_{0.28}$ | 0.2 | $7.54_{0.54}$ | $7.5_{0.51}$ | 0.03 |
| | | | Gemini 1.5 Pro | - | $1.1_{0.32}$ | $0.5_{0.22}$ | 0.6 | $11.1_{0.36}$ | $11.47_{0.48}$ | -0.37 |
| | | | GPT-4o | - | $14.87_{1.14}$ | $22.46_{1.35}$ | -7.58 | $39.05_{0.99}$ | $48.24_{1.09}$ | -9.19 |
| | | | GPT-4 Turbo | - | $0.2_{0.14}$ | $0.1_{0.1}$ | 0.1 | $8.42_{0.36}$ | $6.97_{0.29}$ | 1.45 |
| | | | Qwen-VL-Max | - | $6.34_{0.08}$ | $9.92_{0.09}$ | -3.58 | $13.45_{0.41}$ | $22.86_{0.46}$ | -9.42 |
| | | | Reka Core | - | $0.0_{0.0}$ | $0.0_{0.0}$ | 0 | $3.43_{0.26}$ | $3.15_{0.2}$ | 0.28 |
| | | Open | CogVLM2-Chinese | 19B | $33.63_{0.15}$ | $31.44_{0.19}$ | 2.2 | $57.97_{0.56}$ | $54.05_{0.54}$ | 3.92 |
| | | | CogVLM2-Chinese-FT | 19B | $63.97_{0.55}$ | $62.67_{0.17}$ | 1.3 | $79.71_{0.41}$ | $79.22_{0.47}$ | 0.49 |
| | | | DeepSeek-VL | 7B | $0.0_{0.0}$ | $0.0_{0.0}$ | 0 | $4.28_{0.07}$ | $7.3_{0.05}$ | -3.02 |
| | | | DeepSeek-VL2 | 28B | $3.79_{0.59}$ | $2.20_{0.46}$ | 1.60 | $10.13_{0.67}$ | $10.27_{0.61}$ | -0.14 |
| | | | DocOwl-1.5-Omni | 8B | $0.0_{0.0}$ | $0.0_{0.0}$ | 0 | $1.19_{0.05}$ | $3.83_{0.06}$ | -2.63 |
| | | | InternLM-XComposer2-VL | 7B | $0.6_{0.05}$ | $0.2_{0.04}$ | 0.4 | $12.34_{0.25}$ | $12.52_{0.14}$ | -0.18 |
| | | | InternLM-XComposer2-VL-4K | 7B | $0.20_{0.14}$ | $0.10_{0.10}$ | 0.10 | $11.93_{0.42}$ | $13.68_{0.41}$ | -1.74 |
| | | | InternLM-XComposer2.5-VL | 7B | $0.30_{0.17}$ | $0.40_{0.20}$ | -0.10 | $12.76_{0.42}$ | $14.99_{0.43}$ | -2.23 |
| | | | InternVL-V2 | 40B | $22.75_{1.36}$ | $16.67_{1.14}$ | 6.09 | $49.51_{1.06}$ | $39.46_{1.10}$ | 10.05 |
| | | | InternVL-V2 | 76B | $19.50_{1.37}$ | $22.48_{1.44}$ | -2.98 | $42.21_{1.15}$ | $45.74_{1.21}$ | -3.53 |
| | | | MiniCPM-V2.5 | 8B | $4.59_{0.14}$ | $4.89_{0.09}$ | -0.3 | $18.12_{0.33}$ | $22.28_{0.18}$ | -4.17 |
| | | | MiniCPM-V2.5-FT | 8B | $7.29_{0.14}$ | $7.09_{0.12}$ | 0.2 | $29.36_{0.39}$ | $30.67_{0.38}$ | -1.31 |
| | | | Monkey | 7B | $0.2_{0.01}$ | $1.4_{0.05}$ | -1.2 | $7.89_{0.3}$ | $10.26_{0.24}$ | -2.37 |
| | | | Ovis2 | 34B | $23.35_{1.30}$ | $15.57_{1.19}$ | 7.78 | $41.31_{1.19}$ | $35.79_{1.14}$ | 5.51 |
| | | | Qwen-VL | 7B | $0.0_{0.0}$ | $0.0_{0.0}$ | 0 | $1.25_{0.03}$ | $0.43_{0.06}$ | 0.82 |
| | | | Qwen2-VL | 7B | $61.08_{1.52}$ | $68.16_{1.46}$ | -7.09 | $78.38_{0.96}$ | $83.48_{0.84}$ | -5.10 |
| | | | Qwen2-VL | 72B | $66.47_{1.50}$ | $74.75_{1.40}$ | -8.28 | $81.70_{0.88}$ | $87.35_{0.75}$ | -5.66 |
| | | | Qwen2.5-VL | 7B | $72.16_{1.44}$ | $47.21_{1.59}$ | 24.95 | $84.40_{0.83}$ | $57.99_{1.34}$ | 26.41 |
| | | | Qwen2.5-VL | 72B | $75.15_{1.41}$ | $74.05_{1.37}$ | 1.10 | $86.54_{0.82}$ | $84.29_{0.94}$ | 2.25 |
| | | | Yi-VL | 34B | $0.0_{0.0}$ | $0.0_{0.0}$ | 0 | $4.69_{0.09}$ | $1.71_{0.06}$ | 2.98 |
| | Hard | Closed | Claude 3 Opus | - | $0.3_{0.18}$ | $0.1_{0.1}$ | 0.2 | $9.22_{0.38}$ | $8.09_{0.33}$ | 1.13 |
| | | | Claude 3.5 Sonnet | - | $0.2_{0.15}$ | $0.0_{0.0}$ | 0.2 | $4.0_{0.33}$ | $2.37_{0.23}$ | 1.63 |
| | | | Gemini 1.5 Pro | - | $0.7_{0.26}$ | $0.5_{0.23}$ | 0.2 | $11.82_{0.51}$ | $11.75_{0.44}$ | 0.07 |
| | | | GPT-4o | - | $2.2_{0.47}$ | $1.8_{0.4}$ | 0.4 | $22.72_{0.67}$ | $22.89_{0.65}$ | -0.17 |
| | | | GPT-4 Turbo | - | $0.0_{0.0}$ | $0.2_{0.13}$ | -0.2 | $8.58_{0.3}$ | $6.87_{0.28}$ | 1.72 |
| | | | Qwen-VL-Max | - | $0.89_{0.06}$ | $1.38_{0.1}$ | -0.49 | $5.4_{0.19}$ | $12.29_{0.14}$ | -6.89 |
| | | | Reka Core | - | $0.0_{0.0}$ | $0.0_{0.0}$ | 0 | $3.35_{0.23}$ | $2.97_{0.2}$ | 0.38 |
| | | Open | CogVLM2-Chinese | 19B | $1.2_{0.07}$ | $2.3_{0.09}$ | -1.1 | $16.83_{0.22}$ | $19.86_{0.23}$ | -3.04 |
| | | | CogVLM2-Chinese-FT | 19B | **$42.51_{0.32}$** | **$45.91_{0.23}$** | -3.39 | **$65.79_{0.24}$** | **$69.46_{0.46}$** | -3.68 |
| | | | DeepSeek-VL | 7B | $0.0_{0.0}$ | $0.0_{0.0}$ | 0 | $5.49_{0.07}$ | $7.57_{0.05}$ | -2.08 |
| | | | DeepSeek-VL2 | 28B | $0.00_{0.00}$ | $0.20_{0.14}$ | -0.20 | $4.45_{0.27}$ | $6.51_{0.29}$ | -2.06 |
| | | | DocOwl-1.5-Omni | 8B | $0.0_{0.0}$ | $0.0_{0.0}$ | 0 | $1.68_{0.04}$ | $4.42_{0.07}$ | -2.73 |
| | | | InternLM-XComposer2-VL | 7B | $0.0_{0.0}$ | $0.0_{0.0}$ | 0 | $8.36_{0.04}$ | $7.92_{0.09}$ | 0.44 |
| | | | InternLM-XComposer2-VL-4K | 7B | $0.00_{0.00}$ | $0.00_{0.00}$ | 0.00 | $7.49_{0.31}$ | $7.25_{0.30}$ | 0.25 |
| | | | InternLM-XComposer2.5-VL | 7B | $0.00_{0.00}$ | $0.00_{0.00}$ | 0.00 | $10.83_{0.31}$ | $10.81_{0.31}$ | 0.02 |
| | | | InternVL-V2 | 40B | $0.40_{0.20}$ | $0.90_{0.29}$ | -0.50 | $12.30_{0.42}$ | $13.80_{0.48}$ | -1.50 |
| | | | InternVL-V2 | 76B | $0.20_{0.15}$ | $0.40_{0.20}$ | -0.20 | $14.96_{0.46}$ | $14.11_{0.46}$ | 0.85 |
| | | | MiniCPM-V2.5 | 8B | $0.2_{0.03}$ | $0.2_{0.01}$ | 0 | $7.23_{0.18}$ | $7.6_{0.13}$ | -0.37 |
| | | | MiniCPM-V2.5-FT | 8B | $2.2_{0.03}$ | $1.4_{0.06}$ | -0.2 | $18.01_{0.35}$ | $15.25_{0.26}$ | 2.76 |
| | | | Monkey | 7B | $0.0_{0.0}$ | $0.0_{0.0}$ | 0 | $5.69_{0.15}$ | $6.3_{0.13}$ | -0.61 |
| | | | Ovis2 | 34B | $4.39_{0.61}$ | $2.69_{0.52}$ | 1.70 | $21.23_{0.78}$ | $18.85_{0.68}$ | 2.38 |
| | | | Qwen-VL | 7B | $0.0_{0.0}$ | $0.0_{0.0}$ | 0 | $1.1_{0.07}$ | $0.15_{0.01}$ | 0.94 |
| | | | Qwen2-VL | 7B | $18.76_{1.22}$ | $26.75_{1.40}$ | -7.98 | $43.84_{1.10}$ | $53.56_{1.09}$ | -9.72 |
| | | | Qwen2-VL | 72B | $15.87_{1.13}$ | $27.54_{1.43}$ | -11.68 | $40.38_{0.99}$ | $53.95_{1.03}$ | -13.57 |
| | | | Qwen2.5-VL | 7B | $18.26_{1.21}$ | $12.67_{1.07}$ | 5.59 | $44.55_{1.04}$ | $31.33_{1.06}$ | 13.22 |
| | | | Qwen2.5-VL | 72B | $31.04_{1.40}$ | $29.64_{1.40}$ | 1.40 | $56.59_{1.09}$ | $54.07_{1.10}$ | 2.52 |
| | | | Yi-VL | 34B | $0.0_{0.0}$ | $0.0_{0.0}$ | 0 | $4.49_{0.09}$ | $1.73_{0.1}$ | 2.76 |

# E  MORE RELATIONSHIP BETWEEN VCR AND OTHER BENCHMARKS

The heatmap shown in Figure 6 provides a detailed view of the pairwise correlation between 23 different benchmarks used to evaluate 38 VLMs. The scores of these models on each benchmark were utilized to compute this correlation matrix. The color intensity and numerical values represent the degree of correlation, ranging from -1 (perfect negative correlation) to 1 (perfect positive correlation), with warmer colors indicating higher positive correlations and cooler colors indicating weaker or negative correlations.

We observe that $VCR_{EN, HARD}$ is markedly different from other benchmarks in the evaluation set. It demonstrates minimal and even sometimes negative correlation with other benchmarks. This suggests that the skill set required for $VCR_{EN, HARD}$ is largely unrelated to those tested by other popular tasks emphasizing more straightforward image-to-text associations or OCR capabilities. Specifically, $VCR_{EN, HARD}$ challenges models with tasks that involve high-level reasoning and minimal reliance on pixel-level information, focusing instead on understanding context, commonsense reasoning, and visual narrative interpretation. These features are less critical in other benchmarks, which explains the weak correlation across tasks. $VCR_{EN, EASY}$ exhibits a slightly stronger correlation with a few other benchmarks but remains moderately independent of most others. Like $VCR_{EN, HARD}$, $VCR_{EN, EASY}$ also evaluates visual commonsense reasoning, but with less stringent requirements, offering models more cues and simpler connections between visual elements and textual understanding. This leads to moderate overlap with benchmarks like TextVQA, which similarly focus on text understanding in a visual context, but $VCR_{EN, EASY}$ still emphasizes a higher level of interpretative reasoning than standard text-based vision benchmarks.

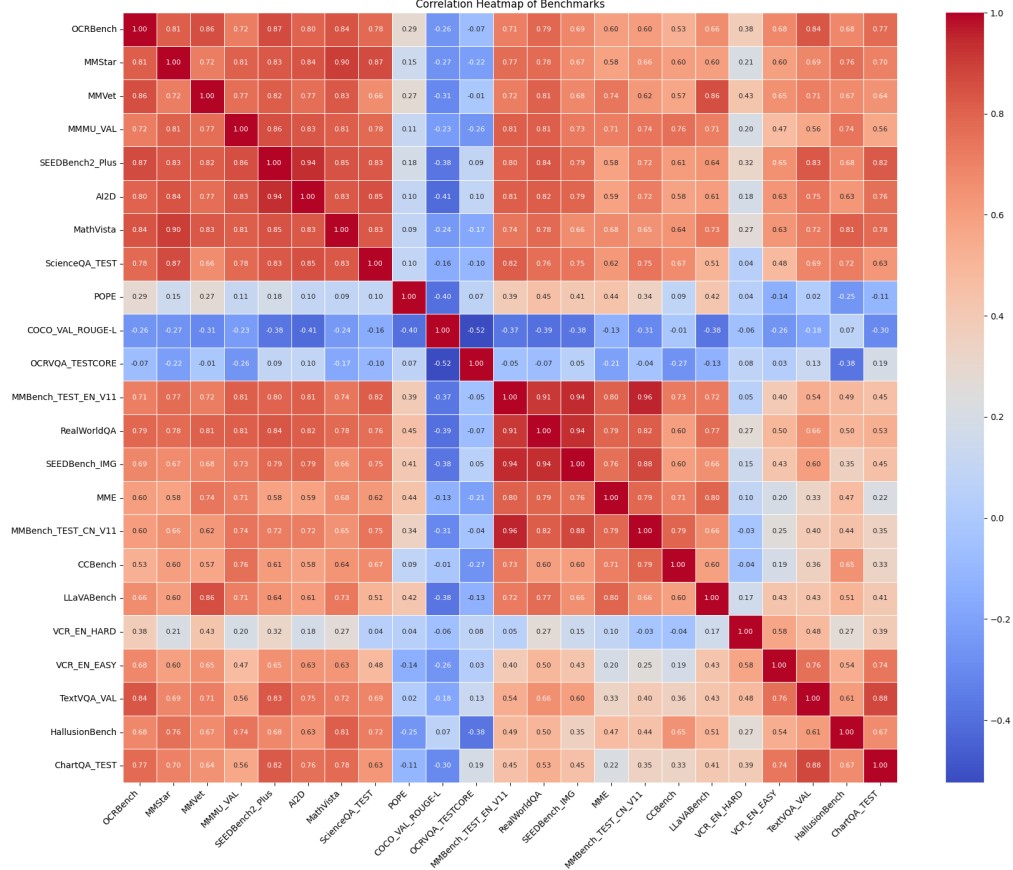

Figure 6: The heat map of benchmarks displays the correlation between the metric scores of 38 models for each benchmark pair.

## F    POTENTIAL QA

**What could be the possible reason that CogVLM performs well in VCR-WIKI series benchmarks?**    Many models we tested (DocOwl-1.5, Monkey, MiniCPM-V2.5, InternLM series, InternVL series) follow a similar inference pipeline to adapt to high-resolution application scenarios:

1. An algorithm divides the input image into segments.
2. Each segment is encoded into tokens using a CILP-based image encoder.
3. A filtering mechanism (algorithm/resampler/abstractor) processes the visual tokens.
4. The filtered tokens are concatenated with language tokens and input to the LLM.

If, in step 3, pixel-level hints embedded in text within the image (TEL) are disregarded, the model cannot correctly answer the question. Consequently, some of these models may perform better on benchmarks emphasizing global features but struggle on the **VCR-WIKI** series benchmarks, particularly in the hard partitions. For example, while InternVL2-40B performs best on $VCR_{EN, EASY}$, it does not perform well on $VCR_{EN, HARD}$. As noted in the paper, the easy partition of the benchmark primarily verifies that the VCR task is feasible for the models. In contrast, the hard partition explores the boundaries of VCR capability for both models and human test-takers (who require more time and focus to solve the puzzles in the hard partition).

The CogVLM2 and Cambrian-1 series, by contrast, do not include step 3 in their inference pipelines. Instead, their image encoders operate at mid-to-high resolutions (1K level), and they resize the input image to match the supported resolution rather than dividing it into segments. The image encoder resolution for CogVLM2 is $1344 \times 1344$, while Cambrian-1 employs four image encoders, the largest supporting a resolution of $1024 \times 1024$. This approach may encounter challenges with extremely shaped input images (e.g., $8192 \times 1024$), but for **VCR-WIKI**, where images are mostly near-square (on average $300 \times 360$ for $VCR_{ZH, EASY}$/$VCR_{ZH, HARD}$ and $300 \times 375$ for $VCR_{EN, EASY}$/$VCR_{EN, HARD}$), high-resolution support is not necessary. For instance, InternLM-XComposer2-VL outperforms InternLM-XComposer2-VL-4KHD on this benchmark.

**What could be the potential way to improve models' capability on VCR?**    To suggest potential avenues for improving VLM performance on VCR, we propose the following:

1. **Include VCR in VLM Pretraining**: Just as OCR parsing tasks are often included in pretraining to improve OCR performance, researchers could consider incorporating VCR tasks during pretraining. We will codebase to facilitate this process, making it as straightforward as data augmentation.

2. **Architectural Exploration**: CogVLM2 is the best-performing model on average across the four partitions, and we believe this is largely due to its vision expert architecture. We contacted the CogVLM2 team and learned that GLM-4 and CogVLM2 share the same training data, yet there is a significant performance gap between them on the VCR benchmarks.

3. **Chain-of-Thought (CoT) Methods**: Researchers could explore multi-modality pipelines based on CoT techniques to improve existing VLMs on VCR tasks (Chen et al., 2024a; Zhang et al., 2023). Although a model might not initially focus on the correct visual area (e.g., pixel-level hints in the TEI), CoT-based techniques could help refine its focus over successive rounds.

**How is the Transferability of VCR-WIKI Finetuning?**    In Table 10, we show the transferability of **VCR-WIKI** by finetuning multiple models on different finetuning datasets' training sets and testing their performance on a series of benchmarks.

The analysis of our experimental results highlights the strong transferability of the proposed **VCR-WIKI** dataset across various benchmarks. Notably, models fine-tuned on **VCR-WIKI** demonstrate significant performance improvements not only within the **VCR-WIKI** benchmarks themselves, but also across different language settings. For example, fine-tuning CogVLM2 on $VCR_{EN, HARD}$ leads to a substantial increase in performance on the $VCR_{ZH, EASY}$ benchmark, elevating the score from 9.15 to 42.55. Similarly, fine-tuning the Chinese version of CogVLM2 on $VCR_{ZH, HARD}$ enhances its

performance on both the $VCR_{EN, EASY}$ and $VCR_{EN, HARD}$ benchmarks, with scores rising from 79.9 to 87.57 and from 25.13 to 44.97, respectively. These enhancements indicate that the VCR-wiki dataset facilitates the learning of effective transferable features even when the fine-tuning and evaluation involve different languages.

Additionally, the consistent achievement of the highest scores within each model's finetuning variations underscores the robustness of **VCR-WIKI** in improving model performance across diverse evaluation metrics. This evidence collectively demonstrates that **VCR-WIKI** serves as a versatile and powerful resource for enhancing model generalization and performance across multiple tasks and linguistic contexts.

Table 10: Performance Comparison of Base Models.

| Base Model | MiniCPM-V2.5 | MiniCPM-V2.5 | MiniCPM-V2.5 | MiniCPM-V2.5 | CogVLM2 | CogVLM2 | CogVLM2 | CogVLM2-Ch. | CogVLM2-Ch. | CogVLM2-Ch. |
|---|---|---|---|---|---|---|---|---|---|---|
| Finetuning_dataset | None | OKVQA-Train | $VCR_{EN, HARD}$ | $VCR_{ZH, HARD}$ | None | OKVQA-Train | $VCR_{EN, HARD}$ | None | OKVQA-Train | $VCR_{ZH, HARD}$ |
| OKVQA | 77.43 | 72.20 | **77.09** | 76.38 | 75.35 | 71.86 | **75.45** | 74.16 | 70.57 | 74.14 |
| $VCR_{EN, EASY}$ | 31.81 | 19.53 | **40.96** | 28.40 | 83.25 | 79.29 | **93.27** | 79.90 | 30.18 | **87.57** |
| $VCR_{EN, HARD}$ | 1.41 | 0.00 | **13.86** | 5.33 | 37.98 | 27.22 | **77.44** | 25.13 | 3.55 | **44.97** |
| $VCR_{ZH, EASY}$ | 4.10 | 2.12 | 2.66 | **7.44** | 9.15 | 16.49 | **42.55** | 33.24 | 16.49 | **61.69** |
| $VCR_{ZH, HARD}$ | 0.09 | 0.53 | 1.06 | **1.53** | 0.08 | 0.00 | **1.60** | 1.34 | 0.00 | **42.11** |
| MMstar | 50.20 | **51.73** | 50.40 | 50.27 | 50.50 | **51.07** | 50.20 | 52.73 | 50.87 | **54.33** |
| MMBench_DEV_EN | **74.54** | 74.46 | **74.54** | 74.30 | 72.70 | **73.53** | 72.60 | **77.32** | 77.09 | 76.78 |
| MME | **2024.6** | 1996.7 | 1923.65 | 1977.13 | 1869.5 | 1860.56 | **1882.21** | **2040.7** | 1898.42 | 1939.8 |
| MMMU_VAL | 45.89 | **47.78** | 46.11 | 46.56 | **42.60** | 38.67 | 38.67 | 42.44 | 41.56 | **45.00** |
| AI2D_test | **78.04** | 77.85 | 77.62 | 77.91 | 73.40 | **74.97** | 73.61 | **72.64** | 70.98 | 71.31 |
| OCR_BENCH | **71.70** | 71.60 | 71.50 | 71.30 | 75.40 | 72.80 | **79.80** | 77.30 | 75.20 | **79.60** |
| MMVet | **53.12** | 45.78 | 51.10 | 52.66 | 57.80 | 45.00 | **59.31** | 56.38 | 39.77 | **56.88** |
| MathVista_MINI | **54.50** | 53.70 | 52.80 | 52.70 | **38.60** | 38.30 | 35.40 | 37.80 | 37.60 | **40.20** |
| ChartQA | 71.80 | 72.12 | 71.92 | **72.52** | 72.80 | **80.40** | 79.56 | 63.16 | 58.92 | **65.84** |
| OCRVQA | 61.85 | **62.70** | 61.36 | 61.59 | 64.90 | 65.56 | **66.11** | 32.65 | **34.41** | 33.24 |

## G    REAL-WORLD CASE STUDY OF MODELS FINE-TUNED ON **VCR-WIKI**

This case study aims to assess the real-world applicability of models fine-tuned on **VCR-WIKI** for recognizing occluded text, a challenging task with significant practical implications. The setting involves evaluating the performance of three state-of-the-art models, namely MiniCPM-V2.5 8B, CogVLM2 19B, and Qwen2-VL 7B, on a curated dataset of eleven photographs featuring occluded text from real-world scenarios, such as street maps and collected images. Since no standard benchmark exists for real-world occluded text recognition, this dataset serves as a proxy to measure the efficacy of VCR fine-tuning in improving performance. We show whether the model completely recovers the occluded or distorted texts in the image with ✓ (correct) or ✗ (partially correct or incorrect). This evaluation provides insight into how VCR fine-tuning translates to practical challenges and complements the quantitative analyses presented in the main paper.

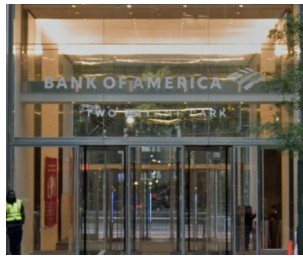

Figure 7: Ground-truth: BANK OF AMERICA TWO BRYANT PARK
- MiniCPM: ✓
- MiniCPM-ft: ✓
- CogVLM2: ✓
- CogVLM2-ft: ✓
- Qwen2-VL-7B: ✓
- Qwen2-VL-7B-ft: ✓

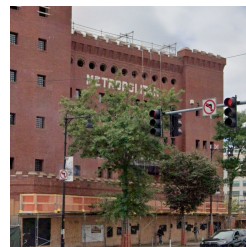

Figure 8: Ground-truth: METROPOLITAN
- MiniCPM: ✓
- MiniCPM-ft: ✓
- CogVLM2: ✓
- CogVLM2-ft: ✓
- Qwen2-VL-7B: ✓
- Qwen2-VL-7B-ft: ✓

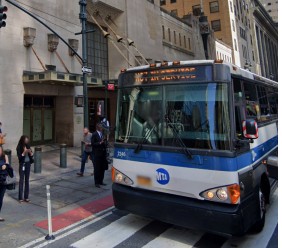

Figure 9: Ground-truth: NOT IN SERVICE
- MiniCPM: ✗
- MiniCPM-ft: ✓
- CogVLM2: ✓
- CogVLM2-ft: ✓
- Qwen2-VL-7B: ✓
- Qwen2-VL-7B-ft: ✓

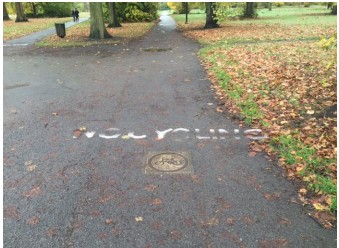

Figure 10: Ground-truth: NO CYCLING
- MiniCPM: ✗
- MiniCPM-ft: ✗
- CogVLM2: ✓
- CogVLM2-ft: ✓
- Qwen2-VL-7B: ✗
- Qwen2-VL-7B-ft: ✓

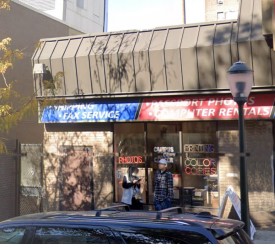

Figure 11: Ground-truth: SHIPPING FAX SERVICE PASSPORT PHOTOS COMPUTER RENTALS
- MiniCPM: ✗
- MiniCPM-ft: ✗
- CogVLM2: ✓
- CogVLM2-ft: ✓
- Qwen2-VL-7B: ✓
- Qwen2-VL-7B-ft: ✓

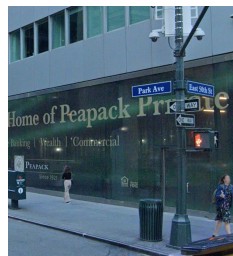

Figure 12: Ground-truth: Home of Peapack Private
- MiniCPM: ✓
- MiniCPM-ft: ✓
- CogVLM2: ✓
- CogVLM2-ft: ✓
- Qwen2-VL-7B: ✓
- Qwen2-VL-7B-ft: ✓

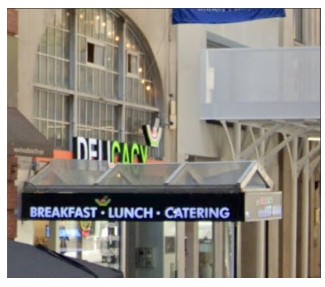

Figure 13: Ground-truth:
**DELICACY BREAKFAST LUNCH
CATERING**

- MiniCPM: ✓
- MiniCPM-ft: ✓
- CogVLM2: ✓
- CogVLM2-ft: ✓
- Qwen2-VL-7B: ✓
- Qwen2-VL-7B-ft: ✓

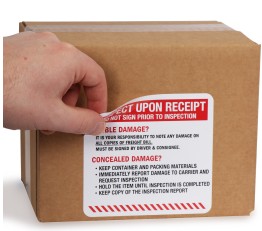

Figure 14: Ground-truth:
**INSPECT UPON RECEIPT...
DO NOT SIGN...
VISIBLE DAMAGE?**

- MiniCPM: ✗
- MiniCPM-ft: ✗
- CogVLM2: ✓
- CogVLM2-ft: ✓
- Qwen2-VL-7B: ✓
- Qwen2-VL-7B-ft: ✓

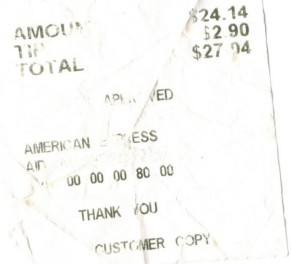

Figure 15: Ground-truth:
**AMOUNT TIP TOTAL APPROVED
AMERICAN EXPRESS AID
THANK YOU / MERCI
CUSTOMER COPY**

- MiniCPM: ✗
- MiniCPM-ft: ✓
- CogVLM2: ✓
- CogVLM2-ft: ✓
- Qwen2-VL-7B: ✓
- Qwen2-VL-7B-ft: ✓

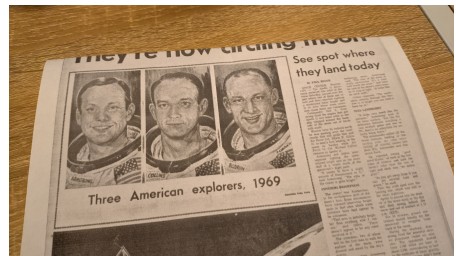

Figure 16: Ground-truth:
**INSPECT UPON RECEIPT...
DO NOT SIGN...
VISIBLE DAMAGE?**

- MiniCPM: ✗
- MiniCPM-ft: ✗
- CogVLM2: ✓
- CogVLM2-ft: ✓
- Qwen2-VL-7B: ✓
- Qwen2-VL-7B-ft: ✓

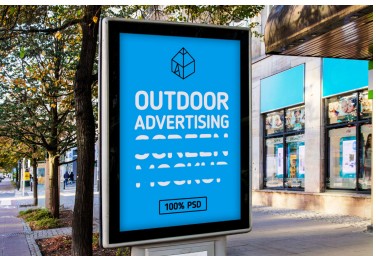

Figure 17: Ground-truth:
**AMOUNT TIP TOTAL APPROVED
AMERICAN EXPRESS AID
THANK YOU / MERCI
CUSTOMER COPY**

- MiniCPM: ✗
- MiniCPM-ft: ✓
- CogVLM2: ✓
- CogVLM2-ft: ✓
- Qwen2-VL-7B: ✓
- Qwen2-VL-7B-ft: ✓

