# OpenReview forum: "VCR: A Task for Pixel-Level Complex Reasoning in Vision Language Models via Restoring Occluded Text"
_ICLR.cc/2025/Conference — ICLR 2025 Poster_

### Official Review · Reviewer_L8oC · 2024-11-01

**Soundness:** 3
**Presentation:** 3
**Contribution:** 3
**Rating:** 6
**Confidence:** 4

**Summary:**

This paper presents Visual Caption Restoration (VCR), a new VL task where models need to accurately restore partially obscured text embedded within images using pixel-level hints and contextual cues. Unlike conventional OCR tasks, VCR requires the model to align vision and language modalities to restore text that is often obscured or fragmented. The authors introduce VCR-WIKI, a large-scale synthetic dataset generated using Wikipedia captions, supporting both English and Chinese text in easy and hard configurations. Experiments demonstrate that state-of-the-art vision-language models fall significantly short of human performance on the VCR task, highlighting the need for new model architectures that handle complex intermodal alignment.

**Strengths:**

The paper identifies a critical gap in vision-language tasks by focusing on restoring partially occluded text within images. This differs from traditional VQA and OCR by challenging models to reconstruct text based on incomplete visual information and context, a complex multimodal problem.

The VCR-WIKI dataset provides a scalable and controllable environment for testing models under different levels of text occlusion. The authors’ pipeline generates a diverse dataset that is balanced across languages, allowing for cross-linguistic studies and detailed analysis of model capabilities.

By analyzing the model failures, the paper provides useful insights into the challenges of multimodal alignment, reasoning, and generalization. The authors' discussion on how models with strong OCR capabilities underperform on VCR highlights the distinct challenges VCR poses.

**Weaknesses:**

The reliance on synthetically generated images raises questions about the generalizability of models trained on VCR-WIKI to real-world applications. Text within real images can exhibit more variability in terms of style, font, and environmental factors that may not be fully captured by VCR-WIKI.

The paper's use of GPT-based metrics could introduce variability in assessment, especially given the subjective nature of scoring masked text restoration. While Exact Match and Jaccard Index provide quantitative metrics, more robust evaluation strategies would be beneficial to standardize results across future works.

The synthetic dataset may not fully replicate the nuances of real-world images with natural text occlusion. Additional discussion on the challenges of transferring VCR models to real-world scenarios would clarify the dataset's practical applicability.

**Questions:**

Since VCR-WIKI relies on synthetic captions and partially obscured text, I was wondering to what extent you expect models trained on this dataset to generalize to real-world scenarios where text may have more complex occlusions or be influenced by environmental factors (e.g., lighting, noise).

---

> ### Author Response · Authors · 2024-11-20
>
> Thank you for your detailed and thoughtful review. We greatly appreciate your insights and constructive feedback. Below, we address the specific concerns you raised to clarify and strengthen our submission.

---

> ### Author Response · Authors · 2024-11-20
>
> **Q1. Generalizability of VCR Models to Real-World Scenarios (Concerns from Weaknesses Q1, Q3, and Questions Q1)**
>
> To address this question, we extend our discussion in the following three aspects:
>
> * **Real-world applicability of models finetuned on VCR**: To evaluate whether fine-tuning on the VCR dataset enhances real-world recognition of partially occluded text, we assess the performance of three models used in Figure 5: MiniCPM-V2.5 8B, CogVLM2 19B, and Qwen2-VL 7B. Since no publicly available datasets for real-world occluded text recognition currently exist, we manually curated a small dataset comprising ten photographs of occluded text for a case study, which is sourced from street maps and collected images. The comparison of pre- and post-finetuning performance is provided below. For clarity, text predictions directly relevant to the results were manually extracted from the model outputs. We define "Incorrect" as not having overlap with the ground truth, and "Partially Correct" as having overlap but not completely correct.
>
> | Index | Ground-truth | MiniCPM | **MiniCPM (finetuned)** | CogVLM2 | **CogVLM2 (finetuned)** | Qwen2-VL | **Qwen2-VL (finetuned)** |
> |---|---|---|---|---|---|---|---|
> | 1 | SCREEN MOCKUP | x | x | SCREEN MOCKUP | SCREEN MOCKUP | SCREEN MOCKUP | SCREEN MOCKUP |
> | 2 | INSPECT UPON RECEIPT... DO NOT SIGN... VISIBLE DAMAGE? | EFFECT UPON RECEIPT... NOT SIGN... BLE DAMAGE? | EFFECT UPON RECEIPT... NOT SIGN... BLE DAMAGE? | INSPECT UPON RECEIPT... DO NOT SIGN... VISIBLE DAMAGE? | INSPECT UPON RECEIPT... DO NOT SIGN... VISIBLE DAMAGE? | INSPECT UPON RECEIPT... DO NOT SIGN... VISIBLE DAMAGE? | INSPECT UPON RECEIPT... DO NOT SIGN... VISIBLE DAMAGE? |
> | 3 | AMOUNT TIP TOTAL APPROVED AMERICAN EXPRESS AID THANK YOU / MERCI CUSTOMER COPY | AMOUNT TIP TOTAL  APPL. YED AMERICAN EXPRESS  AID THANK YOU / MERCI  CUSTOMER COPY | AMOUNT TIP TOTAL APPROVED AMERICAN EXPRESS AID THANK YOU/ MERCI CUSTOMER COPY | AMOUNT TIP TOTAL APPROVED AMERICAN EXPRESS AID THANK YOU / MERCI CUSTOMER COPY | AMOUNT TIP TOTAL APPROVED AMERICAN EXPRESS AID THANK YOU / MERCI CUSTOMER COPY | AMOUNT TIP TOTAL APPROVED AMERICAN EXPRESS AID THANK YOU / MERCI CUSTOMER COPY | AMOUNT TIP TOTAL APPROVED AMERICAN EXPRESS AID THANK YOU / MERCI CUSTOMER COPY |
> | 4 | They're now circling moon | x | They're now circling moon | They're now circling moon | They're now circling moon | They're now circling moon | They're now circling moon |
> | 5 | NO CYCLING | WOLVERHAMPTON | WOLVERHAMPTON | NO CYCLING | NO CYCLING | VOLYCLIN | NO CYCLING |
> | 6 | BANK OF AMERICA TWO BRYANT PARK | BANK OF AMERICA TWO BRYANT PARK | BANK OF AMERICA TWO BRYANT PARK | BANK OF AMERICA TWO BRYANT PARK | BANK OF AMERICA TWO BRYANT PARK | BANK OF AMERICA TWO BRYANT PARK | BANK OF AMERICA TWO BRYANT PARK |
> | 7 | DELICACY BREAKFAST LUNCH CATERING | DELICACY BREAKFAST LUNCH CATERING | DELICACY BREAKFAST LUNCH CATERING | DEUCACY BREAKFAST LUNCH CATERING | DELICACY BREAKFAST LUNCH CATERING | DELICACY BREAKFAST LUNCH CATERING | DELICACY BREAKFAST LUNCH CATERING |
> | 8 | NOT IN SERVICE | x | NOT IN SERVICE | NOT IN SERVICE | NOT IN SERVICE | NOT IN SERVICE | NOT IN SERVICE |
> | 9 | Home of Peapack Private | Home of Peapack Private | Home of Peapack Private | Home of Peapack Private | Home of Peapack Private | Home of Peapack Private | Home of Peapack Private |
> | 10 | METROPOLITAN | METROPOLITAN | METROPOLITAN | METROPOLITAN | METROPOLITAN | METROPOLITAN | METROPOLITAN |
> | 11 | SHIPPING FAX SERVICE PASSPORT PHOTOS COMPUTER RENTALS | FAX SERVICE PHOTOS COMPUTER RENTALS | FAX SERVICE PHOTOS COMPUTER RENTALS | SHIPPING FAX SERVICE PASSPORT PHOTOS COMPUTER RENTALS | SHIPPING FAX SERVICE PASSPORT PHOTOS COMPUTER RENTALS | SHIPPING FAX SERVICE PASSPORT PHOTOS COMPUTER RENTALS | SHIPPING FAX SERVICE PASSPORT PHOTOS COMPUTER RENTALS |
> | **Total** |  | **4 Incorrect, 3 Partially Correct** | **2 Incorrect, 2 Partially Correct** | **1 Partially Correct** | **All Correct** | **1 Incorrect** | **All Correct** |
>
> The evaluation of MiniCPM-V2.5 8B, CogVLM2 19B, and Qwen2-VL 7B models before and after fine-tuning on the VCR dataset demonstrates significant improvements in real-world recognition of partially occluded text.
>
> These enhancements indicate that **fine-tuning on VCR-wiki markedly boosts the models' accuracy and reliability in interpreting occluded text in real-world scenarios.** However, it is important to note that this analysis is based on a case study with only ten examples. Despite the limited sample size, the consistent performance gains across different architectures highlight the potential effectiveness of VCR-based fine-tuning. Future evaluations with larger and more diverse datasets are necessary to further validate and generalize these promising findings, ensuring robust real-world applicability of the fine-tuned models in dynamic and challenging text recognition tasks. **We have added this real-world case study in Appendix E in the revised paper.**

---

> ### Author Response · Authors · 2024-11-20
>
> (continue: answer to Q1)
> * **Transferability validation on other benchmarks**: VCR-wiki serves as targeted training data to improve the model's text-image alignment and text recognition capabilities. Although VCR-wiki does not feature variability in style, fonts, and other factors, we believe it is a strong enhancement based on the VLMs' existing text recognition capabilities. Below, we conducted additional experiments to evaluate how well models fine-tuned on VCR-Wiki generalize to other tasks, which has been included in Appendix D in the original submission. This includes tasks with diverse real-world scenarios. Specifically, we fine-tuned CogVLM2 and CogVLM2-Chinese, two of the "finetune-able" VLMs in Section 4.4, on VCR-wiki (English and Chinese) and OKVQA training sets, then assessed their performance across multiple benchmarks. The results are summarized in the table below, where **bold** entries denote the highest scores for a given base model:
>
> | Base Model              |  CogVLM2 | CogVLM2 | CogVLM2 | CogVLM2-Chinese | CogVLM2-Chinese | CogVLM2-Chinese |
> |-------------------------|---------|---------|---------|-----------------|-----------------|-----------------|
> | Finetuning_dataset  |  None    | OKVQA-Train | VCR-wiki-en-hard-Train | None            | OKVQA-Train     | VCR-wiki-zh-hard-Train |
> | OKVQA               |  75.35   | 71.86   | **75.45**   | **74.16**           | 70.57           | 74.14           |
> | VCR_EN_EASY         |  83.25   | 79.29   | **93.27**| 79.90            | 30.18           | **87.57**       |
> | VCR_EN_HARD         |  37.98   | 27.22   | **77.44**| 25.13           | 3.55            | **44.97**           |
> | VCR_ZH_EASY         |  9.15    | 16.49   | **42.55**   | 33.24           | 16.49           | **61.69**       |
> | VCR_ZH_HARD         |  0.08    | 0       | **1.6**     | 1.34            | 0               | **42.11**       |
> | MMstar              |  50.5    | **51.07**   | 50.20    | 52.73           | 50.87           | **54.33**       |
> | MMBench_DEV_EN_V11  |  72.70    | **73.53**   | 72.60    | **77.32**       | 77.09           | 76.78           |
> | MME                 |  1869.5  | 1860.56 | **1882.21** | **2040.7**      | 1898.42         | 1939.80          |
> | MMMU_VAL            |  **42.60**    | 38.67   | 38.67   | 42.44           | 41.56           | **45.00**              |
> | AI2D_test           |  73.40    | **74.97**   | 73.61   | **72.64**           | 70.98           | 71.31           |
> | OCR_BENCH           |  75.40    | 72.80    | **79.8**| 77.3            | 75.2            | **79.6**        |
> | MMVet               |  **57.8**| 45.00      | 59.31   | 56.38           | 39.77           | **56.88**           |
> | MathVista_MINI      |  **38.60**    | 38.30    | 35.4    | 37.8            | 37.60            | **40.20**            |
> | ChartQA             |  72.80    | **80.4**| 79.56   | 63.16           | 58.92           | **65.84**           |
> | OCRVQA              |  64.90    | 65.56   | **66.11**| 32.65           | **34.41**           | 33.24           |
>
> These results demonstrate that **models fine-tuned on VCR-wiki not only achieve competitive or superior performance compared to the base model and OKVQA fine-tuning on standard benchmarks but also show significant gains in tasks like OCR-Bench, which feature diverse real-world scenarios**. This suggests that VCR-wiki fine-tuning enhances models' ability to generalize to varied text recognition and multimodal reasoning tasks.
>
> * **Flexibility of VCR as a framework**: The VCR task framework is *inherently customizable*, allowing researchers to tailor datasets to domain-specific challenges. For example, our open-source codebase enables users to vary data sources, occlusion patterns, fonts, and environmental settings. The current VCR-wiki dataset represents a foundational instantiation of this framework, demonstrating that even simple configurations that mimic the format of an image with a caption still pose significant challenges to state-of-the-art VLMs. **This flexibility supports future extensions to more nuanced real-world scenarios.** In this paper, as a first step, we focus on providing more insights through a series of controllable experiments to extract the essence behind models' success or failure patterns, which would benefit from a simple yet challenging configuration.

---

> ### Author Response · Authors · 2024-11-20
>
> **Q2. Robustness of Evaluation Metrics (Concern from Weaknesses Q2)**
>
> We appreciate your suggestion regarding the evaluation strategies and offer the following clarifications:
>
> * **Characteristics of VCR**: Unlike freeform generation tasks such as Masked Language Modeling (MLM), VCR is an *objective* text recovery task with a single ground truth, determined by partially exposed text in images. This makes evaluation similar to tasks like multiple-choice questions or math problems, where exact-match accuracy is a reliable metric. In this context, generation outputs that deviate even slightly from the ground truth (e.g., incorrect letters) are considered unfaithful to the input, warranting stricter evaluation. This characteristic also allows us to avoid the drawbacks of freeform generation tasks (like MLM or QA tasks), where a proper automatic evaluation is hard to acquire [1]. **The accurate and unambiguous evaluation is a strength of the task of VCR.**
>
> * **Choice of Metrics**: To account for partial correctness, we supplement Exact Match (EM) with the Jaccard Index, a cost-effective and widely used metric. This approach leverages the structure of our task, where covered texts are 5-grams, making partial matching straightforward to compute and interpret. **The combination of Exact Match and Jaccard Index ensures a balance between precision and nuance in evaluation.**
>
> * **Inappropriateness of GPT-Based Metrics**: While GPT-based metrics can approximate human judgment for freeform generation tasks, they are less suitable for VCR. **The high evaluation cost and variability of GPT-based scoring are unnecessary for tasks with a well-defined ground truth**, where automatic metrics like Exact Match offer more reliable and consistent evaluation. Therefore, we believe our current approach is robust and aligns well with the nature of the VCR task.
>
> [1] Wang, Cunxiang, et al. Evaluating Open-QA Evaluation. In NeurIPS 2023.
>
> We hope these responses address your concerns and provide additional clarity on the strengths and flexibility of our proposed task and dataset. Thank you again for your valuable feedback, which has helped us further refine and strengthen our submission.

---

> > ### Comment · Reviewer_L8oC · 2024-11-26
> >
> > Thank you for your hard work. I think all of my concerns are well addressed.

---

### Official Review · Reviewer_kwun · 2024-11-01

**Soundness:** 3
**Presentation:** 3
**Contribution:** 3
**Rating:** 8
**Confidence:** 4

**Summary:**

The paper introduces the Visual Caption Restoration (VCR) benchmark, designed to assess vision-language models' (VLMs) ability to restore partially obscured text using visual cues from images. VCR is built with images and captions sourced from Wikipedia, encompassing 2.11M English and 346K Chinese entries, presented in both easy and hard configurations. Experiments are conducted on state-of-the-art (SOTA) VLMs to evaluate their performance.

**Strengths:**

* The paper is well-written and easy to follow.
* VCR is an interesting benchmark that poses challenges to existing VLMs. It's a nice addition to the VLM community.
* The evaluation of most SOTA VLMs offers insightful findings that advance the field.

**Weaknesses:**

* Certain important details about the dataset are missing from the paper; please refer to the questions section below.
* The experiments on fine-tuning VLMs on VCR are not informative enough, as the VLMs of choice differ too much in terms of backbone/architecture/training pipeline... I feel like stating "the architectural design of CogVLM2 is well-suited for VCR" in line 368 might need more experiments or well-controlled experimental setups.

**Questions:**

* As mentioned above, my main question centered around the details about the dataset:
  * Given that training data for SOTA VLMs often includes Wikipedia content, how does the benchmark ensure that images and captions in the dataset have not been seen during VLM pre-training or SFT? When were these images and captions posted?
  * The difference between easy and hard cases is vague.Separating these cases based solely on the area of visible pixels may be insufficient, as the difficulty level for restoring a word can also depend on the word itself. For example, if both "area" and "idea" are 80% covered, it may be challenging to determine which is easier to recognize. The word "idea" initially occupies more pixels, and the "d" token could serve as a clue, making it more recognizable. This suggests that the difficulty of a sample may also depend on word combinations in the sentence. What are the detailed criteria (e.g. covering proportion) for choosing easy/hard samples?
  * Following up on the previous point, if the covering proportion is automatically used to select easy and hard samples, would it be possible to manually curate these samples based on word combinations? I understand that this would involve additional effort, so this is just an open question I'm interested in.



---------
Post rebuttal:


Having reviewed all the reviews and rebuttals again, I have decided to raise my rating. I also encourage the authors to continue exploring and developing this promising topic.

---

> ### Author Response · Authors · 2024-11-20
>
> Thank you for your thoughtful review and for raising these important points. We have addressed each concern in detail below.
>
> ---
>
> **Q1. Regarding the statement: "the architectural design of CogVLM2 is well-suited for VCR"**
>
> We appreciate your observation and agree that attributing a model's success to specific design choices requires more comprehensive and well-controlled experiments. Based on your feedback, we will revise this statement in the paper to ensure it is more balanced and reflective of the current evidence. We aim to avoid overstatements and have clarified that this claim is hypothesis-driven and warrants further empirical validation *in the revised Section 4.4. Please refer to the purple text in the revision.*
>
> **Q2. Certain important details about the dataset are missing from the paper.**
>
> We address your concerns about the dataset design and experimental setup in the following points:
>
> **Q2.1 how does the benchmark ensure that images and captions in the dataset have not been seen during VLM pre-training or SFT?**
>
> We acknowledge that Wikipedia is commonly used in the pretraining data for many state-of-the-art VLMs. This, in fact, *was one of the motivating factors for building the VCR dataset from Wikipedia*. As evidenced in Table 2, even though many VLMs have likely been exposed to similar data during pretraining, their performance on the VCR task is subpar. Notably, models struggle significantly with the Chinese VCR-wiki benchmarks.
>
> This substantial performance gap highlights that **state-of-the-art VLMs fail on VCR-wiki not due to unfamiliarity with the underlying text, but because they lack the necessary capabilities to effectively perform the VCR task**. Furthermore, we selected Wikipedia due to its general licensing, which enables us to fully open-source the dataset and align with the principles of open science. *We have included this analysis in the revised Section 4.4. Please refer to the purple text in the revision.*
>
> **Q2.2 The difference between easy and hard cases is vague.**
>
> To clarify, we do not *independently* select samples for the easy and hard settings. Instead, **both settings share the exact same examples, data splits, and covered text spans**. The difference lies solely in the degree of pixel masking applied: the hard setting obscures more pixels compared to the easy setting. **This design ensures that the easy and hard settings are directly comparable, as they differ only in the pixel-level masking applied to each instance.**
>
> For example, if the word "area" is partially masked in the easy setting, the same word will appear with greater masking in the corresponding hard setting. Thus, the easy setting is always relatively easier than the hard setting for a given example. Figure 3 ("Step 3: Create TEI") illustrates this process. *We have changed our narrative to be clearer in the revised Section 3. Please refer to the purple text in the revision.*
>
> **Q2.3 Would it be possible to manually curate easy samples based on word combinations?**
>
> Thank you for this insightful suggestion! As stated above, the easy and hard examples are created from the same image-text pairs, with the difference being the proportion of pixels masked. While we currently do not select examples based on specific word combinations or letter characteristics, your idea to consider letter heights or other semantic attributes is intriguing and represents a promising direction for future work.
>
> We have integrated your suggestion into our codebase, adding support for whitelisting and blacklisting specific words for masking, along with fine-grained masking based on letter heights. Future iterations will explore more nuanced strategies leveraging letter height or other word-level attributes. *The features and flexibility of our VCR codebase are highlighted in the revised Section 3.*
>
> We hope these clarifications and updates address your concerns and strengthen the contribution of our work. Thank you again for your valuable feedback.

---

> > ### Comment · Reviewer_kwun · 2024-11-22
> >
> > Thank you for your response. After reviewing the authors' rebuttal and considering the other reviews, most of my concerns have been addressed. I will maintain my original positive score.

---

> > > ### Author Response · Authors · 2024-11-22
> > >
> > > Thank you for taking the time to review our rebuttal and for your positive feedback. We sincerely appreciate your thoughtful comments and your support of our work!

---

### Official Review · Reviewer_2nhH · 2024-11-06

**Soundness:** 2
**Presentation:** 3
**Contribution:** 2
**Rating:** 6
**Confidence:** 5

**Summary:**

The paper introduces a new vision-language evaluation task (VCR), for assessing models performance on restoring partially obscured text embedded in images, using pixel-level hints (pixels from the top and the bottom of the partially obscured characters) and the visual contexts of the image. This approach mirrors human proficiency in recognising partially hidden objects, a skill supported by complex visual and cognitive processes, as noted in prior studies.

The authors claim, that the traditional VQA tasks, especially in the area of embedded text recognition, work independently the non-text visual context of the image, a gap that the VCR task covers. To support this claim, they are analysing the correlation of the VCR task to other VLM benchmarks and they demonstrate low correlation especial for the VCR-HARD task, where pixel hints are minimal.

For this topic, the authors developed a pipeline for generating synthetic images for the VCR task, using image-caption pairs, with configurable caption visibility to control the difficulty level of the task. Using this pipeline they developed the VCR-wiki dataset, from images and captions coming from Wikipedia. The dataset contains 2.11M English and 346K Chinese samples. There are two versions of the dataset, an EASY one and a HARD one (both feasible for native speakers of the language), where these two versions differ on the amount of the pixels remain visible from the obscured text. Each dataset has a Train, a Validation and a Test split.

For the evaluation task the authors introduce two metrics, an Exact Match, and Jaccard Index (Bag-of-Words as sets) for comparing the ground truth (obscured text) with the models' response (recovery of the obscured text).

The authors include in the paper several VCR evaluation experiments for many open and closed source models. They show that the majority of the models performs poorly on this task, and they draw the below conclusions:
* VCR is a challenging task for SOTA VLMs, while humans score high in this task
* Models that perform well in OCR tasks, can score low in VCR
* Performance degradation on Chinese version of the task, showing the need of improvements in multilingual capabilities
* Larger models, or higher image resolutions are not correlated with better performance in VCR task
* Non-text image information is not effectively leveraged by the most of the models in this task
* Model architecture can be a performance improvements resisting factor when applying fine-tuning on VCR-wiki dataset

**Strengths:**

The paper introduces an original task for evaluating VLMs. The contributions are clearly explained, and there are sufficient details about them (metrics, pipeline for generating synthetic images for the VCR task, VCR-wiki dataset). The experiments are presented with clarity and the paper has a good structure.

**Weaknesses:**

The paper lacks a clear explanation of how improvements in VLMs, as reflected by higher scores on VCR task, would translate to overall advancements in VLM capabilities. There are not any specific ways mentioned in which enhanced performance on this task would lead to meaningful or practical improvements in VLM functionality, generalisation, or real-world applicability. Providing a more detailed rationale on this point would strengthen the paper, as it would help clarify the relevance of VCR task performance as a benchmark for broader VLM development.

There are additional questions regarding the choices made in various experimental settings, and providing more details on these decisions would strengthen the clarity of the paper. Specifically:
* Selected number of n-grams: The authors chose to use a 5-gram approach; however, it is not clear why this number was selected. Was this choice based on empirical results, or was it simply an arbitrary decision? Additionally, it would be informative to know if smaller n-grams were tested and, if so, how the models performed with those alternative configurations.
* Text visibility settings: For the "EASY" mode in the dataset, how did the authors determine the chosen visibility threshold? Was this the minimum level at which open-source OCR tools started to perform poorly, or was it a randomly selected setting? A discussion on how this visibility threshold was established would provide useful context, especially regarding the task's difficulty calibration.
* Fine-tuning configuration: Given the extensive data available in the VCR-wiki dataset, it would be helpful to understand if the authors experimented with fine-tuning the models using a larger subset of this data. Additionally, specifying the number of training epochs used in these experiments would offer valuable insight into the training process and potential impacts on model performance.
* Prompts: It would be beneficial to know if any prompt engineering was applied when experimenting with different models. For instance, how did the authors arrive at the chosen prompt? In relation to the observed performance degradation with VI, did the authors attempt prompts specifically guiding the model to utilize visual cues (e.g., "Use the visual information from the image to recover the missing text.")?

Addressing these points with more detail would enhance the paper’s transparency and rationale behind these experimental choices.

**Questions:**

Quoted from the paper: "Figure[3] is an example VCR task in hard mode, and Figure[1] shows an example VCR task in the easy
mode. Although humans can still fill the blanks easily in the hard mode, it is nearly impossible for models with only OCR capabilities to recover the covered texts without using the context." Instead of Figure[3] I think it should be Figure[2].

Suggestions:
Address the weaknesses mentioned above.

---

> ### Author Response · Authors · 2024-11-20
>
> Thank you for raising these important questions. We've provided our responses below.
>
> ---
>
> **Q1. How do models fine-tuned on VCR-Wiki perform on other benchmarks?**
>
> While VCR serves primarily as a benchmark, the training set is designed to enhance a model's text-image alignment capabilities for partially occluded text. To evaluate its transferability, we conducted additional experiments where we fine-tuned three VLMs—MiniCPM-V2.5, CogVLM2, and CogVLM2-Chinese—on VCR-wiki-en, VCR-wiki-zh, and OKVQA training sets. We then assessed their performance on other tasks. All fine-tuned models follow the same fine-tuning procedure. The table below summarizes the results. The **bold** scores are the best among the same model but different finetuning datasets. Please note that this experiment has been included in Appendix D in the original submission.
>
> | Base Model              | MiniCPM-V2.5 | MiniCPM-V2.5 | MiniCPM-V2.5 | MiniCPM-V2.5 | CogVLM2 | CogVLM2 | CogVLM2 | CogVLM2-Chinese | CogVLM2-Chinese | CogVLM2-Chinese |
> |-------------------------|--------------|--------------|--------------|--------------|---------|---------|---------|-----------------|-----------------|-----------------|
> | Finetuning_dataset  | None         | OKVQA-Train  | VCR-wiki-en-hard-Train | VCR-wiki-zh-hard-Train | None    | OKVQA-Train | VCR-wiki-en-hard-Train | None            | OKVQA-Train     | VCR-wiki-zh-hard-Train |
> | OKVQA               | 77.43        | 72.20         | **77.09**    | 76.38        | 75.35   | 71.86   | **75.45**   | **74.16**           | 70.57           | 74.14           |
> | VCR_EN_EASY         | 31.81        | 19.53        | **40.96**        | 28.40         | 83.25   | 79.29   | **93.27**| 79.90            | 30.18           | **87.57**       |
> | VCR_EN_HARD         | 1.41         | 0            | **13.86**        | 5.33         | 37.98   | 27.22   | **77.44**| 25.13           | 3.55            | **44.97**           |
> | VCR_ZH_EASY         | 4.10          | 2.12         | 2.66         | **7.44**         | 9.15    | 16.49   | **42.55**   | 33.24           | 16.49           | **61.69**       |
> | VCR_ZH_HARD         | 0.09         | 0.53         | 1.06         | **1.53**         | 0.08    | 0       | **1.6**     | 1.34            | 0               | **42.11**       |
> | MMstar              | 50.20         | **51.73**    | 50.40         | 50.27        | 50.5    | **51.07**   | 50.20    | 52.73           | 50.87           | **54.33**       |
> | MMBench_DEV_EN_V11  | **74.54**    | 74.46        | **74.54**    | 74.30         | 72.70    | **73.53**   | 72.60    | **77.32**       | 77.09           | 76.78           |
> | MME                 | **2024.6**   | 1996.70       | 1923.65      | 1977.13      | 1869.5  | 1860.56 | **1882.21** | **2040.7**      | 1898.42         | 1939.80          |
> | MMMU_VAL            | 45.89        | **47.78**    | 46.11        | 46.56        | **42.60**    | 38.67   | 38.67   | 42.44           | 41.56           | **45.00**              |
> | AI2D_test           | **78.04**    | 77.85        | 77.62        | 77.91        | 73.40    | **74.97**   | 73.61   | **72.64**           | 70.98           | 71.31           |
> | OCR_BENCH           | **71.70**         | 71.60         | 71.50         | 71.30         | 75.40    | 72.80    | **79.8**| 77.3            | 75.2            | **79.6**        |
> | MMVet               | **53.12**        | 45.78        | 51.10         | 52.66        | **57.8**| 45.00      | 59.31   | 56.38           | 39.77           | **56.88**           |
> | MathVista_MINI      | **54.5**     | 53.70         | 52.80         | 52.70         | **38.60**    | 38.30    | 35.4    | 37.8            | 37.60            | **40.20**            |
> | ChartQA             | 71.80         | 72.12    | 71.92        | **72.52**        | 72.80    | **80.4**| 79.56   | 63.16           | 58.92           | **65.84**           |
> | OCRVQA              | 61.85        | **62.7**     | 61.36        | 61.59        | 64.90    | 65.56   | **66.11**| 32.65           | **34.41**           | 33.24           |
>
> The results demonstrate that **fine-tuning on VCR-Wiki generally achieves on-par or superior performance compared to fine-tuning on OKVQA or no fine-tuning**, highlighting the transferability of the VCR-Wiki dataset.
>
> Interestingly, fine-tuning CogVLM2 on VCR-Wiki-En-Hard notably improved its performance on VCR-Wiki-Zh-Easy, raising the score from 9.15 to 42.55. Similarly, fine-tuning CogVLM2-Chinese on VCR-Wiki-Zh-Hard enhanced its performance on VCR-Wiki-En benchmarks. These results suggest that models trained on VCR-Wiki can effectively learn to utilize pixel-level hints in text-embedded images (TEI) and apply this knowledge across languages, further validating the dataset's utility.

---

> ### Author Response · Authors · 2024-11-20
>
> **Q2. Why did we choose 5-grams for masking?**
>
> The decision to use 5-grams for masking was informed by both linguistic and empirical considerations:
>
> * **Linguistic Analysis**: We would like the covered text to contain a relatively longer text span that would only capture a local grammatical structure. **A 5-gram mask typically captures meaningful grammatical structures**, such as noun or verb phrases with determiners, adjectives, prepositional phrases, basic clauses, and subordinating conjunctions. This makes the task *sufficiently challenging without rendering it infeasible*.
>
> * **Empirical Observations**: Longer masking spans (e.g., 6-grams) often resulted in instances with limited viable options for masking, reducing the size of the dataset, while shorter spans (e.g., 3-grams) made the task significantly easier according to our observations. Masking at the 5-gram level struck a balance between difficulty and task feasibility.
>
> We emphasize that VCR is a flexible task framework, allowing researchers to customize masking configurations based on specific domains or datasets. We include the options to mask out complete sentences and certain PoS taggings in our codebase. *We emphasize the reason behind masking 5-grams and the flexibility of our codebase in the revised version of the paper. Please refer to the purple text added in Section 3.*
>
> **Q3. How did we determine the visibility threshold of the 'easy' setting?**
>
> The visibility threshold in the "easy" setting was chosen to ensure that the task remained easy for human participants while becoming nearly impossible for widely used OCR engines like Tesseract and MMOCR. This threshold was empirically validated, as noted in Section 3 of the paper and calibrated to create a meaningful challenge for current VLMs.
>
> **Q4. Why not finetune the models using a larger subset of VCR-wiki?**
>
> Our fine-tuning experiments aim to examine models’ "finetune-ability" (i.e., whether the model is able to achieve performance gains after finetuning on VCR), as shown in Figure 5 of the paper. Due to resource constraints, we selected a relatively small but representative subset (16K samples). Despite this limitation, the results successfully highlight different model behaviors, which is the aim of the finetuning experiments:
>
> * Models with minimal improvement after fine-tuning (e.g., MiniCPM-V2.5).
> * Models performing well with or without fine-tuning (e.g., Qwen2-VL).
> * Models significantly benefiting from fine-tuning (e.g., CogVLM2).
>
> Further exploring the relationship between finetune-ability and model architecture is left for future work.
>
> **Q5. Specifying the number of training epochs would be valuable.**
>
> Each model was fine-tuned for 1 epoch. This detail, already included in Section 4.1 (Line 271), will be further emphasized in the revised version of the paper for clarity.
>
> **Q6. Is there any prompt engineering applied?**
>
> We tested 15 task description prompts generated by GPT-4o in our preliminary experiments, but we observed similar results across them. Therefore, we chose to use a single, generic and standardized prompt for simplicity and consistency, avoiding per-model optimization for each VLM. This decision reflects three considerations:
>
> * Optimized prompts for each model would increase experimental costs significantly.
> * A universal prompt aligns with our goal of creating VCR-wiki as a standardized and accessible testbed. A different prompt for each model or a prompt optimized specifically for a single model would make the test more costly or biased.
> * We call for future works that develop improved prompt-based VLM reasoning frameworks and tailored prompts to enhance models' reasoning capabilities on the VCR task.
>
> **Q7. For the performance degradation with VI, did the authors attempt prompts specifically guiding the model to utilize visual cues?**
>
> As mentioned in the response for Q6, we tested 15 different prompts in our preliminary experiments. In our tested prompts, we also included prompts that guide the model to "please utilize the visual cues in the image", which also results in a similar performance for the majority of the models. However, the VI setting does not have a visual cue. To ensure comparability between VI+TEI and VI settings, we kept prompts consistent across both settings to not include the "utilize the visual cues" expression. This ensures that the observed performance differences are only attributable to the image input, not prompt design. However, designing prompts or reasoning frameworks explicitly for better visual cue utilization is an interesting avenue for future work.
>
> **Q8. Figures are mistakenly linked.**
>
> Thank you for catching this oversight. We have corrected the figure references in the revised version.
>
> We appreciate your detailed feedback and believe that addressing these points strengthens the paper. Thank you for your thoughtful review and consideration.

---

> ### Comment · Reviewer_2nhH · 2024-11-26
>
> I thank the authors for the response and clarifications. I will maintain my initial positive score for this work.

---

> > ### Author Response · Authors · 2024-12-03
> >
> > Thank you for your thoughtful review and helpful suggestions. We appreciate your positive feedback on our work and the recognition of its contributions. It would be great if your updated understanding could be reflected in the score, as your insights have significantly enhanced our clarity and direction. Thank you again for your time and effort in reviewing our submission.

---

### Author Response · Authors · 2024-11-25
**TL;DR: Summary of the Rebuttal**

We sincerely thank you for your thorough and insightful feedback on our submission. Your constructive comments have been invaluable in refining our work, and we have addressed each concern meticulously in our detailed rebuttals. Below, we provide a concise summary of our responses, underscoring the significance and contributions of our paper:

* **Enhanced Evaluation and Transferability** (Reviewers 2nhH & L8oC):

  * **Comprehensive Fine-Tuning Results**: We conducted extensive experiments demonstrating that fine-tuning on the VCR-wiki dataset not only improves performance on the VCR task but also enhances model capabilities across various benchmarks, including OCR-Bench and real-world scenarios. These findings, detailed in Appendices D and E, highlight the dataset's robustness and practical applicability.
  * **Real-World Applicability**: Through a case study with real-world occluded text images, we showcased significant performance improvements post fine-tuning, affirming the dataset's relevance beyond synthetic benchmarks.

* **Dataset Design and Masking Strategy** (Reviewers 2nhH & kwun):

  * **Rationale for 5-gram Masking**: We clarified that the 5-gram masking balances linguistic meaningfulness and   empirical feasibility, capturing essential grammatical structures while maintaining task difficulty. This approach ensures the task remains challenging yet attainable.
  * **Distinct Easy and Hard Settings**: Both settings utilize the same data with varying degrees of pixel masking, ensuring direct comparability. This design choice was emphasized to clarify the dataset's structure and the controlled difficulty levels. The fully controlled easy and hard settings, VI+TEI and VI settings, and the "finetune-ability" experiment between pre- and post-finetuning provides directly comparible results for directed guidance for future VLM developments.

* **Fine-Tuning Protocol and Experimental Setup** (Reviewers 2nhH & kwun):

  * **Methodological Clarity**: We provided detailed explanations of our fine-tuning procedures, including the selection of a representative subset and the decision to use a single, standardized prompt. This ensures reproducibility and consistency across evaluations.
  * **Resource Considerations**: Acknowledging resource constraints, we emphasized that our fine-tuning experiments effectively highlight diverse model behaviors, laying the groundwork for future, more extensive studies.

* **Generalizability and Real-World Applicability** (Reviewer L8oC):

  * **Case Study Inclusion**: By adding a real-world case study in Appendix E, we demonstrated tangible improvements in models' ability to interpret occluded text in practical settings, reinforcing the dataset's utility and the task's relevance.

* **Robustness of Evaluation Metrics** (Reviewer L8oC):

  * **Appropriate Metrics Selection**: We justified the use of Exact Match and Jaccard Index as robust, objective metrics tailored to the VCR task's requirements. This ensures reliable and consistent evaluation aligned with the task's objectives.

* **Additional Enhancements and Flexibility** (All Reviewers):

  * **Dataset Flexibility**: We highlighted the VCR framework's adaptability, allowing researchers to customize masking configurations and explore domain-specific challenges. This flexibility fosters broader research applications and future innovations.
  * **Codebase Improvements**: Incorporating suggestions, we enhanced our codebase to support more nuanced masking strategies, such as whitelisting and blacklisting specific words, paving the way for more refined and targeted studies.

In summary, VCR presents a **pioneering vision-language task that bridges the gap between visual cues and textual restoration**. Our meticulously designed dataset, coupled with comprehensive evaluations, reveals significant performance gaps in current models, underscoring the task's challenge and the dataset's value. Besides, the VCR-wiki dataset provides **fully comparable configurations** for difficulty, image layout and finetune-ability **for more targeted model improvements**. By releasing VCR-wiki and the accompanying code, we aim to catalyze future research, fostering advancements that move vision-language models closer to human-like multimodal understanding.

We are confident that our revisions and detailed responses address your concerns comprehensively, highlighting the paper's contributions and its potential impact on the field. We appreciate your consideration and hope that our efforts have clarified the strengths and significance of our work.

Thank you for your time and valuable feedback.

---

### Meta-Review · Area_Chair_RriA · 2024-12-21

**Metareview:**

This paper introduces Visual Caption Restoration (VCR), a new vision-language task designed to evaluate how well models can restore partially obscured text within images using both pixel-level hints (visible parts of the characters) and the surrounding visual context. Unlike traditional OCR, VCR requires models to integrate visual and textual information to reconstruct fragmented or hidden text, mirroring human abilities. The authors created VCR-Wiki, a large-scale synthetic dataset derived from Wikipedia image-caption pairs, available in both English (2.11M samples) and Chinese (346K samples) and in "easy" and "hard" versions based on the degree of text obscuration.  Experiments are conducted on state-of-the-art (SOTA) VLMs to evaluate their performance.

The reviewers agree this is a novel and important task with well-defined methodology and dataset. The experiments and analysis are interesting and insightful.

**Additional Comments On Reviewer Discussion:**

The reviewers agree this could be a nice contribution to the community.

---

### Decision · Program_Chairs · 2025-01-22

Accept (Poster)